# THEORETICAL ANALYSIS OF ROBUST OVERFITTING FOR WIDE DNNS: AN NTK APPROACH

**Shaopeng Fu & Di Wang**
Provable Responsible AI and Data Analytics (PRADA) Lab
King Abdullah University of Science and Technology, Saudi Arabia
`{shaopeng.fu, di.wang}@kaust.edu.sa`

## ABSTRACT

Adversarial training (AT) is a canonical method for enhancing the robustness of deep neural networks (DNNs). However, recent studies empirically demonstrated that it suffers from *robust overfitting*, *i.e.*, a long time AT can be detrimental to the robustness of DNNs. This paper presents a theoretical explanation of robust overfitting for DNNs. Specifically, we non-trivially extend the neural tangent kernel (NTK) theory to AT and prove that an adversarially trained *wide* DNN can be well approximated by a linearized DNN. Moreover, for squared loss, closed-form AT dynamics for the linearized DNN can be derived, which reveals a new ***AT degeneration*** phenomenon: a long-term AT will result in a wide DNN degenerates to that obtained without AT and thus cause robust overfitting. Based on our theoretical results, we further design a method namely ***Adv-NTK***, the first AT algorithm for infinite-width DNNs. Experiments on real-world datasets show that Adv-NTK can help infinite-width DNNs enhance comparable robustness to that of their finite-width counterparts, which in turn justifies our theoretical findings. The code is available at `https://github.com/fshp971/adv-ntk`.

## 1 INTRODUCTION

Despite the advancements of deep neural networks (DNNs) in real-world applications, they are found to be vulnerable to *adversarial attacks*. By adding specially designed noises, one can transform clean data to *adversarial examples* to fool a DNN to behave in unexpected ways (Szegedy et al., 2013; Goodfellow et al., 2015). To tackle the risk, one of the most effective defenses is *adversarial training* (AT), which enhances the robustness of DNNs against attacks via training them on adversarial examples (Madry et al., 2018). However, recent study shows that AT suffers from *robust overfitting*: after a certain point in AT, further training will continue to degrade the robust generalization ability of DNNs (Rice et al., 2020). This breaks the common belief of "training long and generalize well" in deep learning and raises security concerns on real-world deep learning systems.

While a line of methods has been developed to mitigate robust overfitting in practice (Yu et al., 2022; Chen et al., 2021; Wu et al., 2020; Li & Spratling, 2023), recent studies attempt to theoretically understand the mechanism behind robust overfitting. However, existing theoretical results mainly focus on analyzing the robustness of machine learning models that have already been trained to converge but overlook the changes of robustness during AT (Min et al., 2021; Bombari et al., 2023; Zhang & Li, 2023; Clarysse et al., 2023). More recently, a work by Li & Li (2023) has started to incorporate the training process into the study of robust overfitting. However, their analysis currently only applies to two-layer neural networks. Thus, we still cannot answer the question: *Why a DNN would gradually lose its robustness gained from the early stage of AT during continuous training?*

Motivated by the recent success of neural tangent kernel (NTK) theory (Jacot et al., 2018) in approximating wide DNNs in standard training with closed-form training dynamics (Lee et al., 2019), this paper makes the first attempt to address the raised question by non-trivially extending NTK to theoretically analyze the AT dynamics of wide DNNs. Our main result is that for an adversarially trained multilayer perceptron (MLP) with any (finite) number of layers, as the widths of layers approach infinity, the network can be approximated by its linearized counterpart derived from Taylor expansion. When the squared loss is used, we further derive closed-form AT dynamics for the linearized MLP.

The key challenge of our theory arises from the process of searching adversarial examples in AT. In the vanilla AT, the strength of adversarial examples used for training is controlled by searching them within constrained spaces. But such a constrained-spaces condition prevents one from conducting continuous gradient flow-based NTK analysis on that search process. We propose a general strategy to remove this condition from AT by introducing an additional learning rate term into the search process to control the strength of adversarial examples. With our solution, one can now characterize the behavior of DNNs in AT by directly studying the gradient flow descent in AT with NTK.

Our theory then reveals a new ***AT degeneration*** phenomenon that we believe is the main cause of robust overfitting in DNNs. In detail, our theory suggests that the effect of AT on a DNN can be characterized by a regularization matrix introduced into the linearized closed-form AT dynamics, which however will gradually fade away in long-term AT. In other words, a long-term AT will result in an adversarially trained DNN *degenerate* to that obtained without AT, which thus explains why the DNN will lose its previously gained robustness. Based on our analysis, we further propose ***Adv-NTK***, the first AT algorithm for infinite-width DNNs which improves network robustness by directly optimizing the introduced regularization matrix. Experiments on real-world datasets demonstrate that Adv-NTK can help infinite-width DNNs gain robustness that is comparable with finite-width ones, which in turn justifies our theoretical findings.

In summary, our work has three main contributions: (1) We proved that a wide DNN in AT can be strictly approximated by a linearized DNN with closed-form AT dynamics. (2) Our theory reveals a novel ***AT degeneration*** phenomenon that theoretically explains robust overfitting of DNNs for the first time. (3) We designed ***Adv-NTK***, the first AT algorithm for infinite-width DNNs.

## 2 RELATED WORKS

**Robust overfitting.** Rice et al. (2020) first find this phenomenon in adversarially trained DNNs. A series of works then design various regularization techniques to mitigate it in practice (Zhang et al., 2021; Yu et al., 2022; Wu et al., 2020; Li & Spratling, 2023; Chen et al., 2021). Recent studies attempt to theoretically explain robust overfitting. Donhauser et al. (2021) and Hassani & Javanmard (2022) show that the robust generalizability follows a double descent phenomenon concerning the model scale. Wang et al. (2022) show that a two-layer neural network that is closed to the initialization and with frozen second-layer parameter is provably vulnerable to adversarial attacks. However, this result requires the inputs coming from the unit sphere, which is not realistic in the real world. Other advances include Zhu et al. (2022), Bombari et al. (2023), Bubeck et al. (2021), Zhang & Li (2023) and Clarysse et al. (2023). Since these works only focus on analyzing converged models, it remains unclear how robustness of DNN occurs and degrades during AT.

Li & Li (2023) are the first that consider the AT evolution process in studying robust overfitting. Based on the feature learning theory, they find a two-layer CNN in AT will gradually memorize data-wise random noise in adversarial training data, which makes it difficult to generalize well on unseen adversarial data. However, their theory currently is only applicable to shallow networks, and could not explain why networks will lose previously acquired robustness with further AT.

**Neural tangent kernel (NTK).** Jacot et al. (2018) show that for a wide neural network, its gradient descent dynamics can be described by a kernel, named neural tangent kernel (NTK). Based on NTK, the learning process of neural networks can be simplified as a linear kernel regression (Jacot et al., 2018; Lee et al., 2019), which makes NTK a suitable theoretical tool to analyze overparameterized models (Li & Liang, 2018; Zou et al., 2020; Allen-Zhu et al., 2019). Recent studies have extended NTK to various model architectures (Arora et al., 2019; Hron et al., 2020; Du et al., 2019a; Lee et al., 2022), and the theory itself helps understand deep learning from various aspects such as convergence (Du et al., 2019b; Cao & Gu, 2019), generalization (Lai et al., 2023; Huang et al., 2020; Chen et al., 2020; Barzilai et al., 2023; Hu et al., 2020), and trainability (Xiao et al., 2020).

NTK has also been used to analyze AT on overparameterized models. Gao et al. (2019) and Zhang et al. (2020) study the convergence of overparameterized networks in AT and prove upper bounds on the time required for AT. More recent works empirically study robust overfitting with NTK. Tsilivis & Kempe (2022) use eigenspectrums of NTKs to identify robust or non-robust features. Loo et al. (2022) empirically show that a finite-width NTK in AT will rapidly converge to a kernel encodes robust features. But none of them provide theoretical explanations of robust overfitting.

## 3 PRELIMINARIES

**Notations.** Let $\otimes$ denotes Kronecker product, $\mathrm{Diag}(\cdot)$ denotes a diagonal matrix constructed from a given input, $\partial_{(\cdot)}(\cdot)$ denotes the Jacobian of a given function, $\lambda_{\max}(\cdot)$ denotes the maximum eigenvalue of a given matrix, and $I_n$ ($n \in \mathbb{N}^+$) denotes an $n \times n$ identity matrix. For a set of random variables $X_n$ indexed by $n$ and an additional random variable $X$, we use $X_n \xrightarrow{P} X$ to denote that $X_n$ converges in probability to $X$. See Appendices A.2 and A.3 for a full list of notations and definitions of convergence in probability and Lipschitz continuity/smoothness.

Let $\mathcal{D} = \{(x_1, y_1), \cdots (x_M, y_M)\}$ be a dataset consists of $M$ samples, where $x_i \in \mathcal{X} \subseteq \mathbb{R}^d$ is the $i$-th input feature vector and $y_i \in \mathcal{Y} \subseteq \mathbb{R}^c$ is its label. A parameterized DNN is denoted as $f(\theta, \cdot) : \mathcal{X} \to \mathcal{Y}$, where $\theta$ is the model parameter. For simplicity, we let $\mathbf{x} := \oplus_{i=1}^M x_i \in \mathbb{R}^{Md}$ denotes the concatenation of inputs and $\mathbf{y} := \oplus_{i=1}^M y_i \in \mathbb{R}^{Mc}$ denotes the concatenation of labels. Thereby, the concatenation of $f(\theta, x_1), \cdots f(\theta, x_M)$ can be further denoted as $f(\theta, \mathbf{x}) := \oplus_{i=1}^M f(x_i) \in \mathbb{R}^{Mc}$.

**Adversarial training (AT).** Suppose $\mathcal{L} : \mathbb{R}^c \times \mathbb{R}^c \to \mathbb{R}^+$ is a loss function. Then, a standard AT improves the robustness of DNNs against adversarial attacks by training them on *most* adversarial examples. Specifically, it aims to solve the following minimax problem (Madry et al., 2018),

$$\min_\theta \frac{1}{|\mathcal{D}|} \sum_{(x_i, y_i) \in \mathcal{D}} \max_{\|x_i' - x_i\| \leq \rho} \mathcal{L}(f(\theta, x_i'), y_i), \tag{1}$$

where $\rho \in \mathbb{R}$ is the adversarial perturbation radius and $x_i'$ is the most adversarial example within the ball sphere centered at $x_i$. Intuitively, a large perturbation radius $\rho$ would result in the final model achieving strong adversarial robustness.

**Neural tangent kernel (NTK).** For a DNN $f$ that is trained according to some iterative algorithm, let $f_t := f(\theta_t, \cdot)$ where $\theta_t$ is the DNN parameter obtained at the training time $t$. Then, the *empirical* NTK of the DNN at time $t$ is defined as below (Jacot et al., 2018),

$$\hat{\Theta}_{\theta, t}(x, x') := \partial_\theta f_t(x) \cdot \partial_\theta^T f_t(x') \in \mathbb{R}^{c \times c}, \quad \forall x, x' \in \mathcal{X}. \tag{2}$$

In the rest of the paper, we will also use the notations $\hat{\Theta}_{\theta, t}(x, \mathbf{x}) := \partial_\theta f_t(x) \cdot \partial_\theta^T f_t(\mathbf{x}) \in \mathbb{R}^{c \times Mc}$ and $\hat{\Theta}_{\theta, t}(\mathbf{x}, \mathbf{x}) := \partial_\theta f_t(\mathbf{x}) \cdot \partial_\theta^T f_t(\mathbf{x}) \in \mathbb{R}^{Mc \times Mc}$.

When $f_t$ is trained via minimizing the empirical squared loss $\sum_{(x_i, y_i) \in \mathcal{D}} \frac{1}{2} \|f(x_i) - y_i\|_2^2$, Lee et al. (2019) show that it can be approximated by the linearized DNN $f_t^{\mathrm{lin}} : \mathcal{X} \to \mathcal{Y}$ defined as follows,

$$f_t^{\mathrm{lin}}(x) := f_0(x) - \hat{\Theta}_{\theta, 0}(x, \mathbf{x}) \cdot \hat{\Theta}_{\theta, 0}^{-1}(\mathbf{x}, \mathbf{x}) \cdot \left( I - e^{-\hat{\Theta}_{\theta, 0}(\mathbf{x}, \mathbf{x}) \cdot t} \right) \cdot (f_0(\mathbf{x}) - \mathbf{y}), \quad \forall x \in \mathcal{X}. \tag{3}$$

Although the kernel function $\hat{\Theta}_{\theta, t}$ depends on both the initial parameter $\theta_0$ and the time $t$, Jacot et al. (2018) prove that $\hat{\Theta}_{\theta, t} \xrightarrow{P} \Theta_\theta$ as the network widths go to infinity, where $\Theta_\theta$ is a kernel function that is independent of $\theta_0$ and $t$. Based on it, Lee et al. (2019) show that with infinite training time, the average output of infinite-width DNNs over random initialization will converge as follows,

$$\lim_{\text{widths} \to \infty} \lim_{t \to \infty} \mathbb{E}_{\theta_0}[f_t(x)] \xrightarrow{P} \Theta_\theta(x, \mathbf{x}) \cdot \Theta_\theta^{-1}(\mathbf{x}, \mathbf{x}) \cdot \mathbf{y}, \quad \forall x \in \mathcal{X}, \tag{4}$$

where $\Theta_\theta(x, \mathbf{x}) \in \mathbb{R}^{c \times Mc}$ is an $1 \times M$ block matrix that the $i$-th column block is $\Theta_\theta(x, x_i)$, and $\Theta_\theta(\mathbf{x}, \mathbf{x}) \in \mathbb{R}^{Mc \times Mc}$ is an $M \times M$ block matrix that the $i$-th row $j$-th column block is $\Theta_\theta(x_i, x_j)$.

## 4 ADVERSARIAL TRAINING OF WIDE DNNS

In this section, we present our main theoretical results that characterize AT dynamics of wide DNNs. We first introduce the DNN architectures that we are going to analyze.

Suppose $f(\theta, \cdot)$ is a DNN consisting of $L + 1$ fully connected layers, in which the width of the $l$-th *hidden* layer ($1 \leq l \leq L$) is $n_l$. Additionally, the input dimension and the output dimension are denoted as $n_0 := d$ and $n_{L+1} := c$ for simplicity. Then, the forward propagation in the $l$-th fully-connected layer ($1 \leq l \leq L + 1$) is calculated as follows,

$$h^{(l)}(x) = \frac{1}{\sqrt{n_{l-1}}} W^{(l)} \cdot x^{(l-1)}(x) + b^{(l)}, \quad x^{(l)}(x) = \phi(h^{(l)}(x)), \tag{5}$$

where $h^{(l)}$ and $x^{(l)}$ are the pre-activated and post-activated functions at the $l$-th layer, $W^{(l)} \in \mathbb{R}^{n_l \times n_{l-1}}$ is the $l$-th weight matrix, $b^{(l)} \in \mathbb{R}^{n_l}$ is the $l$-th bias vector, and $\phi$ is a point-wise activation function. The final DNN function is $f(\theta, \cdot) := h^{(L+1)}(\cdot)$, with model parameter $\theta := (W^{(1)}, \cdots, W^{(L+1)}, b^{(1)}, \cdots, b^{(L+1)})$, and we use $f_t := f(\theta_t, \cdot)$ denote the DNN at the training time $t$. Finally, for the initialization of $\theta_0$, we draw each entry of weight matrices from a Gaussian $\mathcal{N}(0, \sigma_W^2)$ and each entry of bias vectors from a Gaussian $\mathcal{N}(0, \sigma_b^2)$.

Similar to existing NTK literatures (Jacot et al., 2018; Lee et al., 2019; Arora et al., 2019), we are also interested in the linearized DNN $f_t^{\text{lin}}$ defined as follows,

$$f_t^{\text{lin}}(x) = f_0(x) + \partial_\theta f_0(x) \cdot (\theta_t^{\text{lin}} - \theta_0), \quad \forall x \in \mathcal{X}, \tag{6}$$

where $\theta_0$ is the initial parameter same as that of the non-linear DNN $f_t$ and $\theta_t^{\text{lin}}$ is the parameter of the linearized DNN at the training time $t$.

## 4.1 GRADIENT FLOW-BASED ADVERSARIAL EXAMPLE SEARCH

To characterize the training dynamics of AT, the key step is to analyze the process of searching adversarial examples that will be used to calculate AT optimization directions. Recall Eq. (1), standard minimax AT will search adversarial examples within constrained spaces. However, analyzing such a process with continuous gradient flows is challenging due to the need to explicitly model the boundaries of those constrained spaces.

To tackle the challenge, we notice that the main role of the constrained-spaces condition is to control the adversarial strength (*i.e.*, the strength of the ability to make models misbehave) of searched data. As a solution, we suggest replacing the constrained-spaces condition (controlled by $\rho$ in Eq. (1)) with an additional learning rate term (*i.e.*, $\eta_i(t)$ in Eq. (7)) to control the strength of adversarial examples. This modification will then enable a more convenient continuous gradient flow analysis.

Specifically, for the DNN $f_t$ at the training time $t$, to find the corresponding adversarial example of the $i$-th training data point $(x_i, y_i)$ where $1 \leq i \leq M$, we will start from $x_i$ and perform gradient flow ascent for a total of time $S > 0$, with an introduced learning rate $\eta_i(t) : \mathbb{R} \to \mathbb{R}$ to control the adversarial strength of searched example at the current training time $t$, as below,

$$\partial_s x_{i,t,s} = \eta_i(t) \cdot \partial_x^T f_t(x_{i,t,s}) \cdot \partial_{f(x)}^T \mathcal{L}(f_t(x_{i,t,s}), y_i) \qquad \text{s.t.} \qquad x_{i,t,0} = x_i. \tag{7}$$

Then, the final adversarial example is $x_{i,t,S}$. The learning rate $\eta_i(t)$ plays a similar role with the constrained-spaces condition in Eq. (1). Intuitively, a larger $\eta_i(t)$ at the training time $t$ corresponds to a more adversarial example $x_{i,t,S}$.

Besides, running the gradient flow defined in Eq. (7) also depends on the DNN output of which the evolution of $f_t(x_{i,t,s})$ concerning $s$ can be formalized based on Eq. (7) as follows,

$$\partial_s f_t(x_{i,t,s}) = \partial_x f_t(x_{i,t,s}) \cdot \partial_s x_{i,t,s} = \eta_i(t) \cdot \hat{\Theta}_{x,t}(x_{i,t,s}, x_{i,t,s}) \cdot \partial_{f(x)}^T \mathcal{L}(f_t(x_{i,t,s}), y_i), \tag{8}$$

where $\hat{\Theta}_{x,t} : \mathcal{X} \times \mathcal{X} \to \mathbb{R}^{c \times c}$ is a new kernel function named *Adversarial Regularization Kernel (ARK)* and defined as below,

$$\hat{\Theta}_{x,t}(x, x') := \partial_x f_t(x) \cdot \partial_x^T f_t(x'), \qquad \forall x, x' \in \mathcal{X}. \tag{9}$$

The ARK $\hat{\Theta}_{x,t}$ shares similar structure with the NTK $\hat{\Theta}_{\theta,t}$ defined in Eq. (2). The difference is that the kernel matrix $\hat{\Theta}_{x,t}$ is calculated from Jacobians of DNN $f_t$ concerning input, while the NTK $\hat{\Theta}_{\theta,t}$ is calculated from Jacobians concerning the model parameter.

## 4.2 ADVERSARIAL TRAINING DYNAMICS

With the gradient flow-based adversarial example search in the previous section, we now formalize the gradient flow-based AT dynamics for the wide DNN $f_t$ and the linearized DNN $f_t^{\text{lin}}$ respectively.

**AT dynamics of wide DNN $f_t$.** Suppose $f_t$ is trained via continuous gradient flow descent. Then, the evolution of model parameter $\theta_t$ and model output $f_t(x)$ are formalized as follows,

$$\partial_t \theta_t = -\partial_\theta^T f_t(\mathbf{x}_{t,S}) \cdot \partial_{f(\mathbf{x})}^T \mathcal{L}(f_t(\mathbf{x}_{t,S}), \mathbf{y}), \tag{10}$$

$$\partial_t f_t(x) = \partial_\theta f_t(x) \cdot \partial_t \theta_t = -\hat{\Theta}_{\theta,t}(x, \mathbf{x}_{t,S}) \cdot \partial_{f(\mathbf{x})}^T \mathcal{L}(f_t(\mathbf{x}_{t,S}), \mathbf{y}), \quad \forall x \in \mathcal{X}, \tag{11}$$

where $\mathbf{x}_{t,S} := \oplus_{i=1}^M x_{i,t,S}$ is the concatenation of adversarial examples found at the current training time $t$, and $\hat{\Theta}_{\theta,t}$ is the empirical NTK defined in Eq. (2).

Meanwhile, based on the method proposed in Section 4.1, the gradient flow-based search process for the concatenation of adversarial examples $\mathbf{x}_{t,S}$ is formalized as below,

$$\partial_s \mathbf{x}_{t,s} = \partial_{\mathbf{x}}^T f_t(\mathbf{x}_{t,s}) \cdot \boldsymbol{\eta}(t) \cdot \partial_{f(\mathbf{x})}^T \mathcal{L}(f_t(\mathbf{x}_{t,s}), \mathbf{y}) \qquad \text{s.t.} \qquad \mathbf{x}_{t,0} = \mathbf{x}, \tag{12}$$

$$\partial_s f_t(\mathbf{x}_{t,s}) = \partial_{\mathbf{x}} f_t(\mathbf{x}_{t,s}) \cdot \partial_s \mathbf{x}_{t,s} = \hat{\Theta}_{x,t}(\mathbf{x}_{t,s}, \mathbf{x}_{t,s}) \cdot \boldsymbol{\eta}(t) \cdot \partial_{f(\mathbf{x})}^T \mathcal{L}(f_t(\mathbf{x}_{t,s}), \mathbf{y}), \tag{13}$$

where $\mathbf{x}_{t,s} := \oplus_{i=1}^M x_{i,t,s}$ is the concatenation of intermediate adversarial training examples found at the search time $s$, $\boldsymbol{\eta}(t) := \text{Diag}(\eta_1(t), \cdots, \eta_M(t)) \otimes I_c \in \mathbb{R}^{Mc \times Mc}$ is a block diagonal learning rate matrix, and $\hat{\Theta}_{x,t}(\mathbf{x}_{t,s}, \mathbf{x}_{t,s}) := \text{Diag}(\hat{\Theta}_{x,t}(x_{1,t,s}, x_{1,t,s}), \cdots, \hat{\Theta}_{x,t}(x_{M,t,s}, x_{M,t,s})) \in \mathbb{R}^{Mc \times Mc}$ is a block diagonal matrix consists of ARKs.

**AT dynamics of linearized wide DNN $f_t^{\text{lin}}$.** Suppose $f_t^{\text{lin}}$ is also trained via continuous gradient flow descent. Then, according to the definition of linearized DNN in Eq. (6), we have $\partial_\theta f_t^{\text{lin}} := \partial_\theta f_0$. Therefore, the evolution of parameter $\theta_t^{\text{lin}}$ and output $f_t^{\text{lin}}(x)$ are formalized as follows,

$$\partial_t \theta_t^{\text{lin}} = -\partial_\theta^T f_0(\mathbf{x}) \cdot \partial_{f(\mathbf{x})}^T \mathcal{L}(f_t^{\text{lin}}(\mathbf{x}_{t,S}^{\text{lin}}), \mathbf{y}), \tag{14}$$

$$\partial_t f_t^{\text{lin}}(x) = \partial_\theta f_0(x) \cdot \partial_t \theta_t^{\text{lin}} = -\hat{\Theta}_{\theta,0}(x, \mathbf{x}) \cdot \partial_{f(\mathbf{x})}^T \mathcal{L}(f_t^{\text{lin}}(\mathbf{x}_{t,S}^{\text{lin}}), \mathbf{y}), \quad \forall x \in \mathcal{X}, \tag{15}$$

where $\mathbf{x}_{t,S}^{\text{lin}} := \oplus_{i=1}^M x_{i,t,S}^{\text{lin}}$ is concatenation of the adversarial examples found for the linearized DNN $f_t^{\text{lin}}$, and $\hat{\Theta}_{\theta,0}$ is the empirical NTK (see Eq. (2)) at initialization.

The search of $\mathbf{x}_{t,S}^{\text{lin}}$ is slightly different from the gradient flow-based method in Section 4.1. When following Eqs. (7) and (8) to search $x_{i,t,s}^{\text{lin}}$, one needs to calculate an intractable Jacobian $\partial_x f_t^{\text{lin}}(x_{i,t,s}^{\text{lin}}) = \partial_x f_0(x_{i,t,s}^{\text{lin}}) + \partial_x(\partial_\theta f_0(x_{i,t,s}^{\text{lin}})(\theta_t - \theta_0))$. To further simplify our analysis, we note that in standard training, a wide DNN is approximately linear concerning model parameters, thus it is also reasonable to deduce that a wide DNN in AT is approximately linear concerning slightly perturbed adversarial inputs. In other words, we deduce that $\partial_x f_t^{\text{lin}}(x_{i,t,s}^{\text{lin}}) \approx \partial_x f_0(x_{i,t,s}^{\text{lin}}) + 0 \approx \partial_x f_0(x_i)$. Thus, we propose to replace $\partial_x f_t^{\text{lin}}(x_{i,t,s}^{\text{lin}})$ with $\partial_x f_0(x_i)$ in the search of $x_{i,t,s}^{\text{lin}}$.

Then, by replacing $\partial_x f_t^{\text{lin}}(x_{i,t,s}^{\text{lin}})$ with $\partial_x f_0(x_i)$ in Eqs.(7) and (8), the overall search process for $\mathbf{x}_{t,s}^{\text{lin}}$ in the linearized AT dynamics is formalized as below,

$$\partial_s \mathbf{x}_{t,s}^{\text{lin}} = \partial_x^T f_0(\mathbf{x}) \cdot \boldsymbol{\eta}(t) \cdot \partial_{f(\mathbf{x})}^T \mathcal{L}(f_t(\mathbf{x}_{t,s}^{\text{lin}}), \mathbf{y}) \qquad \text{s.t.} \qquad \mathbf{x}_{t,0}^{\text{lin}} = \mathbf{x}, \tag{16}$$

$$\partial_s f_t^{\text{lin}}(\mathbf{x}_{t,s}^{\text{lin}}) = \partial_{\mathbf{x}} f_0(\mathbf{x}) \cdot \partial_s \mathbf{x}_{t,s}^{\text{lin}} = \hat{\Theta}_{x,0}(\mathbf{x}, \mathbf{x}) \cdot \boldsymbol{\eta}(t) \cdot \partial_{f(\mathbf{x})}^T \mathcal{L}(f_t^{\text{lin}}(\mathbf{x}_{t,s}^{\text{lin}}), \mathbf{y}), \tag{17}$$

where $\mathbf{x}_{t,s}^{\text{lin}} := \oplus_{i=1}^M x_{i,t,s}^{\text{lin}}$ is the concatenation of intermediate adversarial examples for the linearized DNN $f_t^{\text{lin}}$, the learning rate matrix $\boldsymbol{\eta}(t)$ is same as that in the AT dynamics of $f_t$, and $\hat{\Theta}_{x,0}(\mathbf{x}, \mathbf{x}) := \text{Diag}(\hat{\Theta}_{x,0}(x_1, x_1), \cdots, \hat{\Theta}_{x,0}(x_M, x_M))$ is a block matrix consists of ARKs at initialization.

## 4.3 Adversarial Training in Infinite-Width

This section theoretically characterizes the AT dynamics of the DNN $f_t$ when the network widths approach the infinite limit. We first prove the kernel limits at initialization as Theorem 1.

**Theorem 1** (Kernels limits at initialization; Informal version of Theorem B.1). *Suppose $f_0$ is an MLP defined and initialized as in Section 4. Then, for any $x, x' \in \mathcal{X}$ we have*

$$\lim_{n_L \to \infty} \cdots \lim_{n_0 \to \infty} \hat{\Theta}_{\theta,0}(x, x') = \Theta_\theta(x, x') := \Theta_\theta^\infty(x, x') \cdot I_{n_{L+1}},$$

$$\lim_{n_L \to \infty} \cdots \lim_{n_0 \to \infty} \hat{\Theta}_{x,0}(x, x') = \Theta_x(x, x') := \Theta_x^\infty(x, x') \cdot I_{n_{L+1}},$$

*where $\Theta_\theta^\infty : \mathcal{X} \times \mathcal{X} \to \mathbb{R}$ and $\Theta_x^\infty : \mathcal{X} \times \mathcal{X} \to \mathbb{R}$ are two deterministic kernel functions.*

The proof is given in Appendix B.

**Remark 1.** *The convergence of NTK $\hat{\Theta}_{\theta,0}$ is first proved in Jacot et al. (2018). We restate it for the sake of completeness. Note that the limit of NTK $\hat{\Theta}_{\theta,0}$ in AT is the same as that in standard training.*

We then prove that a wide DNN $f_t$ can be approximated by its linearized counterpart $f_t^{\mathrm{lin}}$, as shown in Theorem 2. It relies on the following Assumptions 1-4.

**Assumption 1.** *The activation function $\phi : \mathbb{R} \to \mathbb{R}$ is twice-differentiable, $K$-Lipschitz continuous, $K$-Lipschitz smooth, and satisfies $|\phi(0)| < +\infty$.*

**Assumption 2.** *For any fixed $T > 0$, we have that $\int_0^T \|\partial_{f(\mathbf{x})}\mathcal{L}(f(\mathbf{x}_{t,S}, \mathbf{y})\|_2 \mathrm{d}t = O_p(1)$, $\sup_{t\in[0,T]} \int_0^S \|\partial_{f(\mathbf{x})}\mathcal{L}(f(\mathbf{x}_{t,s}), \mathbf{y})\|_2 \mathrm{d}s = O_p(1)$, and $\sup_{t\in[0,T]} \int_0^S \|\partial_t \partial_{f(\mathbf{x})}\mathcal{L}(f(\mathbf{x}_{t,s}), \mathbf{y})\|_2 \mathrm{d}s = O_p(1)$ as $\min\{n_0, \cdots, n_L\} \to \infty$.*

**Assumption 3.** *$\boldsymbol{\eta}(t)$ and $\partial_t \boldsymbol{\eta}(t)$ are continuous on $[0, +\infty)$.*

**Assumption 4.** *The loss function $\mathcal{L} : \mathcal{Y} \times \mathcal{Y} \to \mathbb{R}$ is $K$-Lipschitz smooth.*

Assumptions 1 and 4 are commonly used in existing NTK literatures (Jacot et al., 2018; Lee et al., 2019; 2022). Assumption 3 is mild. Assumption 2 assumes that the cumulated perturbation loss directions as well as AT loss directions are stochastically bounded. Similar assumptions are also widely adopted in NTK studies for standard training.

**Theorem 2** (Equivalence between wide DNN and linearized DNN). *Suppose Assumptions 1-4 hold, and $f_t$ and $f_t^{\mathrm{lin}}$ are trained following the AT dynamics formalized in Section 4.2. Then, if there exists $\tilde{n} \in N^+$ such that $\min\{n_0, \cdots, n_L\} \geq \tilde{n}$ always holds, we have for any $x \in \mathcal{X}$, as $\tilde{n} \to \infty$,*

$$\left\{ \lim_{n_L \to \infty} \cdots \lim_{n_0 \to \infty} \sup_{t\in[0,T]} \|f_t(x) - f_t^{\mathrm{lin}}(x)\|_2 \right\} \xrightarrow{P} 0.$$

The overall proof is presented in Appendix C.

**Remark 2.** *Although our AT dynamics is formed based on the intuition that wide DNNs could be linear concerning slightly perturbed inputs, Theorem 2 does not depend on this intuition. It mainly depends on the large-width condition and also holds when large perturbations present.*

Finally, we calculate the closed-form AT dynamics for the linearized DNN $f_t^{\mathrm{lin}}$ (Theorem 3) as well as the infinite-width DNN $f_t$ (Corollary 1) when squared loss $\mathcal{L}(f(x), y) := \frac{1}{2}\|f(x) - y\|_2^2$ is used.

**Theorem 3** (Close-form AT-dynamics of $f_t^{\mathrm{lin}}$ under squared loss). *Suppose Assumption 3 holds and the linearized DNN $f_t^{\mathrm{lin}}$ is trained following the AT dynamics formalized in Section 4.2 with squared loss $\mathcal{L}(f(x), y) := \frac{1}{2}\|f(x) - y\|_2^2$. Then, for any $x \in \mathcal{X}$, we have*

$$f_t^{\mathrm{lin}}(x) = f_0(x) - \hat{\Theta}_{\theta,0}(x, \mathbf{x}) \cdot \hat{\Theta}_{\theta,0}^{-1}(\mathbf{x}, \mathbf{x}) \cdot \left( I - e^{-\hat{\Theta}_{\theta,0}(\mathbf{x},\mathbf{x}) \cdot \hat{\Xi}(t)} \right) \cdot (f_0(\mathbf{x}) - \mathbf{y}), \qquad (18)$$

*where $\hat{\Xi}(t) := \mathrm{Diag}(\{\int_0^t \exp(\hat{\Theta}_{x,0}(x_i, x_i) \cdot \eta_i(\tau) \cdot S) \mathrm{d}\tau\}_{i=1}^M) \in \mathbb{R}^{Mc \times Mc}$ is a regularization matrix.*

The proof is given in Appendix D.

**Corollary 1.** *Suppose all conditions in Theorems 1, 2, 3 hold. Then, if there exists $\tilde{n} \in N^+$ such that $\min\{n_0, \cdots, n_L\} \geq \tilde{n}$ always holds, we have for any $x \in \mathcal{X}$, as $\tilde{n} \to \infty$, $\{\lim_{n_L \to \infty} \cdots \lim_{n_0 \to \infty}\{f_t(x), f_t^{\mathrm{lin}}(x)\}\} \xrightarrow{P} f_t^\infty(x)$, and*

$$\mathbb{E}_{\theta_0}[f_t^\infty(x)] = \Theta_\theta(x, \mathbf{x}) \cdot \Theta_\theta^{-1}(\mathbf{x}, \mathbf{x}) \cdot \left( I - e^{-\Theta_\theta(\mathbf{x},\mathbf{x}) \cdot \Xi(t)} \right) \cdot \mathbf{y}, \qquad (19)$$

*where $\Xi(t) := \mathrm{Diag}(\{\int_0^t \exp(\Theta_x(x_i, x_i) \cdot \eta_i(\tau) \cdot S) \mathrm{d}\tau\}_{i=1}^M) \in \mathbb{R}^{Mc \times Mc}$ is a diagonal regularization matrix, and $\Theta_\theta$ and $\Theta_x$ are kernel functions in the infinite-width limit.*

*Proof.* The proof is completed by adopting Theorems 1, 2 and Lemma B.1 into Theorem 3. $\square$

**Remark 3.** *Recall Remark 1, the infinite-width NTK function $\Theta_\theta$ in AT is exactly the same as that in standard training. As a result, in practice $\Theta_\theta$ can be calculated by using the Neural-Tangents Python Library (Novak et al., 2020) and the JAX autograd system (Bradbury et al., 2018).*

## 5 ROBUST OVERFITTING IN WIDE DNNs

So far, we have shown that a wide DNN that is adversarially trained with squared loss can be approximated by its linearized counterpart which admits a closed-form AT dynamics. Now, we leverage our theory to theoretically understand and mitigate robust overfitting in wide DNNs. Throughout this section, the loss function is assumed to be the squared loss $\mathcal{L}(f(x), y) := \frac{1}{2}\|f(x) - y\|_2^2$.

## 5.1 AT DEGENERATION LEADS TO ROBUST OVERFITTING

This section reveals a novel *AT degeneration* phenomenon that theoretically explains the mechanism behind deep robust overfitting. Compared with existing theoretical studies on robust overfitting (Donhauser et al., 2021; Bombari et al., 2023; Zhang & Li, 2023; Clarysse et al., 2023; Li & Li, 2023), our result has two significant advantages: (1) it can explain why the gained robustness will gradually lose in long-term AT, and (2) it applies to general deep neural network models.

We propose to study the AT dynamics of the linearized DNN instead of the original DNN since it is proved in Theorem 2 that a wide DNN can be approximated by the linearized one. Comparing the closed-form dynamics of the linearized DNN in AT (Eq. (18) in Theorem 3) with that in standard training (Eq. (3)), one can find that the difference is AT introduces a time-dependent regularization matrix $\hat{\Xi}(t)$ (Theorem 3) into the closed-form dynamics of standard training. Thus, it can be deduced that the introduced matrix $\hat{\Xi}(t)$ fully captures the adversarial robustness of DNNs brought by AT.

To answer why the robustness captured by $\hat{\Xi}(t)$ will gradually degrade in long-term AT, without loss of generality, we first assume that the ARK $\hat{\Theta}_{x,0}(\mathbf{x}, \mathbf{x})$ is positive definite. Thereby, it can be decomposed as $\hat{\Theta}_{x,0}(\mathbf{x}, \mathbf{x}) := QDQ^T$, where $Q$ is a block diagonal matrix consists of orthogonal blocks and $D$ is a diagonal matrix consists of positive diagonal entries. Since $\boldsymbol{\eta}(t)$ is commutative with $\hat{\Theta}_{x,0}(\mathbf{x}, \mathbf{x})$ and thus also with $Q$, the matrix $\hat{\Xi}(t)$ can be further decomposed as follows,

$$\hat{\Xi}(t) = \int_0^t \exp(QDQ^T \cdot \boldsymbol{\eta}(\tau) \cdot S)\mathrm{d}\tau = \int_0^t Q\exp(D\boldsymbol{\eta}(\tau)S)Q^T\mathrm{d}\tau = QA(t)Q^T \cdot a(t), \quad (20)$$

where $a(t) := \lambda_{\max}\{\int_0^t \exp(D\boldsymbol{\eta}(\tau)S)\mathrm{d}\tau\}$ is a strictly increasing scale function and $A(t) := \frac{1}{a(t)}\int_0^t \exp(D\boldsymbol{\eta}(\tau)S)\mathrm{d}\tau$ is a matrix that $\sup_{t\geq 0}\|QA(t)Q^T\|_2 \leq 1$. The here is to decouple the unbounded $a(t)$ from $\hat{\Xi}(t)$ and remain others being bounded, which can simplify our analysis.

Then, for the exponential term in the AT dynamics in Eq. (18), substituting $\hat{\Xi}(t)$ and we have

$$e^{-\hat{\Theta}_{\theta,0}(\mathbf{x},\mathbf{x})\cdot\hat{\Xi}(t)} = QA(t)^{-\frac{1}{2}} \cdot \exp(-A(t)^{\frac{1}{2}}Q^T \cdot \hat{\Theta}_{\theta,0}(\mathbf{x},\mathbf{x}) \cdot QA(t)^{\frac{1}{2}} \cdot a(t)) \cdot A(t)^{\frac{1}{2}}Q^T, \quad (21)$$

where the calculation details is given in Appendix E.1. We further assume that: (1) the adversarial perturbation scale is small enough such that the symmetric $A(\infty)^{\frac{1}{2}}Q^T\hat{\Theta}_{\theta,0}(\mathbf{x},\mathbf{x})QA(\infty)^{\frac{1}{2}}$ stays positive definite, and (2) $a(t) \to \infty$ as $t \to \infty$. Under the first assumption, we have $A(\infty)^{\frac{1}{2}}Q^T\hat{\Theta}_{\theta,0}(\mathbf{x},\mathbf{x})QA(\infty)^{\frac{1}{2}} := Q'D'Q'^T$ where $Q'$ is an orthogonal matrix and $D'$ is a diagonal matrix consisting of positive diagonal entries. Combined with the second one, we have

$$\exp(-A(\infty)^{\frac{1}{2}}Q^T \cdot \hat{\Theta}_{\theta,0}(\mathbf{x},\mathbf{x}) \cdot QA(\infty)^{\frac{1}{2}} \cdot a(\infty)) = Q'e^{-D'a(\infty)}Q'^T = 0, \quad (22)$$

where the calculation is presented in Appendix E.1. As a result,

$$\lim_{t\to\infty} e^{-\hat{\Theta}_{\theta,0}(\mathbf{x},\mathbf{x})\cdot\hat{\Xi}(t)} = QA(\infty)^{-\frac{1}{2}} \cdot 0 \cdot A(\infty)^{\frac{1}{2}}Q^T = 0. \quad (23)$$

Eq. (23) indicates that in a long-term AT, the regularization matrix $\hat{\Xi}(t)$ which captures robustness brought by AT will gradually fade away. Moreover, when at the infinite training time limit, the AT dynamics will converge to $f_0(x) - \hat{\Theta}_{\theta,0}(x,\mathbf{x}) \cdot \hat{\Theta}_{\theta,0}^{-1}(\mathbf{x},\mathbf{x}) \cdot (f_0(\mathbf{x}) - \mathbf{y})$, which is exactly the same limit as that of the standard training dynamics given in Eq. (3). Notice that the analysis up to now relies on that the adversarial perturbation is small such that the matrix in Eq. (21) is positive definite when $t = \infty$. Please refer to Appendix E.2 for further discussion when the perturbation is large.

In conclusion, the analysis in this section suggests a novel ***AT degeneration*** phenomenon that in a long-term AT, the impact brought by AT will graduate disappear and the adversarially trained DNN will eventually degenerate to that obtained without AT. The AT degeneration phenomenon clearly illustrates the mechanism behind robust overfitting in DNNs. It can also explain the empirical finding in Rice et al. (2020) that early-stop can significantly mitigate robust overfitting: it is because early-stop can effectively preserve the regularization matrix $\hat{\Xi}(t)$ brought by AT.

## 5.2 INFINITE WIDTH ADVERSARIAL TRAINING

We have known that the robustness brought by AT can be characterized by $\hat{\Xi}(t)$ defined in Theorem 3. Then, it is natural to ask if one can directly optimize $\hat{\Xi}(t)$ to mitigate robust overfitting. Since

---

**Algorithm 1** Adv-NTK (Solving Eq. (25) with SGD and GradNorm)

---

**Input:** Training set $\mathcal{D}$, validation set size $M_{\text{val}}$, learning rate $\zeta$, training iteration $T$, PGD function
    for finding adversarial validation data.
**Output:** An infinite-width adversarially robust DNN.
 1: Randomly separate $\mathcal{D}$ into subsets $\mathcal{D}_{\text{opt}}$ and $\mathcal{D}_{\text{val}}$ such that $|\mathcal{D}_{\text{val}}| = M_{\text{val}}$.
 2: Initialize trainable parameter $\varpi_0 \in \mathbb{R}^{|\mathcal{D}_{\text{val}}| \cdot c}$ with zeros.
 3: **for** $t$ **in** $1, \cdots, T$ **do**
 4:     Sample a minibatch $(x, y) \sim \mathcal{D}_{\text{val}}$.
 5:     $x' \leftarrow \text{PGD}(x, y, f_{\varpi_{t-1}})$                    ▷ Finding adversarial validation examples.
 6:     $g_t \leftarrow \partial_\varpi \frac{1}{2} \|f_{\varpi_{t-1}}(x') - y\|_2^2$
 7:     $\varpi_t \leftarrow \varpi_{t-1} - \zeta \cdot \frac{g_t}{\|g_t\|_2}$          ▷ Update model parameter via SGD and $\ell_2$-GardNorm.
 8: **end for**
 9: **return** $f_{\varpi_T}$

---

$\hat{\Xi}(t)$ is a $Mc \times Mc$ block diagonal matrix, optimizing it requires maintaining $Mc^2$ variables and is computationally costly. Fortunately, Corollary 1 indicates that in the infinite-width limit, matrix $\hat{\Xi}(t)$ will converge to a diagonal matrix $\Xi(t)$ where only $Mc$ variables need to be maintained. Based on the observation, we propose ***Adv-NTK***, the first AT algorithm for infinite-width DNNs.

Specifically, for a given training set $\mathcal{D}$, we separate it into two disjoint subsets $\mathcal{D}_{\text{opt}}$ and $\mathcal{D}_{\text{val}}$, in which $\mathcal{D}_{\text{opt}}$ is for constructing the infinite width DNN while $\mathcal{D}_{\text{val}}$ is a validation set for model selection. Then, the infinite-width DNN that will be trained in Adv-NTK is constructed as follows based on the infinite-width DNN defined as Eq. (4) in Corollary 1,

$$f_\varpi(x) = \Theta_\theta(x, \mathbf{x}_{\text{opt}}) \cdot \Theta_\theta^{-1}(\mathbf{x}_{\text{opt}}, \mathbf{x}_{\text{opt}}) \cdot \left( I - e^{-\Theta_\theta(\mathbf{x}_{\text{opt}}, \mathbf{x}_{\text{opt}}) \cdot \text{Diag}(\varpi)} \right) \cdot \mathbf{y}_{\text{opt}}, \quad \forall x \in \mathcal{X}, \quad (24)$$

where $\varpi \in \mathbb{R}^{|\mathcal{D}_{\text{opt}}| \cdot c}$ is the trainable parameter in Adv-NTK, $\Theta_\theta$ is the NTK function at the infinite-width limit (see Theorem 1), and $\mathbf{x}_{\text{opt}}$ and $\mathbf{y}_{\text{opt}}$ are concatenations of features and labels in the subset $\mathcal{D}_{\text{opt}}$. Note that the parameter $\varpi$ consists exactly of the diagonal entries of the diagonal matrix $\Xi(t)$. Then, the Adv-NTK algorithm aims to enhance the adversarial robustness of the infinite-width DNN $f_\varpi$ via solving the following minimax optimization problem,

$$\min_\varpi \frac{1}{|\mathcal{D}_{\text{val}}|} \sum_{(x,y) \in \mathcal{D}_{\text{val}}} \max_{\|x'-x\| \le \rho} \frac{1}{2} \|f_\varpi(x') - y\|_2^2, \quad (25)$$

where $\rho > 0$ is the same adversarial perturbation radius as that in the standard AT (see Eq. (1)), and the inner maximization problem can be solved via projected gradient descent (PGD) (Madry et al., 2018). The above Eq. (25) shares similar idea with the early stop method (Rice et al., 2020): they both use a validation set for model selection. The difference is that early stop uses model robust accuracy on the validation set as an indicator to select the model parameter indirectly, while Eq. (25) directly optimizes the model parameter with the validation set. Finally, to further improve the training stability, Adv-NTK leverages stochastic gradient descent (SGD) and gradient normalization (GradNorm) to solve Eq. (25). The overall procedures are presented as Algorithm 1.

## 6 EMPIRICAL ANALYSIS OF ADV-NTK

This section empirically verifies the effectiveness of Adv-NTK on the CIFAR-10 (Krizhevsky et al., 2009) dataset. We briefly introduce the experiment and leave details in Appendix F. Please also refer to Appendix F.4 for analogous experiments on SVHN (Netzer et al., 2011) dataset.

**Loss & Dataset & adversarial perturbation.** Squared loss $\mathcal{L}(f(x), y) := \frac{1}{2} \|f(x) - y\|_2^2$ is used in all experiments. In every experiment, we randomly draw $12,000$ samples from the trainset for model training and use the whole test set to evaluate the robust generalization ability of the model. Projected gradient descent (PGD; Madry et al. (2018)) is used to perform adversarial perturbations in both training and evaluation. We adopt $\ell_\infty$-perturbation with radius $\rho \in \{4/255, 8/255\}$.

**Baseline methods.** We adopt two existing methods for comparison. They are: (1) **AT**, which aims to enhance the robustness of finite-width DNNs via solving the minimax problem in Eq. (1), and (2) **NTK**, which directly obtains closed-form infinite-width DNNs from Eq. (4) without training.

Table 1: Robust test accuracy (%) of models trained with different methods on CIFAR-10. Every experiment is repeated 3 times. A high robust test accuracy suggests a strong robust generalizability.

| | Depth | Adv. Acc. ($\ell_\infty$; $\rho = 4/255$) (%) | | | Adv. Acc. ($\ell_\infty$; $\rho = 8/255$) (%) | | |
|---|---|---|---|---|---|---|---|
| | | AT | NTK | Adv-NTK (Ours) | AT | NTK | Adv-NTK (Ours) |
| MLP-x + CIFAR-10 (Subset 12K) | 3 | **30.64±0.42** | 9.93±0.19 | 27.35±0.66 | **26.93±0.07** | 2.81±0.27 | 23.45±0.80 |
| | 4 | **30.35±0.09** | 13.67±0.20 | 28.47±0.62 | **26.44±0.39** | 3.61±0.09 | 23.01±0.24 |
| | 5 | 28.70±0.45 | 16.24±0.26 | **29.04±0.38** | 21.05±0.21 | 4.74±0.43 | **21.90±0.60** |
| | 8 | 10.00±0.00 | 22.44±0.27 | **30.56±0.48** | 10.00±0.00 | 8.23±0.15 | **20.91±0.72** |
| | 10 | 10.00±0.00 | 24.43±0.37 | **30.91±0.12** | 10.00±0.00 | 10.04±0.25 | **20.21±0.21** |
| CNN-x + CIFAR-10 (Subset 12K) | 3 | 18.29±0.40 | 5.01±0.54 | **29.31±0.61** | 12.62±1.78 | 1.31±0.03 | **26.79±2.25** |
| | 4 | 19.30±0.26 | 6.23±0.69 | **31.04±0.55** | 10.39±0.20 | 1.68±0.14 | **25.57±0.56** |
| | 5 | 20.10±1.32 | 7.99±0.37 | **30.46±0.59** | 11.12±0.14 | 1.65±0.07 | **23.48±0.48** |
| | 8 | 12.68±4.64 | 13.07±0.26 | **28.26±0.54** | 10.00±0.00 | 2.55±0.18 | **16.14±0.83** |
| | 10 | 10.00±0.00 | 16.02±0.50 | **26.61±0.41** | 10.00±0.00 | 3.50±0.09 | **13.13±0.28** |

Figure 1: The robust test accuracy curves of finite-width MLP-5/CNN-5 along AT on CIFAR-10. The robust test accuracy of infinite width DNNs learned by NTK and Adv-NTK are also plotted.

**Model architectures.** We study two types of multi-layer DNNs, MLPs and CNNs. Although our theory is originally for MLPs, it can be generalized for CNNs. We use "MLP-x" and "CNN-x" to denote an MLP consisting of $x$ fully-connected (FC) layers and CNN consists $x$ convolutional layers and one FC layer, respectively. The architecture depth $x$ is choosen from the set $\{3, 4, 5, 8, 10\}$.

**Model training.** For **Adv-NTK**, we use $10,000$ data to construct the infinite-width DNN defined in Eq. (24) and $2,000$ data as the validation data for model training. For **AT**, we use the overall $12,000$ data to train the model following Eq. (1). For **NTK**, there is no need for model training and we use the overall $12,000$ data to construct the closed-form infinite-width DNN defined in Eq. (4).

**Results.** The robust test accuracy of models trained with different methods on CIFAR-10 is reported in Table 1. We have two observations: **Firstly**, Adv-NTK achieves significantly higher robust test accuracy than NTK in almost every experiment, which suggests that Adv-NTK can improve the robustness of infinite-width DNNs and $\Xi(t)$ indeed captures robustness brought by AT. **Secondly**, in some experiments, Adv-NTK achieves higher performance than AT, which suggests that Adv-NTK has the potential to be used as an empirical tool to study adversarial robustness. **In summary**, these results not only indicate the effectiveness of our algorithm but also justify our theoretical findings.

We further plot the curves of robust test accuracy of finite-width DNNs along AT, as shown in Fig. 1. We have two observations: **Firstly,** in most of the cases, the robust test accuracy will first rapidly increase and then slowly decrease, which illustrates a clear robust overfitting phenomenon. Similar results with larger models and longer AT can also be found in Rice et al. (2020). **Secondly,** although Adv-NTK can achieve comparable or higher performance than the final model obtained by AT, it could not beat the best model during AT. we deduce that it is because the non-linearity of finite-width DNNs in AT can capture additional robustness, which will be left for future studies.

## 7 CONCLUSIONS

This paper presents a novel theoretical analysis of the robust overfitting of DNNs. By extending the NTK theory, we proved that a wide DNN in AT can be strictly approximated by its linearized counterpart, and also calculated the closed-form AT dynamics of the linearized DNN when the squared loss is used. Based on our theory, we suggested analyzing robust overfitting of DNNs with the closed-form AT dynamics of linearized DNNs and revealed a novel ***AT degeneration*** phenomenon that a DNN in long-term AT will gradually degenerate to that obtained without AT. Further, we designed the first AT algorithm for infinite-width DNNs, named ***Adv-NTK***, by directly optimizing the regularization brought by AT. Empirical studies verified the effectiveness of our proposed method.

## ACKNOWLEDGEMENTS

Di Wang and Shaopeng Fu are supported in part by the baseline funding BAS/1/1689-01-01, funding from the CRG grand URF/1/4663-01-01, FCC/1/1976-49-01 from CBRC, and funding from the AI Initiative REI/1/4811-10-01 of King Abdullah University of Science and Technology (KAUST). They are also supported by the funding of the SDAIA-KAUST Center of Excellence in Data Science and Artificial Intelligence (SDAIA-KAUST AI).

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

# A   PRELIMINARIES

## A.1   ADDITIONAL ASSUMPTIONS AND NOTATIONS

To avoid technicalities, we assume that all functions are differentiable throughout this paper. Furthermore, the order of differentiation and integration is assumed to be interchangeable.

## A.2   NOTATIONS

This section presents the full list of notations.

Table 2: List of notations.

| Notations | Descriptions |
|---:|---|
| $\oplus$ | Concatenation. |
| $\otimes$ | Kronecker product. |
| $\mathrm{Diag}(\cdot)$ | A diagonal matrix constructed from a given input. |
| $\mathrm{Vec}(\cdot)$ | Vectorization function. |
| $\partial_x f(x)$ | A $m \times n$ Jacobian matrix of the function, $f : \mathbb{R}^n \to \mathbb{R}^m$. |
| $\lambda_{\max}(\cdot)$ | The maximum eigenvalue of a given matrix. |
| $\xrightarrow{P}$ | Convergence in probability. See Definition A.3 |
| $O_p(\cdot)$ | Big O notation in probability. See Definition A.4. |
| $I_n$ | An $n \times n$ identity matrix. |
| $0_n$ | An $n$-dimensional all-zero vector. |
| $1_n$ | An $n$-dimensional all-one vector. |
| $[n]$ | An integer set $\{1, 2, \cdots, n\}$. |
| $[l : r]$ | An integer set $\{l, l + 1, \cdots, r\}$. |
| $T$ | The overall training time. |
| $S$ | The overall time usage for searching adversarial examples in each training step. |
| $W_t^{(l)}$ | Weight matrix in the $l$-th layer at the training time $t$. |
| $b_t^{(l)}$ | bias vector in the $l$-th layer at the training time $t$. |
| $h_t^{(l)}$ | Pre-activated output function in the $l$-th layer at the training time $t$. |
| $x_t^{(l)}$ | Post-activated output function in the $l$-th layer at the training time $t$. |
| $\hat{\Theta}_{\theta,t}$ | Empirical NTK at the training time $t$. |
| $\hat{\Theta}_{x,t}$ | Empirical ARK at the training time $t$. |
| $\Theta_\theta$ | Converged NTK in the infinite width limit. |
| $\Theta_x$ | Converged ARK in the infinite width limit. |

## A.3   DEFINITIONS

This section collects definitions that are omitted from the main text.

**Definition A.1** (Lipschitz continuity). *A function $f : \mathbb{R} \to \mathbb{R}$ is called $K$-Lipschitz continuous if $|f(x) - f(x')| \leq K \cdot |x - x'|$ holds for any $x$ and $x'$ from the domain of $f$. When $f$ is differentiable, we further have $|\partial_x f(x)| \leq K$ holds for any $x$ from the domain of $f$.*

**Definition A.2** (Lipschitz smoothness). *A differentiable function $f : \mathbb{R} \to \mathbb{R}$ is called $K$-Lipschitz smooth if $|\partial_x f(x) - \partial_x f(x')| \leq K \cdot |x - x'|$ holds for any $x$ and $x'$ from the domain of $f$. When $f$ is twice-differentiable, we further have $|\partial_x^2 f(x)| \leq K$ holds for any $x$ from the domain of $f$.*

**Definition A.3** (Convergence in probability). *For a set of random variables $X_n$ indexed by $n$ and an additional random variable $X$, we say $X_n$ converges in probability to $X$, written by $X_n \xrightarrow{P} X$, if $\lim_{n\to\infty} \mathbb{P}(|X_n - X| > \epsilon) = 0$ for any $\epsilon > 0$.*

**Definition A.4** ($O_p(\cdot)$; Big O notation in probability)**.** *For two sets of random variables $X_n$ and $Y_n$ indexed by $n$, we say $X_n = O_p(Y_n)$ as $n \to \infty$ if for any $\delta > 0$, there exists a finite $\varepsilon > 0$ and a finite $N \in \mathbb{N}^+$ such that $\mathbb{P}(|X_n/Y_n| > \varepsilon) < \delta, \forall n > N$.*

## A.4  TECHNICAL LEMMAS

This section presents several technical lemmas that will be used in our proofs.

We first present a result that enables the efficient calculation of Kronecker products.

**Lemma A.1** (c.f. Lemma 4.2.15 in Horn & Johnson (1991))**.** *Suppose real matrices $A$ and $B$ have singular value decompositions $A = U_1 \Sigma_1 V_1^T$ and $B = U_2 \Sigma_2 V_2^T$. Let $r_1 := \mathrm{Rank}(A)$ and $r_2 := \mathrm{Rank}(B)$. Then, $A \otimes B = (U_1 \otimes U_2) \cdot (\Sigma_1 \otimes \Sigma_2) \cdot (V_1 \otimes V_2)^T$. The nonzero singular values of $A \otimes B$ are the $r_1 r_2$ positive numbers $\{\lambda_i(A)\lambda_j(B) : 1 \le i \le r_1, \ 1 \le j \le r_2\}$, where $\lambda_i(\cdot)$ denotes the $i$-th largest singular value (including multiplicities) of a given matrix.*

**Corollary A.1** ($\ell_2$-Norm of Kronecker Product)**.** *For any real matrices $A$ and $B$, we have that $\|A \otimes B\|_2 = \|A\|_2 \cdot \|B\|_2$.*

*Proof.* According to Lemma A.1, we have $\|A \otimes B\|_2 = \lambda_{\max}(A) \cdot \lambda_{\max}(B) = \|A\|_2 \cdot \|B\|_2$. $\qquad\square$

We then introduce two Grönwall-type inequalities.

**Lemma A.2** (Grönwall's Inequality (Gronwall, 1919; Bellman, 1943))**.** *Let $u(t)$ and $f(t)$ be non-negative continuous functions defined on the interval $[a, b]$ that satisfies*

$$u(t) \le A + \int_a^t u(s)f(s)\mathrm{d}s, \quad \forall t \in [a, b],$$

*then*

$$u(t) \le A \, \exp\left(\int_a^t f(s)\mathrm{d}s\right), \quad \forall t \in [a, b].$$

**Lemma A.3** (Adapted from Lemma 1 in LaSalle (1949))**.** *Suppose that*

1. *$g(x)$ is a non-negative function defined on the interval $[0, a]$,*

2. *$u(x)$ is a function such that $\int_0^x u(t)\mathrm{d}t$ exists for all $x \in [0, a]$.*

3. *$f(x)$ is a positive, non-decreasing continuous function defined on $[0, +\infty)$, and the integral $F(x) := \int_0^x \frac{\mathrm{d}t}{f(t)}$ exists for any $x \in [0, +\infty)$,*

4. *The inequality $g(x) \le \int_0^x u(t)f(g(t))\mathrm{d}t$ holds for all $x \in [0, a]$.*

*Then we have*

$$F(g(x)) \le \int_0^x u(t)\mathrm{d}t, \quad \forall x \in [0, a].$$

Also, we need the following Lemma to bound the norms of Gaussian random matrices.

**Lemma A.4** (c.f. Corollary 5.35, Vershynin (2010))**.** *Let $A$ be an $N \times n$ matrix whose entries are independent standard normal random variables. Then for every $t \ge 0$, with probability at least $1 - 2\exp(-t^2/2)$ one has*

$$\sqrt{N} - \sqrt{n} - t \le \lambda_{\min}(A) \le \lambda_{\max}(A) \le \sqrt{N} + \sqrt{n} + t,$$

*where $\lambda_{\min}(A)$ is the minimal eigenvalue of $A$ and $\lambda_{\max}(A)$ is the maximal eigenvalue of $A$.*

Finally, we prove the following convergence in probability result for centered Gaussian variables.

**Lemma A.5.** *Suppose $Y_n = \max\{|X_{n,1}|, |X_{n,2}|, \cdots, |X_{n,n}|\}$, where $X_{n,i} \sim \mathcal{N}(0, \frac{\sigma_n^2}{n^\nu})$ and $\sigma > 0$ and $\nu > 0$ are constants. Then, we have $Y_n \xrightarrow{P} 0$.*

*Proof.* For any $X_{n,i}$ and $\epsilon > 0$, we have

$$\mathbb{P}(|X_{n,i}| > \epsilon) = 2 \cdot \int_{\epsilon}^{+\infty} \frac{\sqrt{n^\nu}}{\sigma\sqrt{2\pi}} \cdot \exp\left(-\frac{n^\nu x^2}{2\sigma^2}\right) \mathrm{d}x$$

$$\leq 2 \cdot \int_{\epsilon}^{+\infty} \frac{x}{\epsilon} \cdot \frac{\sqrt{n^\nu}}{\sigma\sqrt{2\pi}} \cdot \exp\left(-\frac{n^\nu x^2}{2\sigma^2}\right) \mathrm{d}x$$

$$= \frac{-\sqrt{2}\sigma}{\sqrt{n^\nu}\pi\epsilon} \cdot \left[\exp\left(-\frac{n^\nu x^2}{2\sigma^2}\right)\right]_{\epsilon}^{+\infty}$$

$$= \frac{\sqrt{2}\sigma}{\sqrt{n^\nu}\pi\epsilon} \cdot \exp\left(-\frac{n^\nu \epsilon^2}{2\sigma^2}\right),$$

which means

$$\mathbb{P}(|X_{n,i}| \leq \epsilon) = 1 - \mathbb{P}(|X_{n,i}| > \epsilon) \geq 1 - \frac{\sqrt{2}\sigma}{\sqrt{n^\nu}\pi\epsilon} \cdot \exp\left(-\frac{n^\nu \epsilon^2}{2\sigma^2}\right).$$

Therefore,

$$\mathbb{P}(Y_n \leq \epsilon | \sigma) = \mathbb{P}(|X_{n,i}| \leq \epsilon, \forall i \in [n]) \geq \left(1 - \frac{\sqrt{2}\sigma}{\sqrt{n^\nu}\pi\epsilon} \cdot \exp\left(-\frac{n^\nu \epsilon^2}{2\sigma^2}\right)\right)^n.$$

Since $\lim_{n\to\infty} \frac{\sqrt{2}\sigma}{\sqrt{n^\nu}\pi\epsilon} \cdot \exp\left(-\frac{n^\nu \epsilon^2}{2\sigma^2}\right) = 0$, we thus have

$$\lim_{n\to\infty} \mathbb{P}(Y_n > \epsilon) = 1 - \lim_{n\to\infty} \left(1 - \frac{\sqrt{2}\sigma}{\sqrt{n^\nu}\pi\epsilon} \cdot \exp\left(-\frac{n^\nu \epsilon^2}{2\sigma^2}\right)\right)^n$$

$$= 1 - \lim_{n\to\infty} \left(\frac{1}{e}\right)^{\frac{\sqrt{2}\sigma}{\sqrt{n^\nu}\pi\epsilon} \cdot \exp\left(-\frac{n^\nu \epsilon^2}{2\sigma^2}\right) \cdot n}$$

$$= 1 - \left(\frac{1}{e}\right)^0 = 0,$$

which indicates $Y_n \xrightarrow{P} 0$.

The proof is completed. $\square$

## B  PROOF OF THEOREM 1

This section presents the proof of Theorem 1.

We first introduce the following Lemma B.1.

**Lemma B.1** (Proposition 1 in Jacot et al. (2018)). *As the network widths $n_0, \cdots, n_L \to \infty$ sequentially, the output functions $h_{0,i}^{(l)}$ for $i = 1, \cdots, n_l$ at the l-th layer tend to centered Gaussian processes of covariance $\Sigma^{(l)}$, where $\Sigma^{(l)}$ is defined recursively by:*

$$\Sigma^{(1)}(x, x') = \lim_{n_0 \to \infty} \frac{\sigma_W^2}{n_0} x^T x' + \sigma_b^2,$$

$$\Sigma^{(l+1)}(x, x') = \sigma_W^2 \mathbb{E}_{f \sim \mathcal{GP}(0, \Sigma^{(l)})}[\phi(f(x))\phi(f(x'))] + \sigma_b^2,$$

*where $\mathcal{GP}(0, \Sigma^{(l)})$ denotes a centered Gaussian process with covariance $\Sigma^{(l)}$.*

Then, Theorem 1 is proved as the following Theorem B.1.

**Theorem B.1** (Kernels limits at initialization; Formal version of Theorem 1). *Let $\hat{\Theta}_{t,0}^{(l)}$ and $\hat{\Theta}_{x,0}^{(l)}$ denote the empirical NTK and ARK in the l-th layer. Then, for any $x, x' \in \mathcal{X}$, we have that:*

1. *(Theorem 1 in Jacot et al. (2018)) For any $1 \leq l \leq L+1$,*

$$\lim_{n_{l-1}\to\infty} \cdots \lim_{n_0\to\infty} \hat{\Theta}_{\theta,0}^{(l)}(x,x') = \Theta_\theta^{(l)}(x,x') := \Theta_\theta^{\infty,(l)}(x,x') \cdot I_{n_l},$$

*where $\Theta_\theta^{\infty,(l)} : \mathcal{X} \times \mathcal{X} \to \mathbb{R}$ is a deterministic kernel function that can be defined recursively as follows,*

$$\Theta_\theta^{\infty,(1)}(x,x') = \lim_{n_0\to\infty} \frac{1}{n_0} x^T x' + 1,$$

$$\Theta_\theta^{\infty,(l+1)}(x,x') = \sigma_W^2 \cdot \Theta_\theta^{\infty,(l)}(x,x') \cdot \mathbb{E}_{f\sim\mathcal{GP}(0,\Sigma^{(l)})}[\phi'(f(x))\phi'(f(x'))]$$
$$+ \mathbb{E}_{f\sim\mathcal{GP}(0,\Sigma^{(l)})}[\phi(f(x))\phi(f(x'))] + 1.$$

2. *For any $1 \leq l \leq L+1$,*

$$\lim_{n_{l-1}\to\infty} \cdots \lim_{n_0\to\infty} \hat{\Theta}_{x,0}^{(l)}(x,x') = \Theta_x^{(l)}(x,x') := \Theta_x^{\infty,(l)}(x,x') \cdot I_{n_l},$$

*where $\Theta_x^{\infty,(l)} : \mathcal{X} \times \mathcal{X} \to \mathbb{R}$ is a deterministic kernel function that can be defined recursively as follows,*

$$\Theta_x^{\infty,(1)}(x,x') = \sigma_W^2,$$

$$\Theta_x^{\infty,(l+1)}(x,x') = \sigma_W^2 \cdot \Theta_x^{\infty,(l)}(x,x') \cdot \mathbb{E}_{f\sim\mathcal{GP}(0,\Sigma^{(l)})}[\phi'(f(x))\phi'(f(x'))].$$

*Proof.* The first proposition, *i.e.*, the NTK limit at initialization, has been proved by Jacot et al. (2018). Thus the remaining task is to prove the ARK limit at initialization, which is done through mathematical induction.

Specifically, for the base case where $l = 1$, for the $i$-th row and $i'$-th column entry of the ARK matrix $\hat{\Theta}_{x,0}^{(1)}(x,x')$, we have that

$$\hat{\Theta}_{x,0}^{(1)}(x,x')_{i,i'} = \frac{1}{n_0} \sum_{j=1}^{n_0} W_{i,j,0}^{(1)} W_{i',j,0}^{(1)}.$$

Since each $W_{i,j,0}^{(1)}$ is drawn from the Gaussian $\mathcal{N}(0, \sigma_W^2)$, we have

$$\lim_{n_0\to\infty} \hat{\Theta}_{x,0}^{(1)}(x,x')_{i,i'} = 1_{[i=i']} \cdot \sigma_W^2 = 1_{[i=i']} \cdot \Theta_x^{\infty,(1)}(x,x')$$

which indicates

$$\lim_{n_0\to\infty} \hat{\Theta}_{x,0}^{(1)}(x,x') = \sigma_W^2 \cdot I_{n_1} = \Theta_x^{\infty,(1)}(x,x') \cdot I_{n_1}.$$

For the induction step, suppose the lemma already holds for the $l$-th layer and we aim to prove it also holds for the $(l+1)$-th layer. For the ARK matrix $\hat{\Theta}_{x,0}^{(l+1)}(x,x')$, we have that

$$\lim_{n_l\to\infty} \cdots \lim_{n_0\to\infty} \hat{\Theta}_{x,0}^{(l+1)}(x,x')_{i,i'}$$

$$= \lim_{n_l\to\infty} \frac{1}{n_l} \sum_{1\leq j,j'\leq n_l} \left( \lim_{n_{l-1}\to\infty} \cdots \lim_{n_0\to\infty} \hat{\Theta}_{x,0}^{(l)}(x,x')_{j,j'} \right) \cdot \partial_{h^{(l)}(x)_j} h_0^{(l+1)}(x)_i \cdot \partial_{h^{(l)}(x)_{j'}} h_0^{(l+1)}(x')_{i'}$$

$$= \lim_{n_l\to\infty} \frac{1}{n_l} \sum_{1\leq j,j'\leq n_l} \underbrace{1_{[j=j']} \cdot \Theta_x^{\infty,(l)}(x,x')}_{\text{Induction hypothesis}} \cdot W_{i,j,0}^{(l)} \cdot \phi'(h_0^{(l)}(x)_j) \cdot W_{i',j',0}^{(l)} \cdot \phi'(h_0^{(l)}(x')_{j'})$$

$$= \lim_{n_l\to\infty} \frac{1}{n_l} \sum_{1\leq j\leq n_l} \Theta_x^{\infty,(l)}(x,x') \cdot W_{i,j,0}^{(l)} \cdot \phi'(h_0^{(l)}(x)_j) \cdot W_{i',j,0}^{(l)} \cdot \phi'(h_0^{(l)}(x')_j)$$

By Lemma B.1, $h_0^{(l)}(\cdot)_k$ converges to the centered Gaussian process $\mathcal{GP}(0, \Sigma^{(l)})$, which further indicates

$$
\begin{aligned}
&\lim_{n_l \to \infty} \cdots \lim_{n_0 \to \infty} \hat{\Theta}_{x,0}^{(l+1)}(x, x')_{i,i'} \\
&= \lim_{n_l \to \infty} \frac{1}{n_l} \sum_{1 \le j \le n_l} \Theta_x^{\infty,(l)}(x, x') \cdot \phi'(h_0^{(l)}(x)_j) \cdot W_{0,i,j}^{(l+1)} \cdot W_{0,i',j}^{(l+1)} \cdot \phi'(h_0^{(l)}(x')_j) \\
&= 1_{[i=i']} \cdot \sigma_W^2 \cdot \Theta_x^{\infty,(l)}(x) \cdot \mathbb{E}_{f \sim \mathcal{GP}(0,\Sigma^{(l)})}[\phi'(f(x))\phi'(f(x'))] \\
&= 1_{[i=i']} \cdot \Theta_x^{\infty,(l+1)}(x, x').
\end{aligned}
$$

As a result,

$$
\lim_{n_l \to \infty} \cdots \lim_{n_0 \to \infty} \hat{\Theta}_{x,0}^{(l+1)}(x, x') = \Theta_x^{\infty,(l+1)}(x, x') \cdot I_{n_l},
$$

which justifies the induction step.

The proof is completed. $\qquad\square$

## C  PROOF OF THEOREM 2

This section presents the proof of Theorem 2.

### C.1  PROOF SKELETON

The Proof idea of Theorem 2 is inspired by that in Jacot et al. (2018). Specifically, we will first extend the result of Theorem B.1 and show that the empirical NTK $\hat{\Theta}_{\theta,t}$ and empirical ARK $\hat{\Theta}_{x,t}$ during AT also converge to the same deterministic kernels as that in Theorem B.1. These results are stated as Theorems C.1, C.2 and presented as follows:

**Theorem C.1.** *Suppose Assumptions 1, 2, 3 hold. Then, if there exists $\tilde{n} \in \mathbb{N}^+$ such that $\min\{n_0, \cdots, n_L\} \ge \tilde{n}$, we have as $\tilde{n} \to \infty$,*

$$
\lim_{n_L \to \infty} \cdots \lim_{n_0 \to \infty} \sup_{t \in [0,T], s \in [0,S]} \|\hat{\Theta}_{\theta,t}(\mathbf{x}_{t,s}, \mathbf{x}_{t,s}) - \Theta_\theta(\mathbf{x}, \mathbf{x})\|_2 \xrightarrow{P} 0,
$$

$$
\lim_{n_L \to \infty} \cdots \lim_{n_0 \to \infty} \sup_{t \in [0,T], s \in [0,S]} \|\hat{\Theta}_{x,t}(\mathbf{x}_{t,s}, \mathbf{x}_{t,s}) - \Theta_x(\mathbf{x}, \mathbf{x})\|_2 \xrightarrow{P} 0,
$$

*where*

$$
\begin{aligned}
\Theta_\theta(\mathbf{x}, \mathbf{x}) &:= \Theta_\theta^\infty(\mathbf{x}, \mathbf{x}) \otimes I_{n_{L+1}}, \\
\Theta_x(\mathbf{x}, \mathbf{x}) &:= \mathrm{Diag}(\Theta_x^\infty(x_1, x_1), \cdots, \Theta_x^\infty(x_M, x_M)) \otimes I_{n_{L+1}}.
\end{aligned}
$$

**Theorem C.2.** *Suppose Assumptions 1, 2, 3 hold. Then, if there exists $\tilde{n} \in \mathbb{N}^+$ such that $\{n_0, \cdots, n_L\} \ge \tilde{n}$, we have for any $x \in \mathcal{X}$, as $\tilde{n} \to \infty$,*

$$
\lim_{n_L \to \infty} \cdots \lim_{n_0 \to \infty} \sup_{t \in [0,T], s \in [0,S]} \|\hat{\Theta}_{\theta,t}(x, \mathbf{x}_{t,s}) - \Theta_\theta(x, \mathbf{x})\|_2 \xrightarrow{P} 0,
$$

$$
\lim_{n_L \to \infty} \cdots \lim_{n_0 \to \infty} \sup_{t \in [0,T], s \in [0,S]} \|\hat{\Theta}_{x,t}(x, \mathbf{x}_{t,s}) - \Theta_x(x, \mathbf{x})\|_2 \xrightarrow{P} 0,
$$

*where*

$$
\begin{aligned}
\Theta_\theta(x, \mathbf{x}) &:= \Theta_\theta^\infty(x, \mathbf{x}) \otimes I_{n_{L+1}} \\
\Theta_x(x, \mathbf{x}) &:= \mathrm{Diag}(\Theta_x^\infty(x, x_1), \cdots, \Theta_x^\infty(x, x_M)) \otimes I_{n_{L+1}}.
\end{aligned}
$$

Based on the kernels convergences during AT, we then prove the final Theorem 2 which show that the adversarially trained DNN $f_t$ is equivalent to the linearized DNN $f_t^{\mathrm{lin}}$. Compared with Jacot et al. (2018), the main technical challenge in our proof arises from bounding terms that involve adversarial examples such as $\mathbf{x}_{t,s}, \partial_t \mathbf{x}_{t,s}, \partial_t h_t^{(l)}(\mathbf{x}_{t,s})$.

The rest of this section is organized as follows:

1. In Appendix C.2, we prove that as the widths of $f_t$ become infinite, its parameters and outputs will converge to those at the initial of AT by properly rescaling.

2. In Appendix C.3, we prove Theorems C.1 and C.2 that in the infinite-widths limit, the NTK and ARK in AT will converge to deterministic kernel functions. The proofs are based on the results in Appendix C.2.

3. Finally, in Appendix C.4 we prove Theorem 2 that in the infinite-widths limit, $f_t$ is equivalent to $f_t^{\text{lin}}$. The proof is based on Theorems C.1 and C.2.

## C.2 Convergences of Parameters and Outputs Under Rescaling

The goal of this section is to prove Lemma C.5, which indicates that by properly rescaling, for the wide DNN $f_t$, its parameter $W_t^{(l)}$, pre-activation $x_t^{(l)}(\mathbf{x}_{t,S})$, and post-activation $h_t^{(l)}(\mathbf{x}_{t,S})$ in each layer can converge to those at the initialization of AT.

Firstly, we let

$$\text{Poly}_t := \text{Poly}\left(\frac{1}{\sqrt{n_{l-1}}}\|W_t^{(l)}\|_2, \frac{1}{\sqrt{n_l}}\|h_t^{(l)}(\mathbf{x}_{t,S})\|_2, \frac{1}{\sqrt{n_l}}\|x_t^{(l)}(\mathbf{x}_{t,S})\|_2\right)$$

denote any deterministic polynomial with **finite degree** and **finite positive constant coefficients**, and depends only on terms $\frac{1}{\sqrt{n_{l-1}}}\|W_t^{(l)}\|_2$ (where $l \in [1 : L+1]$), $\frac{1}{\sqrt{n_l}}\|h_t^{(l)}(\mathbf{x}_{t,S})\|_2$ (where $l \in [1 : L]$), and $\frac{1}{\sqrt{n_l}}\|x_t^{(l)}(\mathbf{x}_{t,S})\|_2$ (where ($l \in [0 : L]$).

From the definition of $\text{Poly}_t$, the following properties worth highlighting:

- The sum of any two (different) such type polynomials is also such type polynomial:
$$\text{Poly}_t + \text{Poly}_t = \text{Poly}_t.$$

- The product of any two (different) such type polynomials is also such type polynomial:
$$\text{Poly}_t \cdot \text{Poly}_t = \text{Poly}_t.$$

With the definition of $\text{Poly}_t$, we then bound $\|\partial_t W_t^{(l)}\|_2$, $\|\partial_t h_t^{(l)}(\mathbf{x}_{t,S})\|_2$, and $\|\partial_t x_t^{(l)}(\mathbf{x}_{t,S})\|_2$, as shown in the following Lemmas C.1, C.2, C.3, C.4.

**Lemma C.1.** *Suppose Assumption 1 holds. Then for every $1 \le l \le L+1$ and $t \in [0, T]$, we have*

$$\|\partial_t W_t^{(l)}\|_2, \quad \|\partial_t W_t^{(l)}\|_F, \quad \|\partial_t b_t^{(l)}\|_2 \le \text{Poly}_t \cdot \|\partial_{f(\mathbf{x})}\mathcal{L}(f_t(\mathbf{x}_{t,S}, \vec{y})\|_2.$$

*Proof.* By the fact that $\|\cdot\|_2 \le \|\cdot\|_F$, we have

$$\|\partial_t W_t^{(l)}\|_2 \le \|\partial_t W_t^{(l)}\|_F = \|-\partial_{\text{Vec}(W^{(l)})}^T h_t^{(L+1)}(\mathbf{x}_{t,S}) \cdot \partial_{f(\mathbf{x})}^T \mathcal{L}(f_t(\mathbf{x}_{t,S}), \mathbf{y})\|_2$$

$$\le \|\partial_{\text{Vec}(W^{(l)})} h_t^{(L+1)}(\mathbf{x}_{t,S})\|_2 \cdot \|\partial_{f(\mathbf{x})}\mathcal{L}(f_t(\mathbf{x}_{t,S}), \mathbf{y})\|_2.$$

By applying Lipschitz activation Assumption 1 and Corollary A.1,

$$\|\partial_{\text{Vec}(W^{(l)})} h_t^{(L+1)}(\mathbf{x}_{t,S})\|_2$$

$$\le \left(\prod_{l'=l}^{L} \|\partial_{x^{(l')}(\mathbf{x})} h_t^{(l'+1)}(\mathbf{x}_{t,S})\|_2 \cdot \|\partial_{h^{(l')}(\mathbf{x})} x_t^{(l')}(\mathbf{x}_{t,S})\|_2\right) \cdot \|\partial_{\text{Vec}(W^{(l)})} h_t^{(l)}(\mathbf{x}_{t,S})\|_2$$

$$\le \left(\prod_{l'=l}^{L} \left\|I_M \otimes \frac{1}{\sqrt{n_{l'}}} W_t^{(l'+1)}\right\|_2 \cdot K\right) \cdot \sqrt{\frac{1}{n_l}\left\|\left(x^{(l)}(x_{i,t,S})^T x^{(l)}(x_{j,t,S})\right) \otimes I_{n_l}\right\|_2}$$

$$\le K^{L-l+1} \cdot \prod_{l'=l}^{L} \frac{1}{\sqrt{n_{l'}}} \|W_t^{(l'+1)}\|_2 \cdot \frac{1}{\sqrt{n_l}} \left\|\left(x_t^{(l)}(x_{1,t,S}), \cdots, x_t^{(l)}(x_{M,t,S})\right)\right\|_2$$

$$\le \text{Poly}_t \cdot \frac{1}{\sqrt{n_l}} \left\|\left(x_t^{(l)}(x_{1,t,S}), \cdots, x_t^{(l)}(x_{M,t,S})\right)\right\|_F$$

$$= \text{Poly}_t \cdot \frac{1}{\sqrt{n_l}} \|x_t^{(l)}(\mathbf{x}_{t,S})\|_2 = \text{Poly}_t,$$

where $\left(x^{(l)}(x_{i,t,S})^T x^{(l)}(x_{j,t,S})\right)$ is a $M \times M$ matrix such that its $i$-th row and $j$-th column entry is $x^{(l)}(x_{i,t,S})^T x^{(l)}(x_{j,t,S})$. Combining the above results, we therefore have

$$\|\partial_t W_t^{(l)}\|_2 \leq \mathrm{Poly}_t \cdot \|\partial_{f(\mathbf{x})}\mathcal{L}(f_t(\mathbf{x}_{t,S}, \mathbf{y})\|_2.$$

For $\|\partial_t b_t\|_2$, we similarly have that

$$\begin{aligned}
\|\partial_t b_t^{(l)}\|_2 &= \|\partial_{b^{(l)}}^T h_t^{(L+1)}(\mathbf{x}_{t,S}) \cdot \partial_{f(\mathbf{x})}^T \mathcal{L}(f_t(\mathbf{x}_{t,S}), \mathbf{y})\|_2 \\
&\leq \|\partial_{b^{(l)}} h_t^{(l)}(\mathbf{x}_{t,S})\|_2 \cdot \|\partial_{h^{(l)}(\mathbf{x})} h_t^{(L+1)}(\mathbf{x}_{t,S})\|_2 \cdot \|\partial_{f(\mathbf{x})}\mathcal{L}(f_t(\mathbf{x}_{t,S}), \mathbf{y})\|_2 \\
&= \|1_M \otimes I_{n_l}\|_2 \cdot \|\partial_{h^{(l)}(\mathbf{x})} h_t^{(L+1)}(\mathbf{x}_{t,S})\|_2 \cdot \|\partial_{f(\mathbf{x})}\mathcal{L}(f_t(\mathbf{x}_{t,S}), \mathbf{y})\|_2 \\
&\leq \sqrt{M} \cdot \mathrm{Poly}_t \cdot \|\partial_{f(\mathbf{x})}\mathcal{L}(f_t(\mathbf{x}_{t,S}), \mathbf{y})\|_2 \\
&= \mathrm{Poly}_t \cdot \|\partial_{f(\mathbf{x})}\mathcal{L}(f_t(\mathbf{x}_{t,S}), \mathbf{y})\|_2.
\end{aligned}$$

The proof is completed. $\qquad\qquad\square$

**Lemma C.2.** *Suppose Assumptions 1, 2, 3 hold. Then for any $t \in [0, T]$, as $\min_{l' \in [0:L]}\{n_{l'}\} \to \infty$, we have that*

$$\sup_{s_1, s_2 \in [0,S]} \|x_t^{(l)}(\mathbf{x}_{t,s_1}) - x_t^{(l)}(\mathbf{x}_{t,s_2})\|_2 \leq O_p(1) \cdot \mathrm{Poly}_t,$$

*where $0 \leq l \leq L$.*

*Proof.* By Assumption 3, $\boldsymbol{\eta}(t)$ is continuous on the closed interval $[0, T]$, which indicates there exists a constant $C$ such that $\sup_{t \in [0,T]} \|\boldsymbol{\eta}(t)\|_2 \leq C$.

Then, when $l = 0$, according to definition, we have

$$\begin{aligned}
\sup_{s_1, s_2 \in [0,S]} \|\mathbf{x}_{t,s_1} - \mathbf{x}_{t,s_2}\|_2 &= \sup_{s_1, s_2 \in [0,S]} \left\|\int_{s_2}^{s_1} \partial_\tau x_t^{(l)}(\mathbf{x}_{t,\tau}) \mathrm{d}\tau\right\|_2 \leq \int_0^S \|\partial_\tau x_t^{(l)}(\mathbf{x}_{t,\tau})\|_2 \mathrm{d}\tau \\
&\leq \|\boldsymbol{\eta}(t)\|_2 \cdot \int_0^S \|\partial_\mathbf{x} h_t^{(L+1)}(\mathbf{x}_{t,\tau})\|_2 \cdot \|\partial_{f(\mathbf{x})}\mathcal{L}(f_t(\mathbf{x}_{t,\tau}), \mathbf{y})\|_2 \mathrm{d}\tau \\
&\leq C \cdot \int_0^S \|\partial_\mathbf{x} h_t^{(L+1)}(\mathbf{x}_{t,\tau})\|_2 \cdot \|\partial_{f(\mathbf{x})}\mathcal{L}(f_t(\mathbf{x}_{t,\tau}), \mathbf{y})\|_2 \mathrm{d}\tau.
\end{aligned}$$

By applying Lipschitz activation Assumption 1,

$$\begin{aligned}
&\sup_{\tau \in [0,S]} \|\partial_\mathbf{x} h_t^{(L+1)}(\mathbf{x}_{t,\tau})\|_2 \\
&\leq \sup_{\tau \in [0,S]} \left\{\left(\prod_{l'=1}^L \|\partial_{x^{(l')}(\mathbf{x})} h_t^{(l'+1)}(\mathbf{x}_{t,\tau})\|_2 \cdot \|\partial_{h^{(l')}(\mathbf{x})} x_t^{(l')}(\mathbf{x}_{t,\tau})\|_2\right) \cdot \|\partial_\mathbf{x} h_t^{(1)}(\mathbf{x}_{t,\tau})\|_2\right\} \\
&\leq \left(\prod_{l'=1}^L \frac{1}{\sqrt{n_{l'}}}\|W_t^{(l'+1)}\|_2 \cdot K\right) \cdot \frac{1}{\sqrt{n_0}}\|W_t^{(1)}\|_2 = \mathrm{Poly}_t.
\end{aligned} \qquad (\mathrm{C.1})$$

Combining with Assumption 2,

$$\begin{aligned}
\sup_{s_1, s_2 \in [0,S]} \|\mathbf{x}_{t,s_1} - \mathbf{x}_{t,s_2}\|_2 &\leq C \cdot \mathrm{Poly}_t \cdot \sup_{t \in [0,T]} \int_0^S \|\partial_{f(\mathbf{x})}\mathcal{L}(f_t(\mathbf{x}_{t,\tau}), \mathbf{y})\|_2 \mathrm{d}\tau \\
&= \mathrm{Poly}_t \cdot O_p(1).
\end{aligned} \qquad (\mathrm{C.2})$$

On the other hand, for any $1 \le l \le L$, by again using Assumption 1,

$$\sup_{s_1,s_2\in[0,S]} \|x_t^{(l)}(\mathbf{x}_{t,s_1}) - x_t^{(l)}(\mathbf{x}_{t,s_2})\|_2 = \sup_{s_1,s_2\in[0,S]} \|\phi(h_t^{(l)}(\mathbf{x}_{t,s_1})) - \phi(h_t^{(l)}(\mathbf{x}_{t,s_2}))\|_2$$

$$\le K \cdot \sup_{s_1,s_2\in[0,S]} \|h_t^{(l)}(\mathbf{x}_{t,s_1}) - h_t^{(l)}(\mathbf{x}_{t,s_2})\|_2$$

$$\le K \cdot \frac{1}{\sqrt{n_{l-1}}} \|W_t^{(l)}\|_2 \cdot \sup_{s_1,s_2\in[0,S]} \|x_t^{(l-1)}(\mathbf{x}_{t,s_1}) - x_t^{(l-1)}(\mathbf{x}_{t,s_2})\|_2$$

$$= \text{Poly}_t \cdot \sup_{s_1,s_2\in[0,S]} \|x_t^{(l-1)}(\mathbf{x}_{t,s_1}) - x_t^{(l-1)}(\mathbf{x}_{t,s_2})\|_2$$

$$\le \cdots \le \text{Poly}_t \cdot \sup_{s_1,s_2\in[0,S]} \|x_t^{(0)}(\mathbf{x}_{t,s_1}) - x_t^{(0)}(\mathbf{x}_{t,s_2})\|_2$$

$$\le \text{Poly}_t \cdot \text{Poly}_t \cdot O_p(1) = O_p(1) \cdot \text{Poly}_t.$$

The proof is completed. $\qquad\qquad\square$

**Lemma C.3.** *Suppose Assumptions 1, 2, 3 hold. Then, as $\min_{l\in[0:L]}\{n_l\} \to \infty$, for any $t \in [0, T]$, we uniformly have that*

$$\sup_{s\in[0,S]} \|\partial_t \vec{x}_{t,s}\|_2 \le \exp(O_p(1) \cdot \text{Poly}_t) \cdot O_p(1) \cdot \text{Poly}_t \cdot (\|\partial_{f(\mathbf{x})}\mathcal{L}(f_t(\mathbf{x}_{t,S}), \mathbf{y})\|_2 + 1).$$

*Proof.* By Assumption 3, both $\boldsymbol{\eta}(t)$ and $\partial_t\boldsymbol{\eta}(t)$ are continuous on the closed interval $[0, T]$, which indicates there exists a constant $C$ such that $\sup_{t\in[0,T]}\{\|\boldsymbol{\eta}(t)\|_2, \|\partial_t\boldsymbol{\eta}(t)\|_2\} \le C$.

Then, by assuming differentiation and integration to be interchangeable, for any $s \in [0, S]$, we have

$$\|\partial_t\mathbf{x}_{t,s}\|_2 = \left\| \partial_t \int_0^s \partial_{\mathbf{x}}^T h_t^{(L+1)}(\mathbf{x}_{t,\tau}) \cdot \boldsymbol{\eta}(t) \cdot \partial_{f(\mathbf{x})}^T \mathcal{L}(f_t(\mathbf{x}_{t,\tau}), \mathbf{y})\mathrm{d}\tau \right\|_2$$

$$\le \int_0^s \left\| \partial_t \left( \partial_{\mathbf{x}}^T h_t^{(L+1)}(\mathbf{x}_{t,\tau}) \cdot \boldsymbol{\eta}(t) \cdot \partial_{f(\mathbf{x})}^T \mathcal{L}(f_t(\mathbf{x}_{t,\tau}), \mathbf{y}) \right) \right\|_2 \mathrm{d}\tau$$

$$\le \underbrace{\sup_{\tau\in[0,S]} \left\| \sum_i \partial_{[\theta]_i}\partial_{\mathbf{x}} h_t^{(L+1)}(\mathbf{x}_{t,\tau}) \cdot \partial_t[\theta_t]_i \right\|_2 \cdot C \cdot \int_0^S \|\partial_{f(\mathbf{x})}\mathcal{L}(f_t(\mathbf{x}_{t,\tau}), \mathbf{y})\|_2\mathrm{d}\tau}_{\mathrm{I}_t}$$

$$+ C \cdot \int_0^s \underbrace{\left\| \sum_i \partial_{[\mathbf{x}_{t,\tau}]_i}\partial_{\mathbf{x}} h_t^{(L+1)}(\mathbf{x}_{t,\tau}) \cdot \partial_t[\mathbf{x}_{t,\tau}]_i \right\|_2}_{\mathrm{II}_{t,\tau}} \cdot \|\partial_{f(\mathbf{x})}\mathcal{L}(f_t(\mathbf{x}_{t,\tau}), \mathbf{y})\|_2\mathrm{d}\tau$$

$$+ \underbrace{\sup_{\tau\in[0,S]} \|\partial_{\mathbf{x}} h_t^{(L+1)}(\mathbf{x}_{t,\tau})\|_2}_{\mathrm{III}_t,\ \text{bounded by Eq. (C.1)}} \cdot C \cdot \int_0^S \|\partial_{f(\mathbf{x})}\mathcal{L}(f_t(\mathbf{x}_{t,\tau}), \mathbf{y})\|_2\mathrm{d}\tau$$

$$+ \underbrace{\sup_{\tau\in[0,S]} \|\partial_{\mathbf{x}} h_t^{(L+1)}(\mathbf{x}_{t,\tau})\|_2}_{\mathrm{III}_t,\ \text{bounded by Eq. (C.1)}} \cdot C \cdot \int_0^S \|\partial_t\partial_{f(\mathbf{x})}\mathcal{L}(f_t(\mathbf{x}_{t,\tau}), \mathbf{y})\|_2\mathrm{d}\tau, \qquad (\text{C.3})$$

where $[\cdot]_i$ denotes the $i$-th entry of a given vector. In the above Eq. (C.3), the term $\mathrm{III}_t$ can be bounded by Eq. (C.1) from the proof of Lemma C.2. Thereby, the remaining task is to first bound terms $\mathrm{I}_t$ and $\mathrm{II}_{t,\tau}$ respectively, and then bound the overall $\sup_{s\in[0,S]} \|\partial_t\mathbf{x}_{t,s}\|_2$.

**Stage 1:** Bounding term $\mathrm{I}_t$ in Eq. (C.3).

By Assumption 1, we have

$$
\begin{aligned}
\mathrm{I}_t &= \sup_{\tau \in [0,S]} \left\| \sum_i \partial_{[\theta]_i} \partial_{\mathbf{x}} h_t^{(L+1)}(\mathbf{x}_{t,\tau}) \cdot \partial_t [\theta_t]_i \right\|_2 \\
&= \sup_{\tau \in [0,S]} \left\| \sum_{l=1}^{L+1} \partial_{\mathrm{Vec}(W^{(l)}),b^{(l)}} \left( \prod_{l'=L}^{1} \partial_{h^{(l')}(\mathbf{x})} h_t^{(l'+1)}(\mathbf{x}_{t,\tau}) \cdot \partial_{\mathbf{x}} h_t^{(1)}(\mathbf{x}_{t,\tau}) \right) \cdot \partial_t(\mathrm{Vec}(W_t^{(l)}), b_t^{(l)}) \right\|_2 \\
&\leq \sup_{\tau \in [0,S]} \left\{ \prod_{l'=L}^{1} \|\partial_{h^{(l')}(\mathbf{x})} h_t^{(l'+1)}(\mathbf{x}_{t,\tau})\|_2 \cdot \|\partial_{\mathrm{Vec}(W^{(1)}),b^{(1)}} \partial_{\mathbf{x}} h_t^{(1)}(\mathbf{x}_{t,\tau}) \cdot \partial_t(\mathrm{Vec}(W_t^{(1)}), b_t^{(1)})\|_2 \right\} \\
&\quad + \mathrm{Poly}_t \cdot \sup_{\tau \in [0,S]} \left\{ \sum_{l \leq l'+1} \prod_{l'' \neq l'} \|\partial_{h^{(l'')}(\mathbf{x})} h_t^{(l''+1)}(\mathbf{x}_{t,\tau})\|_2 \cdot \|\partial_{\mathrm{Vec}(W^{(l)}),b^{(l)}} \partial_{h^{(l')}(\mathbf{x})} h_t^{(l'+1)}(\mathbf{x}_{t,\tau}) \cdot \partial_t(\mathrm{Vec}(W_t^{(l)}), b_t^{(l)})\|_2 \right\} \\
&\leq \mathrm{Poly}_t \cdot \sum_{l=1}^{L+1} \frac{1}{\sqrt{n_{l-1}}} \underbrace{\|\partial_{\mathrm{Vec}(W^{(l)})} W_t^{(l)} \cdot \partial_t \mathrm{Vec}(W_t^{(l)})\|_2}_{\mathrm{I}_{t,l}^{(1)}} \\
&\quad + \mathrm{Poly}_t \cdot \sum_{l=1}^{L} \sum_{l'=l}^{L} \underbrace{\sup_{\tau \in [0,S]} \|\partial_{\mathrm{Vec}(W^{(l)}),b^{(l)}} \mathrm{Diag}(\phi'(h_t^{(l')}(\mathbf{x}_{t,\tau}))) \cdot \partial_t(\mathrm{Vec}(W_t^{(l)}), b_t^{(l)})\|_2}_{\mathrm{I}_{t,l,l'}^{(2)}}. \quad (\text{C.4})
\end{aligned}
$$

For the term $\mathrm{I}_{t,l}^{(1)}$, according to Lemma C.1, for $1 \leq l \leq L+1$,

$$
\mathrm{I}_{t,l}^{(1)} = \|W_t^{(l)}\|_2 \leq \mathrm{Poly}_t \cdot \|\partial_{f(\mathbf{x})} \mathcal{L}(f_t(\mathbf{x}_{t,S}), \mathbf{y})\|_2. \quad (\text{C.5})
$$

Besides, the term $\mathrm{I}_{t,l,l'}^{(2)}$ can be expanded as below,

$$
\begin{aligned}
\mathrm{I}_{t,l,l'}^{(2)} &= \sup_{\tau \in [0,S]} \|\partial_{\mathrm{Vec}(W^{(l)}),b^{(l)}} \mathrm{Diag}(\phi'(h_t^{(l')}(\mathbf{x}_{t,\tau}))) \cdot \partial_t(\mathrm{Vec}(W_t^{(l)}), b_t^{(l)})\|_2 \\
&\leq \sup_{\tau \in [0,S]} \|\partial_{\mathrm{Vec}(W^{(l)}),b^{(l)}} \phi'(h_t^{(l')}(\mathbf{x}_{t,\tau})) \cdot \partial_t(\mathrm{Vec}(W_t^{(l)}), b_t^{(l)})\|_2 \\
&\leq \sup_{\tau \in [0,S]} \|\partial_{h^{(l)}(\mathbf{x})} \phi'(h_t^{(l')}(\mathbf{x}_{t,\tau}))\|_2 \cdot \|\partial_{\mathrm{Vec}(W^{(l)}),b^{(l)}} h_t^{(l)}(\mathbf{x}_{t,\tau}) \cdot \partial_t(\mathrm{Vec}(W_t^{(l)}), b_t^{(l)})\|_2 \\
&\leq \sup_{\tau \in [0,S]} \|\phi''(h_t^{(l')}(\mathbf{x}_{t,\tau}))\|_\infty \cdot \|\partial_{h^{(l)}(\mathbf{x})} h_t^{(l')}(\mathbf{x}_{t,\tau})\|_2 \cdot \left( \|\partial_{\mathrm{Vec}(W^{(l)})} h_t^{(l)}(\mathbf{x}_{t,\tau}) \cdot \partial_t \mathrm{Vec}(W_t^{(l)})\|_2 + \|\partial_{b^{(l)}} h_t^{(l)}(\mathbf{x}_{t,\tau}) \cdot \partial_t b_t^{(l)}\|_2 \right) \\
&\leq K \cdot \mathrm{Poly}_t \cdot \sup_{\tau \in [0,S]} \left( \sqrt{\frac{1}{n_{l-1}} \|(x_t^{(l-1)}(x_{i,t,\tau})^T \cdot x_t^{(l-1)}(x_{j,t,\tau})) \otimes I_{n_l}\|_2} \cdot \|\partial_t \mathrm{Vec}(W_t^{(l)})\|_2 + \|1_M \otimes I_{n_l}\| \cdot \|\partial_t b_t^{(l)}\|_2 \right) \\
&\leq \mathrm{Poly}_t \cdot \sup_{\tau \in [0,S]} \left( \frac{1}{\sqrt{n_{l-1}}} \|x_t^{(l-1)}(\mathbf{x}_{t,\tau})\|_2 \cdot \underbrace{\mathrm{Poly}_t \cdot \|\partial_{f(\mathbf{x})} \mathcal{L}(f_t(\mathbf{x}_{t,S}), \mathbf{y})\|_2}_{\text{Lemma C.1}} + \sqrt{M} \cdot \underbrace{\mathrm{Poly}_t \cdot \|\partial_{f(\mathbf{x})} \mathcal{L}(f_t(\mathbf{x}_{t,S}), \mathbf{y})\|_2}_{\text{Lemma C.1}} \right) \\
&\leq \mathrm{Poly}_t \cdot \|\partial_{f(\mathbf{x})} \mathcal{L}(f_t(\mathbf{x}_{t,S}), \vec{y})\|_2 \cdot \sup_{\tau \in [0,S]} \left( \frac{1}{\sqrt{n_{l-1}}} \|x_t^{(l-1)}(\mathbf{x}_{t,\tau})\|_2 + 1 \right), \quad (\text{C.6})
\end{aligned}
$$

where $(x_t^{(l-1)}(x_{i,t,\tau})^T \cdot x_t^{(l-1)}(x_{j,t,\tau}))$ denotes a $M \times M$ matrix that its $i$-th row and $j$-th column entry is $x_t^{(l-1)}(x_{i,t,\tau})^T \cdot x_t^{(l-1)}(x_{j,t,\tau})$. By Lemma C.2, we further have

$$
\sup_{\tau \in [0,S]} \left( \frac{1}{\sqrt{n_{l-1}}} \|x_t^{(l-1)}(\mathbf{x}_{t,\tau})\|_2 + 1 \right)
$$

$$
\leq \sup_{\tau \in [0,S]} \left( \frac{1}{\sqrt{n_{l-1}}} \left( \|x_t^{(l-1)}(\mathbf{x}_{t,S})\|_2 + \|x_t^{(l-1)}(\mathbf{x}_{t,\tau}) - x_t^{(l-1)}(\mathbf{x}_{t,S})\|_2 \right) + 1 \right)
$$

$$
= \mathrm{Poly}_t + \frac{1}{\sqrt{n_{l-1}}} \sup_{\tau \in [0,S]} \|x_t^{(l-1)}(\mathbf{x}_{t,\tau}) - x_t^{(l-1)}(\mathbf{x}_{t,S})\|_2
$$

$$
\leq \mathrm{Poly}_t + \frac{1}{\sqrt{n_{l-1}}} \cdot \underbrace{O_p(1) \cdot \mathrm{Poly}_t}_{\text{Lemma C.2}} = \left( 1 + \frac{O_p(1)}{\sqrt{n_{l-1}}} \right) \cdot \mathrm{Poly}_t. \tag{C.7}
$$

Combining, Eqs. (C.6) and (C.7), the term $\mathrm{I}_{t,l,l'}^{(2)}$ can then be bounded as follows,

$$
\mathrm{I}_{t,l,l'}^{(2)} \leq \left( 1 + \frac{O_p(1)}{\sqrt{\min_{l'' \in [0:L]}\{n_{l''}\}}} \right) \cdot \mathrm{Poly}_t \cdot \|\partial_{f(\mathbf{x})}\mathcal{L}(f_t(\mathbf{x}_{t,S}), \mathbf{y})\|_2
$$

$$
\leq O_p(1) \cdot \mathrm{Poly}_t \cdot \|\partial_{f(\mathbf{x})}\mathcal{L}(f_t(\mathbf{x}_{t,S}), \mathbf{y})\|_2. \tag{C.8}
$$

Finally, by inserting Eqs. (C.5) and (C.8) into Eq. (C.4), we have

$$
\mathrm{I}_t \leq O_p(1) \cdot \mathrm{Poly}_t \cdot \|\partial_{f(\mathbf{x})}\mathcal{L}(f_t(\mathbf{x}_{t,S}), \mathbf{y})\|_2. \tag{C.9}
$$

**Stage 2:** Bounding term $\mathrm{II}_{t,\tau}$ in Eq. (C.3).

We expand the term as follows,

$$
\mathrm{II}_{t,\tau} = \left\| \sum_i \partial_{[\mathbf{x}_{t,\tau}]_i} \partial_{\mathbf{x}} h_t^{(L+1)}(\mathbf{x}_{t,\tau}) \cdot \partial_t [\mathbf{x}_{t,\tau}]_i \right\|_2
$$

$$
= \left\| \sum_i \partial_{[\mathbf{x}_{t,\tau}]_i} \left( \prod_{l=L}^{1} \partial_{h_t^{(l)}(\mathbf{x})} h_t^{(l+1)}(\mathbf{x}_{t,\tau}) \cdot \partial_{\mathbf{x}} h_t^{(1)}(\mathbf{x}_{t,\tau}) \right) \cdot \partial_t [\mathbf{x}_{t,\tau}]_i \right\|_2
$$

$$
\leq \|\partial_{\mathbf{x}} h_t^{(1)}(\mathbf{x}_{t,\tau})\|_2 \cdot \sum_{l=1}^{L} \prod_{1 \leq l' \leq L, l' \neq l} \|\partial_{h_t^{(l')}(\mathbf{x})} h_t^{(l'+1)}(\mathbf{x}_{t,\tau})\|_2 \cdot \|\partial_{\mathbf{x}_{t,\tau}}(\partial_{h_t^{(l)}(\mathbf{x})} h_t^{(l+1)}(\mathbf{x}_{t,\tau})) \cdot \partial_t \mathbf{x}_{t,\tau}\|_2
$$

$$
\leq \mathrm{Poly}_t \cdot \|\partial_{\mathbf{x}_{t,\tau}} \mathrm{Diag}(\phi'(h_t^{(l)}(\mathbf{x}_{t,\tau}))) \cdot \partial_t \mathbf{x}_{t,\tau}\|_2.
$$

Since

$$
\|\partial_{\mathbf{x}_{t,\tau}} \mathrm{Diag}(\phi'(h_t^{(l)}(\mathbf{x}_{t,\tau}))) \cdot \partial_t \mathbf{x}_{t,\tau}\|_2
$$

$$
\leq \|\partial_{\mathbf{x}_{t,\tau}} \phi'(h_t^{(l)}(\mathbf{x}_{t,\tau})) \cdot \partial_t \mathbf{x}_{t,\tau}\|_2
$$

$$
\leq \|\phi''(h_t^{(l)}(\mathbf{x}_{t,\tau}))\|_2 \cdot \prod_{l'=1}^{l-1} \|\partial_{h^{(l')}(\mathbf{x})} h_t^{(l'+1)}(\mathbf{x}_{t,\tau})\|_2 \cdot \|\partial_{\mathbf{x}_{t,\tau}} h_t^{(1)}(\mathbf{x}_{t,\tau}) \cdot \partial_t \mathbf{x}_{t,\tau}\|_2
$$

$$
\leq \mathrm{Poly}_t \cdot \|\partial_{\mathbf{x}_{t,\tau}} h_t^{(1)}(\mathbf{x}_{t,\tau})\|_2 \cdot \|\partial_t \mathbf{x}_{t,\tau}\|_2 = \mathrm{Poly}_t \cdot \|\partial_t \mathbf{x}_{t,\tau}\|_2,
$$

therefore, $\mathrm{II}_{t,\tau}$ is eventually bounded as below,

$$
\mathrm{II}_{t,\tau} \leq \mathrm{Poly}_t \cdot \mathrm{Poly}_t \cdot \|\partial_t \mathbf{x}_{t,\tau}\|_2 = \mathrm{Poly}_t \cdot \|\partial_t \mathbf{x}_{t,\tau}\|_2. \tag{C.10}
$$

**Stage 3:** Bounding the original $\|\partial_t \mathbf{x}_{t,s}\|_2$ via the Grönwall's inequality (see Lemma A.2).

By inserting Eqs. (C.1), (C.9) and (C.10) into Eq. (C.3) and applying Assumption 2, we have that for every $s \in [0, S]$,

$$\|\partial_t \mathbf{x}_{t,s}\|_2$$
$$\leq O_p(1) \cdot \text{Poly}_t \cdot \|\partial_{f(\mathbf{x})}\mathcal{L}(f_t(\mathbf{x}_{t,S}), \mathbf{y})\|_2 \cdot C \cdot O_p(1)$$
$$+ \text{Poly}_t \cdot C \cdot \int_0^s \|\partial_t \mathbf{x}_{t,\tau}\|_2 \cdot \|\partial_{f(\mathbf{x})}\mathcal{L}(f_t(\mathbf{x}_{t,\tau}), \mathbf{y})\|_2 \mathrm{d}\tau + \text{Poly}_t \cdot C \cdot O_p(1) + \text{Poly}_t \cdot C \cdot O_p(1)$$
$$\leq \text{Poly}_t \cdot \int_0^s \|\partial_t \mathbf{x}_{t,\tau}\|_2 \cdot \|\partial_{f(\mathbf{x})}\mathcal{L}(f_t(\mathbf{x}_{t,\tau}), \mathbf{y})\|_2 \mathrm{d}\tau + O_p(1) \cdot \text{Poly}_t \cdot (\|\partial_{f(\mathbf{x})}\mathcal{L}(f_t(\mathbf{x}_{t,S}), \mathbf{y})\|_2 + 1).$$

Note that both $\|\partial_t \mathbf{x}_{t,s}\|_2$ and $\|\partial_{f(\mathbf{x})}\mathcal{L}(f_t(\mathbf{x}_{t,s}), \mathbf{y})\|_2$ are non-negative functions with respect to $s$ on the interval $[0, S]$. Therefore, by applying Grönwall's inequality (Lemma A.2), we have

$$\|\partial_t \mathbf{x}_{t,s}\|_2$$
$$\leq \exp\left(\text{Poly}_t \cdot \int_0^s \|\partial_{f(\mathbf{x})}\mathcal{L}(f_t(\mathbf{x}_{t,s}), \mathbf{y})\|_2 \mathrm{d}\tau\right) \cdot O_p(1) \cdot \text{Poly}_t \cdot (\|\partial_{f(\mathbf{x})}\mathcal{L}(f_t(\mathbf{x}_{t,S}), \mathbf{y})\|_2 + 1)$$
$$= \exp(O_p(1) \cdot \text{Poly}_t) \cdot O_p(1) \cdot \text{Poly}_t \cdot (\|\partial_{f(\mathbf{x})}\mathcal{L}(f_t(\mathbf{x}_{t,S}), \mathbf{y})\|_2 + 1).$$

The proof is completed. $\qquad\square$

**Lemma C.4.** *Suppose Assumptions 1 and 2 hold. Then, for every $t \in [0, T]$, we have that*
$$\|\partial_t h_t^{(l_1)}(\mathbf{x}_{t,S})\|_2, \ \|\partial_t x_t^{(l_2)}(\mathbf{x}_{t,S})\|_2 \leq \text{Poly}_t \cdot \left(\|\partial_{f(\mathbf{x})}\mathcal{L}(f_t(\mathbf{x}_{t,S}), \mathbf{y})\|_2 + \|\partial_t \mathbf{x}_{t,S}\|_2\right),$$
*where $l_1 \in [1 : L+1]$ and $l_2 \in [1 : L]$.*

*Proof.* For $\|\partial_t h_t^{(l_1)}(\mathbf{x}_{t,S})\|_2$, when $l_1 = 1$, by applying Assumption 1 and Lemma C.1, we have that

$$\|\partial_t h_t^{(1)}(\mathbf{x}_{t,S})\|_2 = \frac{1}{\sqrt{n_0}}\|\partial_t(W_t^{(1)} \cdot \mathbf{x}_{t,S})\|_2$$
$$\leq \frac{1}{\sqrt{n_0}} \cdot \left(\|\partial_t W_t^{(1)}\|_2 \cdot \|\mathbf{x}_{t,S}\|_2 + \|W_t^{(1)}\|_2 \cdot \|\partial_t \mathbf{x}_{t,S}\|_2\right)$$
$$= \text{Poly}_t \cdot \|\partial_{f(\mathbf{x})}\mathcal{L}(f_t(\mathbf{x}_{t,S}), \mathbf{y})\|_2 + \text{Poly}_t \cdot \|\partial_t \mathbf{x}_{t,S}\|_2. \qquad (\text{C.11})$$

Meanwhile, when $l_1 \geq 2$,

$$\|\partial_t h_t^{(l_1)}(\mathbf{x}_{t,S})\|_2 = \frac{1}{\sqrt{n_{l_1-1}}}\|\partial_t(W_t^{(l_1)} x_t^{(l_1-1)}(\mathbf{x}_{t,S}))\|_2$$
$$\leq \frac{1}{\sqrt{n_{l_1-1}}} \cdot \left(\|\partial_t W_t^{(l_1)}\|_2 \cdot \|x_t^{(l_1-1)}(\mathbf{x}_{t,S})\|_2 + \|W_t^{(l_1)}\|_2 \cdot \|\partial_t x_t^{(l_1-1)}(\mathbf{x}_{t,S})\|_2\right)$$
$$\leq \text{Poly}_t \cdot \|\partial_t W_t^{(l)}\|_2 + \text{Poly}_t \cdot \|\partial_t x_t^{(l-1)}(\mathbf{x}_{t,S})\|_2$$
$$\leq \text{Poly}_t \cdot \underbrace{\|\partial_{f(\mathbf{x})}\mathcal{L}(f_t(\mathbf{x}_{t,S}), \mathbf{y})\|_2}_{\text{Lemma C.1}} + \text{Poly}_t \cdot \underbrace{K \cdot \|\partial_t h_t^{(l-1)}(\mathbf{x}_{t,S})\|_2}_{\text{Assumption 1}}$$
$$\leq \cdots \leq \text{Poly}_t \cdot \|\partial_{f(\mathbf{x})}\mathcal{L}(f_t(\mathbf{x}_{t,S}), \mathbf{y})\|_2 + \text{Poly}_t \cdot \|\partial_t \mathbf{x}_{t,S}\|_2. \qquad (\text{C.12})$$

Combining Eqs.(C.11) and (C.12), we thus have for every $l_1 \in [1 : L+1]$,
$$\|\partial_t h_t^{(l_1)}(\mathbf{x}_{t,S})\|_2 \leq \text{Poly}_t \cdot \left(\|\partial_{f(\mathbf{x})}\mathcal{L}(f_t(\mathbf{x}_{t,S}), \mathbf{y})\|_2 + \|\partial_t \mathbf{x}_{t,S}\|_2\right).$$

On the other hand, by applying Assumption 1, for every $l_2 \in [1 : L]$,
$$\|\partial_t x_t^{(l_2)}(\mathbf{x}_{t,S})\|_2 \leq \|\partial_{h_t^{(l_2)}(\mathbf{x})} x_t^{(l_2)}(\mathbf{x}_{t,S})\|_2 \cdot \|\partial_t h_t^{(l_2)}(\mathbf{x}_{t,S})\|_2$$
$$\leq K \cdot \|\partial_t h_t^{(l_2)}(\mathbf{x}_{t,S})\|_2$$
$$\leq \text{Poly}_t \cdot \left(\|\partial_{f(\mathbf{x})}\mathcal{L}(f_t(\mathbf{x}_{t,S}), \mathbf{y})\|_2 + \|\partial_t \mathbf{x}_{t,S}\|_2\right).$$

The proof is completed. $\qquad\square$

Based on Lemmas C.1-C.4, we are now able to prove the following Lemma C.5.

**Lemma C.5.** *Suppose Assumptions 1, 2, 3 hold. Then as $\min_{l \in [0:L]}\{n_l\} \to \infty$, we have*

$$\sup_{t \in [0,T]} \max_{1 \leq l \leq L+1} \left\{ \frac{1}{\sqrt{n_{l-1}}} \|W_t^{(l)} - W_0^{(l)}\|_2, \ \frac{1}{\sqrt{n_{l-1}}} \|W_t^{(l)} - W_0^{(l)}\|_F \right\} \xrightarrow{P} 0,$$

$$\sup_{t \in [0,T]} \max_{1 \leq l \leq L} \left\{ \frac{1}{\sqrt{n_l}} \|h_t^{(l)}(\mathbf{x}_{t,S}) - h_0^{(l)}(\mathbf{x}_{0,S})\|_2 \right\} \xrightarrow{P} 0,$$

$$\sup_{t \in [0,T]} \max_{0 \leq l \leq L} \left\{ \frac{1}{\sqrt{n_l}} \|x_t^{(l)}(\mathbf{x}_{t,S}) - x_0^{(l)}(\mathbf{x}_{0,S})\|_2 \right\} \xrightarrow{P} 0.$$

*Proof.* Suppose $A_t$ denotes

$$A_t := \sum_{l=1}^{L+1} \frac{1}{\sqrt{n_{l-1}}} \left( \|W_0^{(l)}\|_F + \|W_t^{(l)} - W_0^{(l)}\|_F \right)$$

$$+ \sum_{l=0}^{L} \frac{1}{\sqrt{n_l}} \left( \|x_0^{(l)}(\mathbf{x}_{0,S})\|_2 + \|x_t^{(l)}(\mathbf{x}_{t,S}) - x_0^{(l)}(\mathbf{x}_{0,S})\|_2 \right)$$

$$+ \sum_{l=1}^{L} \frac{1}{\sqrt{n_l}} \left( \|h_0^{(l)}(\mathbf{x}_{0,S})\|_2 + \|h_t^{(l)}(\mathbf{x}_{t,S}) - h_0^{(l)}(\mathbf{x}_{0,S})\|_2 \right).$$

Then, by applying Lemmas C.1, C.4 and the fact that $\partial_t \| \cdot \|_2 \leq \|\partial_t \cdot \|_2$ holds for any given vector, we have the following,

$$\partial_t A_t \leq \sum_{l=1}^{L+1} \frac{1}{\sqrt{n_{l-1}}} \|\partial_t W_t^{(l)}\|_F + \sum_{l=0}^{L} \frac{1}{\sqrt{n_l}} \|\partial_t x_t^{(l)}(\mathbf{x}_{t,S})\|_2 + \sum_{l=1}^{L} \frac{1}{\sqrt{n_l}} \|\partial_t h_t^{(l)}(\mathbf{x}_{t,S})\|_2$$

$$\leq \frac{L+1}{\sqrt{\min_{l \in [0:L]}\{n_l\}}} \cdot \underbrace{\text{Poly}_t \cdot \|\partial_{f(\mathbf{x})} \mathcal{L}(f_t(\mathbf{x}_{t,S}), \mathbf{y})\|_2}_{\text{Lemma C.1}}$$

$$+ \frac{(L+1)+L}{\sqrt{\min_{l \in [0:L]}\{n_l\}}} \cdot \text{Poly}_t \cdot \underbrace{\left( \|\partial_{f(\mathbf{x})} \mathcal{L}(f_t(\mathbf{x}_{t,S}), \mathbf{y})\|_2 + \|\partial_t \mathbf{x}_{t,S}\|_2 \right)}_{\text{Lemma C.4}}$$

$$\leq \frac{1}{\sqrt{\min_{l \in [0:L]}\{n_l\}}} \cdot \left( \text{Poly}_t \cdot \|\partial_{f(\mathbf{x})} \mathcal{L}(f_t(\mathbf{x}_{t,S}), \mathbf{y})\|_2 + \|\partial_t \mathbf{x}_{t,S}\|_2 \right). \tag{C.13}$$

By further applying Lemma C.3 into Eq. (C.13), we have that for every $t \in [0,T]$,

$$\partial_t A_t$$

$$\leq \frac{\text{Poly}_t \cdot \|\partial_{f(\mathbf{x})} \mathcal{L}(f_t(\mathbf{x}_{t,S}), \mathbf{y})\|_2 + \exp(O_p(1) \cdot \text{Poly}_t) \cdot O_p(1) \cdot \text{Poly}_t \cdot (\|\partial_{f(\mathbf{x})} \mathcal{L}(f_t(\mathbf{x}_{t,S}), \mathbf{y})\|_2 + 1)}{\sqrt{\min_{l \in [0:L]}\{n_l\}}}$$

$$\leq \frac{1}{\sqrt{\min_{l \in [0:L]}\{n_l\}}} \cdot \left( 1 + e^{O_p(1) \cdot \text{Poly}_t} \right) \cdot O_p(1) \cdot \text{Poly}_t \cdot \left( 1 + \|\partial_{f(\mathbf{x})} \mathcal{L}(f_t(\mathbf{x}_{t,S}), \mathbf{y})\|_2 \right). \tag{C.14}$$

According to the definition, the polynomial $\text{Poly}_t$ in the above Eq. (C.14) is a deterministic combination of $\|W_t^{(l)}\|_F$, $\|x_t^{(l)}(\mathbf{x}_{t,S})\|_2$, and $\|h_t^{(l)}(\mathbf{x}_{t,S})\|_2$. Therefore, given the fact that

$$\begin{cases} \|W_t^{(l)}\|_F \leq \|W_0^{(l)}\|_F + \|W_t^{(l)} - W_0^{(l)}\|_F \\ \|x_t^{(l)}(\mathbf{x}_{t,S})\|_2 \leq \|x_0^{(l)}(\mathbf{x}_{0,S})\|_2 + \|x_t^{(l)}(\mathbf{x}_{t,S}) - x_0^{(l)}(\mathbf{x}_{0,S})\|_2 \ , \\ \|h_t^{(l)}(\mathbf{x}_{t,S})\|_2 \leq \|h_0^{(l)}(\mathbf{x}_{0,S})\|_2 + \|h_t^{(l)}(\mathbf{x}_{t,S}) - h_0^{(l)}(\mathbf{x}_{0,S})\|_2 \end{cases}$$

for the polynomial $\text{Poly}_t$ in Eq. (C.14), one can find a polynomial function $P(\cdot)$ with finite degree and finite positive coefficients such that

$$\text{Poly}_t \leq P(A_t)$$

holds for every $t \in [0, T]$. As a result, $\partial_t A_t$ can be further bounded as below,

$$\partial_t A_t \leq \frac{1}{\sqrt{\min_{l \in [0:L]}\{n_l\}}} \cdot \left(1 + e^{O_p(1) \cdot P(A_t)}\right) \cdot O_p(1) \cdot P(A_t) \cdot \left(1 + \|\partial_{f(\mathbf{x})}\mathcal{L}(f_t(\mathbf{x}_{t,S}), \mathbf{y})\|_2\right).$$

which means for every $t \in [0, T]$,

$$A_t - A_0 \leq \int_0^t \partial_\tau A_\tau \mathrm{d}\tau \leq \frac{1}{\sqrt{\min_{l \in [0:L]}\{n_l\}}} \cdot \int_0^t g(A_\tau, O_p(1)) \cdot \left(1 + \|\partial_{f(\mathbf{x})}\mathcal{L}(f_\tau(\mathbf{x}_{\tau,S}), \mathbf{y})\|_2\right)\mathrm{d}\tau,$$

where $g(x, y)$ is defined as

$$g(x, y) := y \cdot P(x) \cdot \left(1 + e^{y \cdot P(x)}\right).$$

By the definition of $O_p(1)$, we have that for any $\delta > 0$, there exists a finite $C > 0$ and a finite $N \in \mathbb{N}^+$ such that $\mathbb{P}(O_p(1) \leq C) \geq 1 - \delta$ as long as $\min_{l \in [0:L]}\{n_l\} > N$. Notice that $g(x, y)$ is increasing concerning $y$, thus when $\min_{l \in [0:L]}\{n_l\} > N$, with probability at least $1 - \delta$, we have

$$A_t - A_0 \leq \frac{1}{\sqrt{\min_{l \in [0:L]}\{n_l\}}} \cdot \int_0^t g(A_\tau, C) \cdot \left(1 + \|\partial_{f(\mathbf{x})}\mathcal{L}(f_\tau(\mathbf{x}_{\tau,S}), \mathbf{y})\|_2\right)\mathrm{d}\tau.$$

Furthermore, notice that (1) $(A_t - A_0)$ is non-negative on $[0, T]$ by definition, (2) $(1 + \|\partial_{f(\mathbf{x})}\mathcal{L}(f_t(\mathbf{x}_{t,S}), \mathbf{y})\|_2)$ is integrable on any sub-interval $[0, a] \subset [0, T]$, and (3) $g(x, C)$ is continuous and positive on $[0, +\infty)$ concerning $x$ and thus $1/g(x, C)$ is also integerable on any interval $[0, a]$. Therefore, we can apply a nonlinear form of the Grönwall's inequality, *i.e.*, Lemma A.3, as well as Assumption 2, and have that

$$\sup_{t \in [0,T]} G_C(A_t - A_0) \leq \frac{1}{\sqrt{\min_{l \in [0:L]}\{n_l\}}} \int_0^t \left(1 + \|\partial_{f(\mathbf{x})}\mathcal{L}(f_\tau(\mathbf{x}_{\tau,S}), \mathbf{y})\|_2\right)\mathrm{d}\tau$$

$$\leq \frac{O_p(1)}{\sqrt{\min_{l \in [0:L]}\{n_l\}}} \leq \frac{C}{\sqrt{\min_{l \in [0:L]}\{n_l\}}},$$

where $G_C(x) := \int_0^x \frac{\mathrm{d}\tau}{g(\tau, C)}$ is a continuous increasing function on $[0, +\infty)$ and thus the inverse function $G_C^{-1}$ also exists and is continuous on $[0, G_C(+\infty))$.

Thus, as $\min_{l \in [0:L]}\{n_l\} \to \infty$, we have that $\sup_{t \in [0:T]} G_C(A_t - A_0) \to 0$. Combining with the fact that $G_C(x) = 0$ in the range $[0, +\infty)$ if and only if $x = 0$ and the continuity of $G_C^{-1}$ on $[0, +\infty)$, we further have that with probability at least $1 - \delta$, $\sup_{t \in [0,T]}(A_t - A_0)$ can be arbitrarily small for sufficiently large $\min_{l \in [0:L]}\{n_l\}$.

The above justification suggests that as $\min_{l \in [0:L]}\{n_l\} \to \infty$,

$$\sup_{t \in [0,T]} \left\{ \frac{\|W_t^{(l)} - W_0^{(l)}\|_F}{\sqrt{n_{l-1}}}, \frac{\|h_t^{(l)}(\mathbf{x}_{t,S}) - h_0^{(l)}(\mathbf{x}_{0,S})\|_2}{\sqrt{n_l}}, \frac{\|x_t^{(l)}(\mathbf{x}_{t,S}) - x_0^{(l)}(\mathbf{x}_{0,S})\|_2}{\sqrt{n_l}} \right\}$$

$$\leq \sup_{t \in [0,T]}(A_t - A_0) \xrightarrow{P} 0.$$

The proof is completed. $\qquad \square$

## C.3 CONVERGENCES OF KERNELS DURING AT

This section aims to prove Theorems C.1 and C.2, which show that as network widths $n_0, \cdots, n_L$ approach infinite limits, the empirical NTK $\hat{\Theta}_{\theta,t}$ and the empirical ARK $\hat{\Theta}_{x,t}$ will converge to the deterministic kernels $\Theta_\theta$ and $\Theta_x$ defined in Theorem B.1 respectively.

### C.3.1 PREPARATIONS

We first show that rescaled network parameters $\frac{1}{\sqrt{n_{l-1}}}W_t^{(l)}$ are bounded during AT (Lemmas C.6, C.7), which thus indicates the optimization directions (Lemma C.8) as well as network outputs (Lemmas C.9, C.10, C.11) in each layer are also bounded. The proofs relies on the key Lemma C.5 presented in the previous section.

**Lemma C.6.** *Suppose there exists $\tilde{n} \in \mathbb{N}^+$ such that $\{n_0, \cdots, n_L\} \geq \tilde{n}$. For a given $l \in [1 : L+1]$, suppose $\frac{n_l}{n_{l-1}} = O_p(1)$ as $\tilde{n} \to \infty$. Then, we have that as $\tilde{n} \to \infty$,*

$$\frac{1}{\sqrt{n_{l-1}}}\|W_0^{(l)}\|_2 = O_p(1).$$

*Proof.* By definition, each entry of the matrix $\frac{1}{\sigma_W}W_0^{(l)} \in \mathbb{R}^{n_l \times n_{l-1}}$ is drawn from the standard Gaussian $\mathcal{N}(0, 1)$. Therefore, by applying Lemma A.4, we have for any $\delta > 0$, when $\tilde{n} \geq 2\ln\frac{2}{\delta}$, with probability at least $1 - \delta$,

$$\left\|\frac{1}{\sigma_W}W_0^{(l)}\right\|_2 \leq \sqrt{n_l} + \sqrt{n_{l-1}} + \sqrt{2\ln\frac{2}{\delta}} \leq \sqrt{n_l} + \sqrt{n_{l-1}} + \sqrt{\tilde{n}} \leq \sqrt{n_l} + \sqrt{n_{l-1}} + \sqrt{n_l},$$

which means

$$\frac{1}{\sqrt{n_{l-1}}}\|W_0^{(l)}\|_2 \leq \sigma_W \cdot \left(2 + \sqrt{\frac{n_l}{n_{l-1}}}\right) = 2\sigma_W + O_p(1) = O_p(1)$$

as $\tilde{n} \to \infty$. Therefore,

$$\frac{1}{\sqrt{n_{l-1}}}\|W_0^{(l)}\|_2 = O_p(1)$$

as $\tilde{n} \to \infty$, which completes the proof. $\qquad\square$

**Lemma C.7.** *Suppose Assumptions 1, 2 and 3 hold, and there exists $\tilde{n} \in \mathbb{N}^+$ such that $\{n_0, \cdots, n_L\} \geq \tilde{n}$. For a given $l \in [1 : L+1]$, suppose $\frac{n_l}{n_{l-1}} = O_p(1)$ as $\tilde{n} \to \infty$. Then, we have that as $\tilde{n} \to \infty$,*

$$\sup_{t \in [0,T]} \left\{\frac{1}{\sqrt{n_{l-1}}}\|W_t^{(l)}\|_2\right\} = O_p(1).$$

*Proof.* We have that

$$\sup_{t \in [0,T]} \left\{\frac{1}{\sqrt{n_{l-1}}}\|W_t^{(l)}\|_2\right\} \leq \underbrace{\frac{\|W_0^{(l)}\|_2}{\sqrt{n_{l-1}}}}_{\Upsilon_1} + \underbrace{\sup_{t \in [0,T]} \frac{\|W_t^{(l)} - W_0^{(l)}\|_2}{\sqrt{n_{l-1}}}}_{\Upsilon_2}.$$

By Lemma C.6, we have $\Upsilon_1 = O_p(1)$ as $\tilde{n} \to \infty$. By Lemma C.5, we have $\Upsilon_2 \xrightarrow{P} 0$ as $\tilde{n} \to \infty$, which further indicates $\Upsilon_2 = O_p(1)$. As a result,

$$\sup_{t \in [0,T]} \left\{\frac{1}{\sqrt{n_{l-1}}}\|W_t^{(l)}\|_2\right\} \leq O_p(1) + O_p(1) = O_p(1),$$

which demostrates that $\sup_{t \in [0,T]} \left\{\frac{1}{\sqrt{n_{l-1}}}\|W_t^{(l)}\|_2\right\} = O_p(1)$ as $\tilde{n} \to \infty$.

The proof is completed. $\qquad\square$

**Lemma C.8.** *Suppose Assumptions 1, 2 and 3 hold, and there exists $\tilde{n} \in \mathbb{N}^+$ such that $\{n_0, \cdots, n_L\} \geq \tilde{n}$. For a given $l \in [1 : L+1]$, suppose $\frac{n_{l'+1}}{n_{l'}} = O_p(1)$ as $\tilde{n} \to \infty$ holds for any $l \leq l' \leq L+1$. Then, as $\tilde{n} \to \infty$, we have that*

$$\sup_{t \in [0,T], s \in [0,S]} \|\partial_{h^{(l)}(\mathbf{x})}h_t^{(L+1)}(\mathbf{x}_{t,s})\|_2 = O_p(1).$$

*Proof.* Since

$$\sup_{t\in[0,T],s\in[0,S]}\|\partial_{h^{(l)}(\mathbf{x})}h_t^{(L+1)}(\mathbf{x}_{t,s})\|_2 \leq \prod_{l'=l}^{L}\sup_{t\in[0,T],s\in[0,S]}\|\partial_{h^{(l')}(\mathbf{x})}h_t^{(l'+1)}(\mathbf{x}_{t,s})\|_2$$

$$\leq \prod_{l'=l}^{L}\sup_{t\in[0,T]}\underbrace{K}_{\text{Assumption 1}}\cdot\frac{\|W_t^{(l'+1)}\|_2}{\sqrt{n_{l'}}} \leq \prod_{l'=l}^{L}K\cdot\underbrace{O_p(1)}_{\text{Lemma C.7}} = O_p(1),$$

thus $\sup_{t\in[0,T],s\in[0,S]}\|\partial_{h^{(l)}(\mathbf{x})}h_t^{(L+1)}(\mathbf{x}_{t,s})\|_2 = O_p(1)$ as $\tilde{n}\to\infty$.

The proof is completed. $\qquad\square$

**Lemma C.9.** *Suppose Assumptions 1, 2 and 3 hold, and there exists $\tilde{n}\in\mathbb{N}^+$ such that $\{n_0,\cdots,n_L\}\geq\tilde{n}$. Suppose for any $l\in[0:L]$, we have $\frac{n_{l+1}}{n_l}=O_p(1)$ as $\tilde{n}\to\infty$. Then, as $\tilde{n}\to\infty$,*

$$\max_{l\in[0:L]}\sup_{t\in[0,T]}\sup_{s_1,s_2\in[0,S]}\|x_t^{(l)}(\mathbf{x}_{t,s_1})-x_t^{(l)}(\mathbf{x}_{t,s_2})\|_2=O_p(1),$$

$$\max_{l\in[1:L+1]}\sup_{t\in[0,T]}\sup_{s_1,s_2\in[0,S]}\|h_t^{(l)}(\mathbf{x}_{t,s_1})-h_t^{(l)}(\mathbf{x}_{t,s_2})\|_2=O_p(1).$$

*Proof.* The proof is completed by applying Lemma C.8 to the proof of Lemma C.2. $\qquad\square$

**Lemma C.10.** *Suppose Assumptions 1, 2 and 3 hold, and there exists $\tilde{n}\in\mathbb{N}^+$ such that $\{n_0,\cdots,n_L\}\geq\tilde{n}$. For a given $l\in[1:L+1]$, suppose $\frac{n_{l'+1}}{n_{l'}}=O_p(1)$ as $\tilde{n}\to\infty$ holds for any $l\leq l'\leq L$. Then, as $\tilde{n}\to\infty$,*

$$\lim_{n_{l-1}\to\infty}\cdots\lim_{n_0\to\infty}\sup_{t\in[0,T],s\in[0,S]}\frac{1}{\sqrt{n_l}}\|h_t^{(l)}(\mathbf{x}_{t,s})\|_2=O_p(1),$$

$$\lim_{n_{l-1}\to\infty}\cdots\lim_{n_0\to\infty}\sup_{t\in[0,T],s\in[0,S]}\|h_t^{(l)}(\mathbf{x}_{t,s})-h_0^{(l)}(\mathbf{x})\|_2=O_p(1).$$

*Proof.* The proof is based on mathematical induction. For the base case where $l=1$, as $\tilde{n}\to\infty$,

$$\lim_{n_0\to\infty}\sup_{t\in[0,T],s\in[0,S]}\|h_t^{(1)}(\mathbf{x}_{t,s})-h_0^{(1)}(\mathbf{x})\|_2=\underbrace{\lim_{n_0\to\infty}\sup_{t\in[0,T]}\|h_t^{(1)}(\mathbf{x})-h_0^{(1)}(\mathbf{x})\|_2}_{\text{Lemma C.9}}+O_p(1)$$

$$\leq O_p(1)+\lim_{n_0\to\infty}\sup_{t\in[0,T]}\int_0^t\|\partial_\tau W_\tau^{(1)}\|_2\cdot\frac{\|\mathbf{x}\|_2}{\sqrt{n_0}}\mathrm{d}\tau$$

$$\leq O_p(1)+\lim_{n_0\to\infty}\frac{\|\mathbf{x}\|_2}{\sqrt{n_0}}\cdot\lim_{n_0\to\infty}\sup_{t\in[0,T]}\frac{\|\partial_{\mathrm{Vec}(W^{(1)})}h_t^{(1)}(\mathbf{x}_{t,S})\|_2}{\sqrt{n_0}}\cdot\|\partial_{h^{(1)}(\mathbf{x})}\cdot h^{(L+1)}(\mathbf{x}_{t,S})\|_2\cdot\int_0^T\|\partial_{f(\mathbf{x})}\mathcal{L}(f_\tau(\mathbf{x}),\mathbf{y})\|_2\mathrm{d}\tau$$

$$=O_p(1)+O_p(1)\cdot\lim_{n_0\to\infty}\sup_{t\in[0,T]}\frac{\|\mathbf{x}_{t,S}\|_2}{\sqrt{n_0}}\underbrace{O_p(1)}_{\text{Lemma C.8}}\cdot\underbrace{O_p(1)}_{\text{Assumption 2}}$$

$$\leq O_p(1)+\underbrace{\lim_{n_0\to\infty}\frac{\|\mathbf{x}\|_2+O_p(1)}{\sqrt{n_0}}}_{\text{Lemma C.9}}\cdot O_p(1)=O_p(1),$$

which indicates $\lim_{n_0\to\infty}\sup_{t\in[0,T],s\in[0,S]}\|h_t^{(1)}(\mathbf{x}_{t,s})-h_0^{(1)}(\mathbf{x})\|_2=O_p(1)$.

As a result,

$$\lim_{n_0\to\infty}\sup_{t\in[0,T],s\in[0,S]}\frac{1}{\sqrt{n_1}}\|h_t^{(1)}(\mathbf{x}_{t,s})\|_2 \leq \lim_{n_0\to\infty}\frac{1}{\sqrt{n_1}}\|h_0^{(1)}(\mathbf{x})\|_2+\frac{O_p(1)}{\sqrt{n_1}}$$

$$\leq \underbrace{\sum_{m=1}^{M}\sqrt{\Sigma^{(1)}(x_m,x_m)}}_{\text{Lemma B.1}}+O_p(1)=O_p(1),$$

which indicates $\lim_{n_0 \to \infty} \sup_{t \in [0,T], s \in [0,S]} \frac{1}{\sqrt{n_1}} \|h_t^{(1)}(\mathbf{x}_{t,s})\|_2 = O_p(1)$.

Then, for the induction step, suppose the lemma already holds for the $l$-th case and we aim to show it also holds for the $(l+1)$-th case. Following the same derivation as that in the proof of base case, we have as $\tilde{n} \to \infty$,

$$\lim_{n_{l-1} \to \infty} \cdots \lim_{n_0 \to \infty} \sup_{t \in [0,T], s \in [0,S]} \|h_t^{(l+1)}(\mathbf{x}_{t,s}) - h_0^{(l+1)}(\mathbf{x})\|_2$$

$$\leq O_p(1) + \lim_{n_{l-1} \to \infty} \cdots \lim_{n_0 \to \infty} \frac{\|x_0^{(l)}(\mathbf{x})\|_2}{\sqrt{n_l}} \cdot \lim_{n_{l-1} \to \infty} \cdots \lim_{n_0 \to \infty} \sup_{t \in [0,T]} \frac{\|x_t^{(l)}(\mathbf{x}_{t,S})\|_2}{\sqrt{n_l}} \cdot O_p(1)$$

$$\leq O_p(1) + \underbrace{\sum_{i=1}^M \sqrt{\mathbb{E}_{f \sim \mathcal{GP}(0, \Sigma^{(l)})}[\phi(f(x_m))^2]}}_{\text{Lemma B.1}} \cdot \underbrace{\lim_{n_{l-1}} \cdots \lim_{n_0 \to \infty} \frac{\|x_0^{(l)}(\mathbf{x})\|_2 + O_p(1)}{\sqrt{n_l}}}_{\text{Induction hypothesis}} \cdot O_p(1)$$

$$\leq O_p(1) + O_p(1) \cdot \underbrace{\sum_{i=1}^M \sqrt{\mathbb{E}_{f \sim \mathcal{GP}(0, \Sigma^{(l)})}[\phi(f(x_m))^2]}}_{\text{Lemma B.1}} \cdot O_p(1) = O_p(1).$$

Therefore,

$$\lim_{n_{l-1} \to \infty} \cdots \lim_{n_0 \to \infty} \sup_{t \in [0,T], s \in [0,S]} \frac{1}{\sqrt{n_l}} \|h_t^{(l)}(\mathbf{x}_{t,s})\|_2$$

$$\leq \lim_{n_{l-1} \to \infty} \cdots \lim_{n_0 \to \infty} \frac{1}{\sqrt{n_l}} \|h_0^{(l)}(\mathbf{x})\|_2 + \frac{O_p(1)}{\sqrt{n_l}}$$

$$= \underbrace{\sum_{m=1}^M \sqrt{\Sigma^{(l)}(x_m, x_m)}}_{\text{Lemma B.1}} + O_p(1) = O_p(1).$$

The proof is completed. $\qquad\square$

**Lemma C.11.** *Suppose Assumptions 1, 2 and 3 hold, and there exists $\tilde{n} \in \mathbb{N}^+$ such that $\{n_0, \cdots, n_L\} \geq \tilde{n}$. For a given $l \in [1 : L+1]$, suppose $\frac{n_{l'+1}}{n_{l'}} = O_p(1)$ as $\tilde{n} \to \infty$ holds for any $l \leq l' \leq L$. Then, for any $x \in \mathcal{X}$, as $\tilde{n} \to \infty$,*

$$\lim_{n_{l-1} \to \infty} \cdots \lim_{n_0 \to \infty} \sup_{t \in [0,T]} \frac{1}{\sqrt{n_l}} \|h_t^{(l)}(x)\|_2 = O_p(1),$$

$$\lim_{n_{l-1} \to \infty} \cdots \lim_{n_0 \to \infty} \sup_{t \in [0,T]} \|h_t^{(l)}(x) - h_0^{(l)}(x)\|_2 = O_p(1).$$

*Proof.* The proof is completed by following the proof of Lemma C.10 and also applying Lemma C.10 itself. $\qquad\square$

### C.3.2 PROOF OF THEOREM C.1

Based on Lemmas C.6-C.11, we are now able to prove Theorem C.1 as follows.

*Proof of **Theorem C.1**.* To prove Theorem C.1, it is enough to prove the following Claim C.1.

**Claim C.1.** *For a given $l \in [L+1]$, suppose $\frac{n_{l+1}}{n_l}, \cdots, \frac{n_{L+1}}{n_L} = O_p(1)$, Then, for any $m, m' \in [M]$, as $\tilde{n} \to \infty$,*

$$\lim_{n_{l-1} \to \infty} \cdots \lim_{n_0 \to \infty} \sup_{t \in [0,T], s \in [0,S]} \|\hat{\Theta}_{\theta,t}^{(l)}(x_{m,t,s}, x_{m',t,s}) - \Theta_\theta^{(l)}(x_m, x_{m'})\|_2 \xrightarrow{P} 0,$$

$$\lim_{n_{l-1} \to \infty} \cdots \lim_{n_0 \to \infty} \sup_{t \in [0,T], s \in [0,S]} \|\hat{\Theta}_{x,t}^{(l)}(x_{m,t,s}, x_{m,t,s}) - \Theta_x^{(l)}(x_m, x_m)\|_2 \xrightarrow{P} 0.$$

Theorem C.1 can be directly obtained by setting $l = L + 1$ in Claim C.1. Therefore, we now turn to prove this claim.

*Proof.* The proof is based on mathematical induction. For the base case where $l = 1$, for $\hat{\Theta}_{\theta,t}^{(1)}$, as $\tilde{n} \to \infty$,

$$\sup_{t \in [0,T], s \in [0,S]} \|\hat{\Theta}_{\theta,t}^{(1)}(x_{m,t,s}, x_{m',t,s}) - \hat{\Theta}_{\theta,0}^{(1)}(x_m, x_{m'})\|_2 = \sup_{t \in [0,T], s \in [0,S]} \frac{\|x_{m,t,s}^T x_{m',t,s} - x_m^T x_{m'}\|_2}{n_0}.$$

By Lemma C.10 and Assumption 1, we have

$$\lim_{n_0 \to \infty} \frac{\|x_{m,t,s} - x_m\|_2}{\sqrt{n_0}} = \lim_{n_0 \to \infty} \frac{O_p(1)}{\sqrt{n_0}} \xrightarrow{P} 0,$$

which thus indicates

$$\lim_{n_0 \to \infty} \sup_{t \in [0,T], s \in [0,S]} \|\hat{\Theta}_{\theta,t}^{(1)}(x_{m,t,s}, x_{m',t,s}) - \hat{\Theta}_{\theta,0}^{(1)}(x_m, x_{m'})\|_2 \xrightarrow{P} 0.$$

Besides, for $\hat{\Theta}_{x,t}^{(1)}$, we have

$$\sup_{t \in [0,T], s \in [0,S]} \|\hat{\Theta}_{x,t}^{(1)}(x_{m,t,s}, x_{m,t,s}) - \Theta_x^{(1)}(x_m, x_m)\|_2 = \sup_{t \in [0,T], s \in [0,S]} \frac{\|W_t^{(1)} W_t^{(1)^T} - W_0^{(1)} W_0^{(1)^T}\|_2}{n_0}.$$

By Lemma C.5, we have $\lim_{n_0 \to \infty} \frac{\|W_t^{(1)} - W_0^{(1)}\|_2}{\sqrt{n_0}} \xrightarrow{P} 0$ as $\tilde{n} \to \infty$, which further demonstrates

$$\lim_{n_0 \to \infty} \sup_{t \in [0,T], s \in [0,S]} \|\hat{\Theta}_{x,t}^{(1)}(x_{m,t,s}, x_{m,t,s}) - \Theta_x^{(1)}(x_m, x_m)\|_2 \xrightarrow{P} 0$$

and thereby completes the proof for the base case.

For the induction step, suppose Claim C.1 already holds for the $l$-th case and we want to prove it also holds for the case $(l + 1)$ in which $\frac{n_{l+2}}{n_{l+1}}, \cdots, \frac{n_{L+1}}{n_L} = O_p(1)$ holds.

To do so, we first prove technical Claims C.2 and C.3.

**Claim C.2.** *As $\tilde{n} \to \infty$,*

$$\lim_{n_l \to \infty} \cdots \lim_{n_0 \to \infty} \sup_{t \in [0,T]} \|h_t^{(l)}(x_m) - h_0^{(l)}(x_m)\|_\infty \xrightarrow{P} 0.$$

*Proof.* As $\tilde{n} \to \infty$, for every $j \in [n_l]$, we uniformly have that

$$\sup_{t \in [0,T]} \|h_t^{(l)}(x_m) - h_0^{(l)}(x_m)\|_\infty$$

$$\leq K \cdot \sup_{t \in [0,T]} \max_{j \in [n_l]} \int_0^t |\partial_\tau h_\tau^{(l)}(x_m)_j| d\tau \leq K \cdot \max_{j \in [n_l]} \int_0^T |\partial_\theta h_\tau^{(l)}(x_m)_j \cdot \partial_\tau \theta_t| d\tau$$

$$= K \cdot \max_{j \in [n_l]} \int_0^T |\partial_\theta h_\tau^{(l)}(x_m)_j \cdot \partial_\theta^T h_\tau^{(l)}(\mathbf{x}_{t,S}) \cdot \partial_{h^{(l)}(\mathbf{x})}^T h_t^{(l+1)}(\mathbf{x}_{t,S}) \cdot \partial_{h^{(l+1)}(\mathbf{x})}^T h_\tau^{(L+1)}(\mathbf{x}_{t,S}) \cdot \partial_{f(\mathbf{x})}^T \mathcal{L}(f_\tau(\mathbf{x}_{t,S}), \mathbf{y})| d\tau$$

$$\leq K \cdot \max_{j \in [n_l]} \int_0^T \|\hat{\Theta}_{\theta,\tau}^{(l)}(x_m, \mathbf{x}_{\tau,S})_{j,:} \cdot \partial_{h^{(l)}(\mathbf{x})}^T h_\tau^{(l+1)}(\mathbf{x}_{\tau,S})\|_2 \cdot \|\partial_{h^{(l+1)}(\mathbf{x})} h_\tau^{(L+1)}(\mathbf{x}_{\tau,S})\|_2 \cdot \|\partial_{f(\mathbf{x})} \mathcal{L}(f_\tau(\mathbf{x}_{\tau,S}), \mathbf{y})\|_2 d\tau$$

$$\leq K \cdot \sup_{\tau \in [0,T]} \max_{j \in [n_l]} \frac{1}{\sqrt{n_l}} \|\hat{\Theta}_{\theta,\tau}^{(l)}(x_m, \mathbf{x}_{\tau,S})_{j,:} \cdot \mathrm{Diag}(\underbrace{(W_\tau^{(l+1)}, \cdots, W_\tau^{(l+1)})}_{M \text{ matrices } W_\tau^{(l+1)} \text{ at all}})^T\|_2 \cdot \underbrace{O_p(1)}_{\text{Lemma C.8}} \cdot \underbrace{O_p(1)}_{\text{Assumption 2}}$$

$$\leq K \cdot \max_{m' \in [M]} \sup_{\tau \in [0,T]} \max_{j \in [n_l]} \frac{1}{\sqrt{n_l}} \|\hat{\Theta}_{\theta,\tau}^{(l)}(x_m, x_{m',\tau,S})_{j,:} \cdot W_\tau^{(l+1)^T}\|_2 \cdot O_p(1), \tag{C.15}$$

where $\hat{\Theta}_{\theta,\tau}^{(l)}(\cdot, \cdot)_{j,:}$ denotes the $j$-th row of $\hat{\Theta}_{\theta,\tau}^{(l)}(\cdot, \cdot)$ and $W_{i,:,\tau}^{(l+1)}$ denotes the $i$-th row of $W_\tau^{(l+1)}$.

Then, by the induction hypothesis and Lemma C.5, we have

$$\lim_{n_l \to \infty} \cdots \lim_{n_0 \to \infty} \max_{m' \in [M]} \sup_{\tau \in [0,T]} \max_{j \in [n_l]} \|\hat{\Theta}_{\theta,\tau}^{(l)}(x_m, x_{m',\tau,S})_{j,:} - \Theta_\theta^{(l)}(x_m, x_{m'})_{j,:}\|_2 \xrightarrow{P} 0,$$

$$\lim_{n_l \to \infty} \cdots \lim_{n_0 \to \infty} \max_{m' \in [M]} \sup_{\tau \in [0,T]} \max_{j \in [n_l]} \frac{\|W_t^{(l+1)} - W_0^{(l+1)}\|_2}{\sqrt{n_l}} \xrightarrow{P} 0,$$

which indicates

$$\lim_{n_l \to \infty} \cdots \lim_{n_0 \to \infty} \max_{m' \in [M]} \sup_{\tau \in [0,T]} \max_{j \in [n_l]} \frac{1}{\sqrt{n_l}} \|\hat{\Theta}_{\theta,\tau}^{(l)}(x_m, x_{m',\tau,S})_{j,:} \cdot W_\tau^{(l+1)^T}\|_2$$

$$\xrightarrow{P} \lim_{n_l \to \infty} \max_{m' \in [M]} \max_{j \in [n_l]} \frac{1}{\sqrt{n_l}} \|\Theta_\theta^{(l)}(x_m, x_{m'})_{j,:} \cdot W_0^{(l+1)^T}\|_2$$

$$\leq \lim_{n_l \to \infty} \max_{m' \in [M]} \max_{i \in [n_{l+1}], j \in [n_l]} \frac{\sqrt{n_{l+1}} \cdot |\Theta_\theta^{\infty,(l)}(x_m, x_{m'})| \cdot |W_{i,j,0}^{(l+1)}|}{\sqrt{n_l}}$$

$$= \sqrt{n_{l+1}} \cdot \max_{m' \in [M]} |\Theta_\theta^{\infty,(l)}(x_m, x_{m'})| \cdot \max_{i \in [n_{l+1}]} \lim_{n_l \to \infty} \frac{\max_{j \in [n_l]} |W_{i,j,0}^{(l+1)}|}{\sqrt{n_l}}$$

$$= \sqrt{n_{l+1}} \cdot \max_{m' \in [M]} |\Theta_\theta^{\infty,(l)}(x_m, x_{m'})| \cdot \underbrace{0}_{\text{Lemma A.5}} = 0. \tag{C.16}$$

Combining Eqs. (C.15) and (C.16), we finally have

$$\lim_{n_l \to \infty} \cdots \lim_{n_0 \to \infty} \sup_{t \in [0,T]} \|h_t^{(l)}(x_m) - h_0^{(l)}(x_m)\|_\infty \xrightarrow{P} 0.$$

The proof is completed. $\qquad\square$

**Claim C.3.** *As $\tilde{n} \to \infty$,*

$$\lim_{n_l \to \infty} \cdots \lim_{n_0 \to \infty} \sup_{t \in [0,T]} \|h_t^{(l)}(x_{m,t,s}) - h_0^{(l)}(x_m)\|_\infty \xrightarrow{P} 0.$$

*Proof.* We first bound $\|h_t^{(l)}(x_{m,t,s}) - h_t^{(l)}(x_m)\|_\infty$. As $\tilde{n} \to \infty$, have that

$$\sup_{t \in [0,T], s \in [0,S]} \|h_t^{(l)}(x_{m,t,s}) - h_t^{(l)}(x_m)\|_\infty$$

$$\leq K \cdot \sup_{t \in [0,T], s \in [0,S]} \max_{j \in [n_l]} \int_0^s |\partial_\tau h_t^{(l)}(x_{m,t,\tau})_j| d\tau \leq K \cdot \sup_{t \in [0,T]} \max_{j \in [n_l]} \int_0^S |\partial_x h_t^{(l)}(x_{m,t,\tau})_j \cdot \partial_\tau x_{m,t,\tau}| d\tau$$

$$\leq K \cdot \sup_{t \in [0,T], \tau \in [0,S]} \max_{j \in [n_l]} \|\hat{\Theta}_{x,t}^{(l)}(x_{m,t,\tau}, x_{m,t,\tau})_{j,:} \cdot \partial_{h^{(l)}(x)}^T h_t^{(l+1)}(x_{m,t,\tau})\|_2$$

$$\cdot \sup_{t \in [0,T], \tau \in [0,S]} \|\partial_{h^{(l+1)}(x)} h_t^{(L+1)}(x_{m,t,\tau})\|_2 \cdot \sup_{t \in [0,T]} \|\boldsymbol{\eta}(t)\|_2 \cdot \int_0^S \|\partial_{f(x)} \mathcal{L}(f_t(x_{m,t,s}), y_m)\|_2 d\tau$$

$$\leq \sup_{t \in [0,T], \tau \in [0,S]} \max_{j \in [n_l]} \frac{1}{\sqrt{n_l}} \|\hat{\Theta}_{x,t}^{(l)}(x_{m,t,\tau}, x_{m,t,\tau})_{j,:} \cdot W_t^{(l+1)^T}\|_2 \cdot \underbrace{O_p(1)}_{\text{Lemma C.8}} \cdot \underbrace{O_p(1)}_{\text{Assumption 3}} \cdot \underbrace{O_p(1)}_{\text{Assumption 2}}, \tag{C.17}$$

where $\hat{\Theta}_{x,t}^{(l)}(x_{m,t,\tau}, x_{m,t,\tau})_{j,:}$ is the $j$-th row of $\hat{\Theta}_{x,t}^{(l)}(x_{m,t,\tau}, x_{m,t,\tau})$.

Similar to that in the proof of Claim C.2, by the induction hypothesis and Lemma C.5, we have

$$\lim_{n_l \to \infty} \cdots \lim_{n_0 \to \infty} \sup_{t \in [0,T], \tau \in [0,S]} \max_{j \in [n_l]} \|\hat{\Theta}_{x,t}^{(l)}(x_{m,t,\tau}, x_{m,t,\tau})_{j,:} - \Theta_x^{(l)}(x_m, x_{m'})_{j,:}\|_2 \xrightarrow{P} 0,$$

$$\lim_{n_l \to \infty} \cdots \lim_{n_0 \to \infty} \sup_{t \in [0,T]} \frac{\|W_t^{(l+1)} - W_0^{(l+1)}\|_2}{\sqrt{n_l}} \xrightarrow{P} 0,$$

which leads to

$$\lim_{n_l \to \infty} \cdots \lim_{n_0 \to \infty} \sup_{t \in [0,T], \tau \in [0,S]} \max_{j \in [n_l]} \frac{1}{\sqrt{n_l}} \|\hat{\Theta}_{x,t}^{(l)}(x_{m,t,\tau}, x_{m,t,\tau})_{j,:} \cdot W_t^{(l+1)^T}\|_2$$

$$\xrightarrow{P} \lim_{n_l \to \infty} \cdots \lim_{n_0 \to \infty} \max_{j \in [n_l]} \frac{1}{\sqrt{n_l}} \|\Theta_x^{(l)}(x_m, x_m)_{j,:} \cdot W_0^{(l+1)^T}\|_2$$

$$= \lim_{n_l \to \infty} \max_{j \in [n_l]} \frac{1}{\sqrt{n_l}} |\Theta_x^{\infty,(l)}(x_m, x_m)| \cdot \|W_{:,j,0}^{(l+1)}\|_2$$

$$\leq \sqrt{n_{l+1}} \cdot |\Theta_x^{\infty,(l)}(x_m, x_m)| \cdot \max_{i \in [n_{l+1}]} \lim_{n_l \to \infty} \frac{\max_{j \in [n_l]} |W_{i,j,0}^{(l+1)}|}{\sqrt{n_l}}$$

$$\leq \sqrt{n_{l+1}} \cdot |\Theta_x^{\infty,(l)}(x_m, x_m)| \cdot \underbrace{0}_{\text{Lemma A.5}} = 0. \tag{C.18}$$

Combining Eqs.(C.17) and (C.18), we thus have that as $\tilde{n} \to \infty$,

$$\lim_{n_l \to \infty} \cdots \lim_{n_0 \to \infty} \sup_{t \in [0,T], s \in [0,S]} \|h_t^{(l)}(x_{m,t,s}) - h_t^{(l)}(x_m)\|_\infty \xrightarrow{P} 0.$$

As a result, by further applying Claim C.2,

$$\lim_{n_l \to \infty} \cdots \lim_{n_0 \to \infty} \sup_{t \in [0,T], s \in [0,S]} \|h_t^{(l)}(x_{m,t,s}) - h_0^{(l)}(x_m)\|_\infty$$

$$\leq \lim_{n_l \to \infty} \cdots \lim_{n_0 \to \infty} \sup_{t \in [0,T], s \in [0,S]} \left( \|h_t^{(l)}(h_{m,t,s}) - x_t^{(l)}(x_m)\|_\infty + \|h_t^{(l)}(x_m) - h_0^{(l)}(x_m)\|_\infty \right)$$

$$\xrightarrow{P} 0 + 0 = 0.$$

The proof is completed. $\qquad\qquad\square$

Based on Claims C.2 and C.3, we are now able to prove the induction step. Specifically, for the NTK $\hat{\Theta}_{\theta,t}^{(l+1)}$, we have

$$\|\hat{\Theta}_{\theta,t}^{(l+1)}(x_{m,t,s}, x_{m',t,s}) - \hat{\Theta}_{\theta,0}^{(l+1)}(x_m, x_{m'})\|_2$$

$$\leq \underbrace{\|D_t^{(l+1)}(x_{m,t,s}) \cdot \hat{\Theta}_{\theta,t}^{(l)}(x_{m,t,s}, x_{m',t,s}) \cdot D_t^{(l+1)}(x_{m',t,s})^T - D_0^{(l+1)}(x_m) \cdot \hat{\Theta}_{\theta,0}^{(l)}(x_m, x_{m'}) \cdot D_0^{(l+1)}(x_{m'})^T\|_2}_{\text{I}_{t,s}}$$

$$+ \underbrace{\frac{|x_t^{(l)}(x_{m,t,s})^T x_t^{(l)}(x_{m',t,s}) - x_t^{(l)}(x_m)^T x_t^{(l)}(x_{m'})|}{n_l}}_{\text{II}_{t,s}}$$

where $D_t^{(l+1)}(x) := \frac{W_t^{(l+1)} \cdot \text{Diag}(\phi'(h_t^{(l)}(x)))}{\sqrt{n_l}}$.

For $\text{Diag}(\phi'(h_t^{(l)}(x_{m,t,s})))$ in $D_t^{(l+1)}(x_{m,t,s})$, by Claim C.3,

$$\lim_{n_l \to \infty} \cdots \lim_{n_0 \to \infty} \sup_{t \in [0,T], s \in [0,S]} \|\text{Diag}(\phi'(h_t^{(l)}(x_{m,t,s}))) - \text{Diag}(\phi'(h_0^{(l)}(x_m)))\|_2$$

$$= \lim_{n_l \to \infty} \cdots \lim_{n_0 \to \infty} \sup_{t \in [0,T], s \in [0,S]} \|\phi'(h_t^{(l)}(x_{m,t,s})) - \phi'(h_0^{(l)}(x_m))\|_\infty$$

$$\leq \lim_{n_l \to \infty} \cdots \lim_{n_0 \to \infty} \sup_{t \in [0,T], s \in [0,S]} K \cdot \|h_t^{(l)}(x_{m,t,s}) - h_0^{(l)}(x_m)\|_\infty$$

$$\xrightarrow{P} 0.$$

Combining with Lemma C.5 which shows that $\lim_{n_l \to \infty} \cdots \lim_{n_0 \to \infty} \sup_{t \in [0,T]} \frac{\|W_t^{(l+1)} - W_0^{(l+1)}\|_2}{\sqrt{n_l}} \xrightarrow{P} 0$ as $\tilde{n} \to \infty$, we thus have

$$\lim_{n_l \to \infty} \cdots \lim_{n_0 \to \infty} \sup_{t \in [0,T], s \in [0,S]} \|D_t^{(l+1)}(x_{m,t,s}) - D_0^{(l+1)}(x_m)\|_2 \xrightarrow{P} 0.$$

Further applying the induction hypothesis that $\lim_{n_l \to \infty} \cdots \lim_{n_0 \to \infty} \sup_{t \in [0,T], s \in [0,S]} \|\hat{\Theta}^{(l)}_{\theta,t}(x_{m,t,s}, x_{m',t,s}) - \hat{\Theta}^{(l)}_{\theta,0}(x_m, x_{m'})\|_2 \xrightarrow{P} 0$, we finally have

$$\lim_{n_l \to \infty} \cdots \lim_{n_0 \to \infty} \sup_{t \in [0,T], s \in [0,S]} \mathrm{I}_{t,s} \xrightarrow{P} 0.$$

Besides, for term $\mathrm{II}_{t,s}$, by Lemma C.10 and Assumption 1, we have

$$\lim_{n_l \to \infty} \cdots \lim_{n_0 \to \infty} \sup_{t \in [0,T], s \in [0,S]} \frac{\|x^{(l)}_t(x_{m,t,s}) - x^{(l)}_0(x_m)\|_2}{\sqrt{n_l}}$$

$$\leq \lim_{n_l \to \infty} \cdots \lim_{n_0 \to \infty} \sup_{t \in [0,T], s \in [0,S]} \frac{K \cdot \|h^{(l)}_t(x_{m,t,s}) - h^{(l)}_0(x_m)\|_2}{\sqrt{n_l}}$$

$$\leq \lim_{n_l \to \infty} \cdots \lim_{n_0 \to \infty} \frac{K \cdot O_p(1)}{\sqrt{n_l}} \xrightarrow{P} 0,$$

which leads to

$$\lim_{n_l \to \infty} \cdots \lim_{n_0 \to \infty} \sup_{t \in [0,T], s \in [0,S]} \mathrm{II}_{t,s} \xrightarrow{P} 0.$$

Finally, Based on the convergences of $\mathrm{I}_{t,s}$ and $\mathrm{II}_{t,s}$, we have that as $\tilde{n} \to \infty$,

$$\lim_{n_l \to \infty} \cdots \lim_{n_0 \to \infty} \|\hat{\Theta}^{(l+1)}_{\theta,t}(x_{m,t,s}, x_{m',t,s}) - \hat{\Theta}^{(l+1)}_{\theta,0}(x_m, x_{m'})\|_2 \xrightarrow{P} 0.$$

According to Theorem B.1, $\lim_{n_l \to \infty} \cdots \lim_{n_0 \to \infty} \hat{\Theta}^{(l+1)}_{\theta,0}(x_m, x_{m'}) = \Theta^{(l+1)}_\theta(x_m, x_{m'})$, which thus indicates

$$\lim_{n_l \to \infty} \cdots \lim_{n_0 \to \infty} \|\hat{\Theta}^{(l+1)}_{\theta,t}(x_{m,t,s}, x_{m',t,s}) - \Theta^{(l+1)}_\theta(x_m, x_{m'})\|_2 \xrightarrow{P} 0.$$

Then, for the ARK $\hat{\Theta}^{(l+1)}_{x,t}$, we have

$$\|\hat{\Theta}^{(l+1)}_{x,t}(x_{m,t,s}, x_{m',t,s}) - \hat{\Theta}^{(l+1)}_{x,0}(x_m, x_{m'})\|_2$$
$$\leq \|D^{(l+1)}_t(x_{m,t,s}) \cdot \hat{\Theta}^{(l)}_{x,t}(x_{m,t,s}, x_{m',t,s}) \cdot D^{(l+1)}_t(x_{m',t,s})^T - D^{(l+1)}_0(x_m) \cdot \hat{\Theta}^{(l)}_{x,0}(x_m, x_{m'}) \cdot D^{(l+1)}_0(x_{m'})^T\|_2.$$

Therefore, following similar derivation as that for $\hat{\Theta}^{(l+1)}_{\theta,t}$, we will have

$$\lim_{n_l \to \infty} \cdots \lim_{n_0 \to \infty} \|\hat{\Theta}^{(l+1)}_{x,t}(x_{m,t,s}, x_{m',t,s}) - \Theta^{(l+1)}_x(x_m, x_{m'})\|_2 \xrightarrow{P} 0$$

as $\tilde{n} \to \infty$, which justifies the induction step and also completes the proof of Claim C.1. $\qquad\square$

The proof is completed. $\qquad\square$

### C.3.3  PROOF OF THEOREM C.2

Theorem C.2 is similarly proved as follows.

*Proof of **Theorem C.2**.* The proof is similar to that of Theorem C.1. Specifically, to prove Theorem C.2, it is enough to prove the following Claim C.4.

**Claim C.4.** *For a given $l \in [L+1]$, suppose $\frac{n_{l+1}}{n_l}, \cdots, \frac{n_{L+1}}{n_L} = O_p(1)$, Then, for any $m \in [M]$, as $\tilde{n} \to \infty$, we have that*

$$\lim_{n_{l-1} \to \infty} \cdots \lim_{n_0 \to \infty} \sup_{t \in [0,T], s \in [0,S]} \|\hat{\Theta}^{(l)}_{\theta,t}(x, x_{m,t,s}) - \Theta^{(l)}_\theta(x, x_m)\|_2 \xrightarrow{P} 0,$$

$$\lim_{n_{l-1} \to \infty} \cdots \lim_{n_0 \to \infty} \sup_{t \in [0,T], s \in [0,S]} \|\hat{\Theta}^{(l)}_{x,t}(x, x_{m,t,s}) - \Theta^{(l)}_x(x, x_m)\|_2 \xrightarrow{P} 0.$$

Then, Theorem C.2 can be directly obtained by setting $l = L + 1$ in Claim C.4.

The proof of Claim C.4 basically follows that of Claim C.1. The only difference is that Claim C.4 requires one to further prove that

$$\lim_{n_l \to \infty} \cdots \lim_{n_0 \to \infty} \sup_{t \in [0,T]} \|h_t^{(l)}(x) - h_0^{(l)}(x)\|_\infty \xrightarrow{P} 0$$

for any $x \in \mathcal{X}$ as $\tilde{n} \to \infty$ in the $(l+1)$-th induction step, which can be done by using Lemma C.11 and following the derivation in Claim C.2. $\qquad\square$

## C.4 Equivalence between wide DNN and linearized DNN

With the kernel convergences proved in Appendix C.3, we are now able to prove Theorem 2.

We will first prove the following Lemma C.12, and then prove Theorem 2.

**Lemma C.12.** *Suppose Assumptions 1, 2, 3, and 4 hold, and there exists $\tilde{n} \in \mathbb{N}^+$ such that $\{n_0, \cdots, n_L\} \geq \tilde{n}$. Then, as $\tilde{n} \to \infty$, we have that*

$$\lim_{n_L \to \infty} \cdots \lim_{n_0 \to \infty} \left\{ \sup_{t \in [0,T], s \in [0,S]} \|\partial_{f(\mathbf{x})}\mathcal{L}(f_t(\mathbf{x}_{t,s}), \mathbf{y}) - \partial_{f(\mathbf{x})}\mathcal{L}(f_t^{\mathrm{lin}}(\mathbf{x}_{t,s}^{\mathrm{lin}}), \mathbf{y})\|_2 \right\} \xrightarrow{P} 0,$$

$$\lim_{n_L \to \infty} \cdots \lim_{n_0 \to \infty} \left\{ \sup_{t \in [0,T], s \in [0,S]} \|f_t(\mathbf{x}_{t,s}) - f_t^{\mathrm{lin}}(\mathbf{x}_{t,s}^{\mathrm{lin}})\|_2 \right\} \xrightarrow{P} 0.$$

*Proof.* By Assumption 4, we have that

$$\|\partial_{f(\mathbf{x})}\mathcal{L}(f_t(\mathbf{x}_{t,s}), \mathbf{y}) - \partial_{f(\mathbf{x})}\mathcal{L}(f_t^{\mathrm{lin}}(\mathbf{x}_{t,s}^{\mathrm{lin}}), \mathbf{y})\|_2 \leq K \cdot \|f_t(\mathbf{x}_{t,s}) - f_t^{\mathrm{lin}}(\mathbf{x}_{t,s}^{\mathrm{lin}})\|_2.$$

Thus, to prove Lemma C.12, it is enough to only show the convergence of $\|f_t(\mathbf{x}_{t,s}) - f_t^{\mathrm{lin}}(\mathbf{x}_{t,s}^{\mathrm{lin}})\|_2$.

Then, following the AT dynamics formalized in Section 4.2, as $\tilde{n} \to \infty$, for any $t \in [0,T]$ and $s \in [0,S]$, we uniformly have

$$\|f_t(\mathbf{x}_{t,s}) - f_t^{\mathrm{lin}}(\mathbf{x}_{t,s}^{\mathrm{lin}})\|_2$$
$$= \left\| f_t(\mathbf{x}) - f_t^{\mathrm{lin}}(\mathbf{x}) + \int_0^s \left( \hat{\Theta}_{x,t}(\mathbf{x}_{t,\tau}, \mathbf{x}_{t,\tau}) \cdot \boldsymbol{\eta}(t) \cdot \partial_{f(\mathbf{x})}^T \mathcal{L}(f_t(\mathbf{x}_{t,\tau}), \mathbf{y}) - \hat{\Theta}_{x,0}(\mathbf{x}, \mathbf{x}) \cdot \boldsymbol{\eta}(t) \cdot \partial_{f(\mathbf{x})}^T \mathcal{L}(f_t^{\mathrm{lin}}(\mathbf{x}_{t,\tau}^{\mathrm{lin}}), \mathbf{y}) \right) \mathrm{d}\tau \right\|_2$$
$$\leq \|f_t(\mathbf{x}) - f_t^{\mathrm{lin}}(\mathbf{x})\|_2 + \int_0^S \|\hat{\Theta}_{x,t}(\mathbf{x}_{t,\tau}, \mathbf{x}_{t,\tau}) - \hat{\Theta}_{x,0}(\mathbf{x}, \mathbf{x})\|_2 \cdot \|\boldsymbol{\eta}(t)\|_2 \cdot \|\partial_{f(\mathbf{x})}\mathcal{L}(f_t(\mathbf{x}_{t,\tau}), \mathbf{y})\|_2 \mathrm{d}\tau$$
$$+ \int_0^s \|\hat{\Theta}_{x,0}(\mathbf{x}, \mathbf{x})\|_2 \cdot \|\boldsymbol{\eta}(t)\|_2 \cdot \|\partial_{f(\mathbf{x})}\mathcal{L}(f_t(\mathbf{x}_{t,\tau}), \mathbf{y}) - \partial_{f(\mathbf{x})}\mathcal{L}(f_t^{\mathrm{lin}}(\mathbf{x}_{t,\tau}^{\mathrm{lin}}), \mathbf{y})\|_2 \mathrm{d}\tau$$
$$\leq \|f_t(\mathbf{x}) - f_t^{\mathrm{lin}}(\mathbf{x})\|_2 + \sup_{\tau \in [0,S]} \|\hat{\Theta}_{x,t}(\mathbf{x}_{t,\tau}, \mathbf{x}_{t,\tau}) - \hat{\Theta}_{x,0}(\mathbf{x}, \mathbf{x})\|_2 \cdot \underbrace{O_p(1)}_{\text{Assumption 3}} \cdot \underbrace{O_p(1)}_{\text{Assumption 2}}$$
$$+ \|\hat{\Theta}_{x,0}(\mathbf{x}, \mathbf{x})\|_2 \cdot \underbrace{O_p(1)}_{\text{Assumption 3}} \cdot \int_0^s \underbrace{\|f_t(\mathbf{x}_{t,\tau}) - f_t^{\mathrm{lin}}(\mathbf{x}_{t,\tau}^{\mathrm{lin}})\|_2}_{\text{Assumption 4}} \mathrm{d}\tau.$$

By applying Grönwall's inequality (Lemma A.2), we further have

$$\|f_t(\mathbf{x}_{t,s}) - f_t^{\mathrm{lin}}(\mathbf{x}_{t,s}^{\mathrm{lin}})\|_2$$
$$\leq \left( \|f_t(\mathbf{x}) - f_t^{\mathrm{lin}}(\mathbf{x})\|_2 + \sup_{\tau \in [0,S]} \|\hat{\Theta}_{x,t}(\mathbf{x}_{t,\tau}, \mathbf{x}_{t,\tau}) - \hat{\Theta}_{x,0}(\mathbf{x}, \mathbf{x})\|_2 \cdot O_p(1) \right) \cdot \exp\left( \|\hat{\Theta}_{x,0}(\mathbf{x}, \mathbf{x})\|_2 \cdot O_p(1) \right),$$

$$\tag{C.19}$$

which indicates

$$\lim_{n_L \to \infty} \cdots \lim_{n_0 \to \infty} \sup_{t \in [0,T]} \|f_t(\mathbf{x}_{t,s}) - f_t^{\mathrm{lin}}(\mathbf{x}_{t,s}^{\mathrm{lin}})\|_2$$

$$\leq \lim_{n_L \to \infty} \cdots \lim_{n_0 \to \infty} \left( \sup_{t \in [0,T]} \|f_t(\mathbf{x}) - f_t^{\mathrm{lin}}(\mathbf{x})\|_2 + \sup_{t \in [0,T], \tau \in [0,S]} \|\hat{\Theta}_{x,t}(\mathbf{x}_{t,\tau}, \mathbf{x}_{t,\tau}) - \hat{\Theta}_{x,0}(\mathbf{x}, \mathbf{x})\|_2 \cdot O_p(1) \right)$$
$$\cdot \exp\left( \|\hat{\Theta}_{x,0}(\mathbf{x}, \mathbf{x})\|_2 \cdot O_p(1) \right)$$

$$\xrightarrow{P} \left( \lim_{n_L \to \infty} \cdots \lim_{n_0 \to \infty} \sup_{t \in [0,T]} \|f_t(\mathbf{x}) - f_t^{\mathrm{lin}}(\mathbf{x})\|_2 + \underbrace{O_p(\xi)}_{\text{Theorem C.1}} \right) \cdot \exp\left( \| \underbrace{\Theta_x(\mathbf{x}, \mathbf{x})\|_2}_{\text{Theorem B.1}} \cdot O_p(1) \right)$$

$$= \left( \lim_{n_L \to \infty} \cdots \lim_{n_0 \to \infty} \sup_{t \in [0,T]} \|f_t(\mathbf{x}) - f_t^{\mathrm{lin}}(\mathbf{x})\|_2 + O_p(\xi) \right) \cdot \exp(O_p(1)) \tag{C.20}$$

as $\tilde{n} \to \infty$, where $\xi$ denotes any term such that $\xi \to 0$ as $\tilde{n} \to \infty$.

For $\|f_t(\mathbf{x}) - f_t^{\mathrm{lin}}(\mathbf{x})\|_2$ in the above Eq. (C.20), we similarly have

$$\|f_t(\mathbf{x}) - f_t^{\mathrm{lin}}(\mathbf{x})\|_2 \leq \left\| \int_0^t \left( \hat{\Theta}_{\theta,\tau}(\mathbf{x}, \mathbf{x}_{\tau,S}) \cdot \partial_{f(\mathbf{x})}^T \mathcal{L}(f_\tau(\mathbf{x}_{\tau,S}), \mathbf{y}) - \hat{\Theta}_{\theta,0}(\mathbf{x}, \mathbf{x}) \cdot \partial_{f(\mathbf{x})}^T \mathcal{L}(f_\tau^{\mathrm{lin}}(\mathbf{x}_{\tau,S}), \mathbf{y}) \right) \mathrm{d}\tau \right\|_2$$

$$\leq \int_0^T \|\hat{\Theta}_{\theta,\tau}(\mathbf{x}, \mathbf{x}_{\tau,S}) - \hat{\Theta}_{\theta,0}(\mathbf{x}, \mathbf{x})\|_2 \cdot \|\partial_{f(\mathbf{x})} \mathcal{L}(f_\tau(\mathbf{x}_{\tau,S}), \mathbf{y})\|_2 \mathrm{d}\tau$$

$$+ \int_0^t \|\hat{\Theta}_{\theta,0}(\mathbf{x}, \mathbf{x})\|_2 \cdot \|\partial_{f(\mathbf{x})} \mathcal{L}(f_\tau(\mathbf{x}_{\tau,S}), \mathbf{y}) - \partial_{f(\mathbf{x})} \mathcal{L}(f_\tau^{\mathrm{lin}}(\mathbf{x}_{\tau,S}^{\mathrm{lin}}), \mathbf{y})\|_2 \mathrm{d}\tau$$

$$\leq \sup_{\tau \in [0,T]} \|\hat{\Theta}_{\theta,\tau}(\mathbf{x}, \mathbf{x}_{\tau,S}) - \hat{\Theta}_{\theta,0}(\mathbf{x}, \mathbf{x})\|_2 \cdot \underbrace{O_p(1)}_{\text{Assumption 2}} + \int_0^t \|\hat{\Theta}_{\theta,0}(\mathbf{x}, \mathbf{x})\|_2 \cdot \underbrace{\|f_\tau(\mathbf{x}_{\tau,S}) - f_\tau^{\mathrm{lin}}(\mathbf{x}_{\tau,S}^{\mathrm{lin}})\|_2}_{\text{Assumption 4}} \mathrm{d}\tau.$$

Adopting Eq. (C.19) into the above inequality and again using Grönwall's Lemma A.2 lead to

$$\|f_t(\mathbf{x}) - f_t^{\mathrm{lin}}(\mathbf{x})\|_2$$

$$\leq \left( \sup_{\tau \in [0,T]} \|\hat{\Theta}_{\theta,\tau}(\mathbf{x}, \mathbf{x}_{\tau,S}) - \hat{\Theta}_{\theta,0}(\mathbf{x}, \mathbf{x})\|_2 \right.$$

$$\left. + \|\hat{\Theta}_{\theta,0}(\mathbf{x}, \mathbf{x})\|_2 \cdot \sup_{\tau \in [0,S]} \|\hat{\Theta}_{x,t}(\mathbf{x}_{t,\tau}, \mathbf{x}_{t,\tau}) - \hat{\Theta}_{x,0}(\mathbf{x}, \mathbf{x})\|_2 \cdot \exp\left( \|\hat{\Theta}_{x,0}(\mathbf{x}, \mathbf{x})\|_2 \cdot O_p(1) \right) \right)$$

$$\cdot O_p(1) \cdot \exp\left( \|\hat{\Theta}_{\theta,0}(\mathbf{x}, \mathbf{x})\|_2 \cdot e^{T \cdot \|\hat{\Theta}_{x,0}(\mathbf{x}, \mathbf{x})\|_2 \cdot O_p(1)} \right),$$

which indicates

$$\lim_{n_L \to \infty} \cdots \lim_{n_0 \to \infty} \sup_{t \in [0,T]} \|f_t(\mathbf{x}) - f_t^{\mathrm{lin}}(\mathbf{x})\|_2$$

$$\leq \lim_{n_L \to \infty} \cdots \lim_{n_0 \to \infty} \left\{ \text{Upper Bound of } \sup_{t \in [0,T]} \|f_t(\mathbf{x}) - f_t^{\mathrm{lin}}(\mathbf{x})\|_2 \right\}$$

$$\xrightarrow{P} \left( \underbrace{O_p(\xi)}_{\text{Theorem C.1}} + \underbrace{\|\Theta_\theta(\mathbf{x}, \mathbf{x})\|_2}_{\text{Theorem B.1}} \cdot \underbrace{O_p(\xi)}_{\text{Theorem C.1}} \cdot \exp\left( \underbrace{\|\Theta_x(\mathbf{x}, \mathbf{x})\|_2}_{\text{Theorem B.1}} \cdot O_p(1) \right) \right)$$

$$\cdot O_p(1) \cdot \exp\left( \underbrace{\|\Theta_\theta(\mathbf{x}, \mathbf{x})\|_2 \cdot e^{\|\Theta_x(\mathbf{x}, \mathbf{x})\|_2 \cdot O_p(1)}}_{\text{Theorem B.1}} \right)$$

$$= \left( O_p(\xi) + O_p(1) \cdot O_p(\xi) \cdot e^{O_p(1)} \right) \cdot O_p(1) \cdot \exp\left( O_p(1) \cdot e^{O_p(1) \cdot O_p(1)} \right)$$

$$= O_p(\xi) \xrightarrow{P} 0.$$

as $\tilde{n} \to \infty$, where $\xi$ denotes any term such that $\xi \to 0$ as $\tilde{n} \to \infty$. Thus,

$$\lim_{n_L \to \infty} \cdots \lim_{n_0 \to \infty} \sup_{t \in [0,T]} \|f_t(\mathbf{x}) - f_t^{\mathrm{lin}}(\mathbf{x})\|_2 \xrightarrow{P} 0 \tag{C.21}$$

as $\tilde{n} \to \infty$.

Finally, inserting Eq. (C.21) into Eq. (C.20) and we have

$$\lim_{n_L \to \infty} \cdots \lim_{n_0 \to \infty} \sup_{t \in [0,T]} \|f_t(\mathbf{x}_{t,s}) - f_t^{\mathrm{lin}}(\mathbf{x}_{t,s}^{\mathrm{lin}})\|_2$$

$$\leq \lim_{n_L \to \infty} \cdots \lim_{n_0 \to \infty} \left\{ \text{Upper Bound of} \sup_{t \in [0,T], s \in [0,S]} \|f_t(\mathbf{x}_{t,s}) - f_t^{\mathrm{lin}}(\mathbf{x}_{t,s})\|_2 \right\}$$

$$\xrightarrow{P} (O_p(\xi) + O_p(\xi)) \cdot e^{O_p(1)}$$

$$\xrightarrow{P} (0 + 0) \cdot e^{O_p(1)} = 0,$$

which means as $\tilde{n} \to \infty$,

$$\lim_{n_L \to \infty} \cdots \lim_{n_0 \to \infty} \sup_{t \in [0,T], s \in [0,S]} \|f_t(\mathbf{x}_{t,s}) - f_t^{\mathrm{lin}}(\mathbf{x}_{t,s}^{\mathrm{lin}})\|_2 \xrightarrow{P} 0.$$

The proof is completed. □

Based on Lemma C.12, we now prove Theorem 2 as follows.

*Proof of **Theorem 2**.* For any $x \in \mathcal{X}$ as $\tilde{n} \to \infty$, we uniformly have that for any $t \in [0,T]$,

$$\|f_t(x) - f_t^{\mathrm{lin}}(x)\|_2$$

$$= \left\| \int_0^t \left( \hat{\Theta}_{\theta,\tau}(x, \mathbf{x}_{\tau,S}) \cdot \partial_{f(\mathbf{x})}^T \mathcal{L}(f_\tau(\mathbf{x}_{\tau,S}), \vec{y}) - \hat{\Theta}_{\theta,0}(x, \mathbf{x}) \cdot \partial_{f(\mathbf{x})}^T \mathcal{L}(f_\tau^{\mathrm{lin}}(\mathbf{x}_{\tau,S}^{\mathrm{lin}}), \mathbf{y}) \right) d\tau \right\|_2$$

$$\leq \int_0^T \|\hat{\Theta}_{\theta,\tau}(x, \mathbf{x}_{\tau,S}) - \hat{\Theta}_{\theta,0}(x, \mathbf{x})\|_2 \cdot \|\partial_{f(\mathbf{x})} \mathcal{L}(f_\tau(\mathbf{x}_{\tau,S}), \mathbf{y})\|_2 d\tau + \int_0^t \|\hat{\Theta}_{\theta,0}(x, \mathbf{x})\|_2 \cdot \|f_\tau(\mathbf{x}_{\tau,S}) - f_\tau^{\mathrm{lin}}(\mathbf{x}_{\tau,S}^{\mathrm{lin}})\|_2 d\tau$$

$$\leq \sup_{\tau \in [0,T]} \|\hat{\Theta}_{\theta,\tau}(x, \mathbf{x}_{\tau,S}) - \hat{\Theta}_{\theta,0}(x, \mathbf{x})\|_2 \cdot \underbrace{O_p(1)}_{\text{Assumption 2}} + \|\hat{\Theta}_{\theta,0}(x, \mathbf{x})\|_2 \cdot T \cdot \sup_{\tau \in [0,T]} \|f_\tau(\mathbf{x}_{\tau,S}) - f_\tau^{\mathrm{lin}}(\mathbf{x}_{\tau,S}^{\mathrm{lin}})\|_2.$$

Therefore, by applying Lemma C.12,

$$\lim_{n_L \to \infty} \cdots \lim_{n_0 \to \infty} \sup_{t \in [0,T]} \|f_t(x) - f_t^{\mathrm{lin}}(x)\|_2$$

$$\leq \lim_{n_L \to \infty} \cdots \lim_{n_0 \to \infty} \left\{ \text{Upper Bound of} \sup_{t \in [0,T]} \|f_t(x) - f_t^{\mathrm{lin}}(x)\|_2 \right\}$$

$$\xrightarrow{P} \underbrace{O_p(\xi)}_{\text{Theorem C.2}} \cdot O_p(1) + \underbrace{\|\Theta_\theta(x, \mathbf{x})\|_2}_{\text{Theorem B.1}} \cdot T \cdot \underbrace{O_p(\xi)}_{\text{Lemma C.12}}$$

$$= O_p(\xi) \cdot O_p(1) + \cdot O_p(1) \cdot O_p(\xi) = O_p(\xi) \xrightarrow{P} 0,$$

where $\xi$ denotes any term such that $\xi \to 0$ as $\tilde{n} \to \infty$. This means that as $\tilde{n} \to \infty$,

$$\lim_{n_L \to \infty} \cdots \lim_{n_0 \to \infty} \|f_t(x) - f_t^{\mathrm{lin}}(x)\|_2 \xrightarrow{P} 0.$$

The proof is completed. □

## D  PROOF OF THEOREM 3

This section presents the proof of Theorem 3.

### D.1 PROOF SKELETON

To calculate the closed-form solution of the linearized DNN $f_t^{\text{lin}}$ with squared loss in AT, the main technical challenge is that the learning rate matrix $\boldsymbol{\eta}(t)$ is non-linear on $[0, T]$, which results in $f_t^{\text{lin}}$ also being non-linear on $[0, T]$ and thus is difficult to calculate a closed-form solution for it. To tackle the challenge, we propose to first use a surrogate piecewise linear function $f_t^{\text{sur}}$ to approximate $f_t^{\text{lin}}$ and then directly calculate the closed-form solution of $f_t^{\text{sur}}$ when the squared loss is used.

Specifically, for a given $t \in [0, T]$, the piecewise linear function $f_v^{\text{sur}}$ on the interval $[0, t]$ (note that $v \in [0, t]$) is constructed in the following three steps:

1. Choose a $D \in \mathbb{N}^+$, and divide the interval $[0, t]$ into $D$ equal-width sub-intervals, where the $d$-th ($d \in [D]$) interval is $[a_{d-1}, a_d]$ and $a_d := \frac{td}{D}$.

2. Construct a surrogate learning rate matrix $\boldsymbol{\eta}^{\text{sur}}(v)$ such that $\boldsymbol{\eta}^{\text{sur}}(v) := \boldsymbol{\eta}(a_{d-1})$ when $v \in [a_{d-1}, a_d)$ for every $d \in [D]$.

3. Initialize $f_v^{\text{sur}}$ with parameter $\theta_0$ and train it following the same AT dynamics as that of $f_v^{\text{lin}}$ except using the $\boldsymbol{\eta}^{\text{sur}}(v)$ as the learning rate matrix for searching adversarial examples.

After finishing the training on $[0, t]$, we will obtained the eventual surrogate DNN $f_t^{\text{sur}}$.

In the rest of this section, we will show that: (1) in Appendix D.2, the designed surrogate $f_t^{\text{sur}}$ can well approximate the original $f_t^{\text{lin}}$ as $D \to \infty$, and (2) in Appendix D.3, the closed-form solution of the surrogate $f_t^{\text{sur}}$ can be easily calculated, which then leads to the closed-form solution of $f_t^{\text{lin}}$.

### D.2 APPROXIMATING LINEARIZED DNN WITH PIECEWISE LINEAR MODEL

**Lemma D.1.** *Suppose Assumptions 3 and 4 hold. For any $t \in [0, T]$, we construct $f_v^{\text{sur}}$ on $[0, t]$ as that introduced in Appendix D. Then, as $D \to \infty$, for any $v \in [0, t]$ and $s \in [0, S]$, we uniformly have that*

$$\|f_v^{\text{lin}}(\mathbf{x}_{v,s}^{\text{lin}}) - f_v^{\text{sur}}(\mathbf{x}_{v,s}^{\text{sur}})\|_2 \to 0.$$

*Proof.* We first upper bound the difference between $f_v^{\text{lin}}(\mathbf{x}_{v,s}^{\text{lin}})$ and $f_v^{\text{sur}}(\mathbf{x}_{v,s}^{\text{sur}})$ on the $d$-th interval. Specifically, for any $v \in [a_{d-1}, a_d]$ and $s \in [0, S]$, we uniformly have that

$$\|f_v^{\text{sur}}(\mathbf{x}_{v,s}^{\text{sur}}) - f_v^{\text{lin}}(\mathbf{x}_{v,s}^{\text{lin}})\|_2$$

$$\leq \|f_v^{\text{sur}}(\mathbf{x}) - f_v^{\text{lin}}(\mathbf{x})\|_2 + \int_0^s \underbrace{\|\hat{\Theta}_{x,0}(\mathbf{x}, \mathbf{x})\|_2}_{\text{Constant}} \cdot \|\boldsymbol{\eta}^{\text{sur}}(v) \cdot \partial_{f(\mathbf{x})}^T \mathcal{L}(f_v^{\text{sur}}(\mathbf{x}_{v,\tau}^{\text{sur}}), \mathbf{y}) - \boldsymbol{\eta}(v) \cdot \partial_{f(\mathbf{x})}^T \mathcal{L}(f_v^{\text{lin}}(\mathbf{x}_{v,\tau}^{\text{lin}}), \mathbf{y})\|_2 d\tau$$

$$\leq \|f_v^{\text{sur}}(\mathbf{x}) - f_v^{\text{lin}}(\mathbf{x})\|_2$$

$$\quad + \int_0^s C \cdot \|\boldsymbol{\eta}^{\text{sur}}(v)\|_2 \cdot \|\partial_{f(\mathbf{x})} \mathcal{L}(f_v^{\text{sur}}(\mathbf{x}_{v,\tau}^{\text{sur}}), \mathbf{y}) - \partial_{f(\mathbf{x})} \mathcal{L}(f_v^{\text{lin}}(\mathbf{x}_{v,\tau}^{\text{lin}}), \mathbf{y})\|_2 d\tau$$

$$\quad + \int_0^s C \cdot \|\boldsymbol{\eta}^{\text{sur}}(v) - \boldsymbol{\eta}(v)\|_2 \cdot \|\partial_{f(\mathbf{x})} \mathcal{L}(f_v^{\text{lin}}(\mathbf{x}_{v,\tau}^{\text{lin}}), \mathbf{y})\|_2 d\tau$$

$$\leq \|f_v^{\text{sur}}(\mathbf{x}) - f_v^{\text{lin}}(\mathbf{x})\|_2$$

$$\quad + C \cdot \underbrace{\sup_{v \in [0, T]} \|\boldsymbol{\eta}^{\text{sur}}(v)\|_2}_{\text{Constant by Assumption 3}} \cdot \int_0^s \|\partial_{f(\mathbf{x})} \mathcal{L}(f_v^{\text{sur}}(\mathbf{x}_{v,\tau}^{\text{sur}}), \mathbf{y}) - \partial_{f(\mathbf{x})} \mathcal{L}(f_v^{\text{lin}}(\mathbf{x}_{v,\tau}^{\text{lin}}), \mathbf{y})\|_2 d\tau$$

$$\quad + C \cdot \sup_{v \in [a_{d-1}, a_d]} \|\boldsymbol{\eta}(a_{d-1}) - \boldsymbol{\eta}(v)\|_2 \cdot \underbrace{\int_0^s \|\partial_{f(\mathbf{x})} \mathcal{L}(f_v^{\text{lin}}(\mathbf{x}_{v,\tau}^{\text{lin}}), \mathbf{y})\|_2 d\tau}_{\text{Constant}}$$

$$\leq \|f_v^{\text{sur}}(\mathbf{x}) - f_v^{\text{lin}}(\mathbf{x})\|_2 + C \cdot \int_0^s \underbrace{K \cdot \|f_v^{\text{sur}}(\mathbf{x}_{v,\tau}^{\text{sur}}) - f_v^{\text{lin}}(\mathbf{x}_{v,\tau}^{\text{lin}})\|_2}_{\text{Assumption 4}} d\tau + C \cdot \sup_{v_1, v_2 \in [0, T],\ |v_1 - v_2| \leq \frac{t}{D}} \|\boldsymbol{\eta}(v_1) - \boldsymbol{\eta}(v_2)\|_2,$$

where $C$ denotes any non-negative constant. Therefore, by Grönwall's Lemma A.2, we uniformly have that for any $v \in [a_{d-1}, a_d]$ and $s \in [0, S]$,

$$
\begin{aligned}
&\|f_v^{\mathrm{sur}}(\mathbf{x}_{v,s}^{\mathrm{sur}}) - f_v^{\mathrm{lin}}(\mathbf{x}_{v,s}^{\mathrm{lin}})\|_2 \\
&\leq \left( \|f_v^{\mathrm{sur}}(\mathbf{x}) - f_v^{\mathrm{lin}}(\mathbf{x})\|_2 + C \cdot \sup_{v_1, v_2 \in [0,T],\ |v_1 - v_2| \leq \frac{t}{D}} \|\boldsymbol{\eta}(v_1) - \boldsymbol{\eta}(v_2)\|_2 \right) \cdot \exp(C \cdot S \cdot K) \\
&\leq C \cdot \|f_v^{\mathrm{sur}}(\mathbf{x}) - f_v^{\mathrm{lin}}(\mathbf{x})\|_2 + C \cdot \sup_{v_1, v_2 \in [0,T],\ |v_1 - v_2| \leq \frac{t}{D}} \|\boldsymbol{\eta}(v_1) - \boldsymbol{\eta}(v_2)\|_2, \quad\quad\quad \text{(D.1)}
\end{aligned}
$$

where $C$ denotes any non-negative constant.

For $\|f_v^{\mathrm{sur}}(\mathbf{x}_{v,s}^{\mathrm{sur}}) - f_v^{\mathrm{lin}}(\mathbf{x}_{v,s}^{\mathrm{lin}})\|_2$ in Eq. (D.1), we bound it on the interval $[a_{d-1}, a_d]$ as follows,

$$
\begin{aligned}
&\|f_v^{\mathrm{sur}}(\mathbf{x}) - f_v^{\mathrm{lin}}(\mathbf{x})\|_2 \\
&\leq \|f_{a_{d-1}}^{\mathrm{sur}}(\mathbf{x}) - f_{a_{d-1}}^{\mathrm{lin}}(\mathbf{x})\|_2 + \int_{a_{d-1}}^{v} \underbrace{\|\hat{\Theta}_{\theta,0}(\mathbf{x},\mathbf{x})\|_2}_{\text{Constant}} \cdot \|\partial_{f(\mathbf{x})}\mathcal{L}(f_\tau^{\mathrm{sur}}(\mathbf{x}_{\tau,S}^{\mathrm{sur}}), \mathbf{y}) - \partial_{f(\mathbf{x})}\mathcal{L}(f_\tau^{\mathrm{lin}}(\mathbf{x}_{\tau,S}^{\mathrm{lin}}), \mathbf{y})\|_2 \mathrm{d}\tau \\
&\leq \|f_{a_{d-1}}^{\mathrm{sur}}(\mathbf{x}) - f_{a_{d-1}}^{\mathrm{lin}}(\mathbf{x})\|_2 + C \cdot \int_{a_{d-1}}^{v} \underbrace{K \cdot \|f_\tau^{\mathrm{sur}}(\mathbf{x}_{\tau,S}^{\mathrm{sur}}) - f_\tau^{\mathrm{lin}}(\mathbf{x}_{\tau,S}^{\mathrm{lin}})\|_2}_{\text{Assumption 4}} \mathrm{d}\tau \\
&\leq \|f_{a_{d-1}}^{\mathrm{sur}}(\mathbf{x}) - f_{a_{d-1}}^{\mathrm{lin}}(\mathbf{x})\|_2 + C \cdot \int_{a_{d-1}}^{v} \underbrace{\left( C \cdot \|f_\tau^{\mathrm{sur}}(\mathbf{x}) - f_\tau^{\mathrm{lin}}(\mathbf{x})\|_2 + C \cdot \sup_{v_1, v_2 \in [0,T],\ |v_1 - v_2| \leq \frac{t}{D}} \|\boldsymbol{\eta}(v_1) - \boldsymbol{\eta}(v_2)\|_2 \right)}_{\text{Eq. (D.1)}} \mathrm{d}\tau \\
&\leq \|f_{a_{d-1}}^{\mathrm{sur}}(\mathbf{x}) - f_{a_{d-1}}^{\mathrm{lin}}(\mathbf{x})\|_2 + \frac{Ct}{D} \cdot \sup_{v_1, v_2 \in [0,T],\ |v_1 - v_2| \leq \frac{t}{D}} \|\boldsymbol{\eta}(v_1) - \boldsymbol{\eta}(v_2)\|_2 + C \cdot \int_{a_{d-1}}^{v} \|f_\tau^{\mathrm{sur}}(\mathbf{x}) - f_\tau^{\mathrm{lin}}(\mathbf{x})\|_2 \mathrm{d}\tau,
\end{aligned}
$$

where $C$ denotes any non-negative constant. As a result, by again using Grönwall's Lemma A.2, we uniformly have that for any $v \in [a_{d-1}, a_d]$,

$$
\begin{aligned}
&\|f_v^{\mathrm{sur}}(\mathbf{x}) - f_v^{\mathrm{lin}}(\mathbf{x})\|_2 \\
&\leq \left( \|f_{a_{d-1}}^{\mathrm{sur}}(\mathbf{x}) - f_{a_{d-1}}^{\mathrm{lin}}(\mathbf{x})\|_2 + \frac{Ct}{D} \cdot \sup_{v_1, v_2 \in [0,T],\ |v_1 - v_2| \leq \frac{t}{D}} \|\boldsymbol{\eta}(v_1) - \boldsymbol{\eta}(v_2)\|_2 \right) \cdot \exp\left( C \cdot \int_{a_{d-1}}^{v} \mathrm{d}\tau \right) \\
&\leq \left( \|f_{a_{d-1}}^{\mathrm{sur}}(\mathbf{x}) - f_{a_{d-1}}^{\mathrm{lin}}(\mathbf{x})\|_2 + \frac{C}{D} \cdot \sup_{v_1, v_2 \in [0,T],\ |v_1 - v_2| \leq \frac{t}{D}} \|\boldsymbol{\eta}(v_1) - \boldsymbol{\eta}(v_2)\|_2 \right) \cdot \exp\left( \frac{C}{D} \right), \\
&\quad\quad\quad\quad\quad\quad\quad\quad\quad\quad\quad\quad\quad\quad\quad\quad\quad\quad\quad\quad\quad\quad\quad\quad\quad\quad\quad\quad\quad\quad\quad\quad\quad\quad\quad\quad \text{(D.2)}
\end{aligned}
$$

where $C$ denotes any non-negative constant.

Therefore, for any $a_d$ where $d \in [D]$, by recursively applying Eq. (D.2), we will have that

$$
\begin{aligned}
&\|f_{a_d}^{\mathrm{sur}}(\mathbf{x}) - f_{a_d}^{\mathrm{lin}}(\mathbf{x})\|_2 \\
&\leq \left( \|f_{a_{d-1}}^{\mathrm{sur}}(\mathbf{x}) - f_{a_{d-1}}^{\mathrm{lin}}(\mathbf{x})\|_2 + \frac{C}{D} \cdot \sup_{v_1, v_2 \in [0,T],\ |v_1 - v_2| \leq \frac{t}{D}} \|\boldsymbol{\eta}(v_1) - \boldsymbol{\eta}(v_2)\|_2 \right) \cdot \exp\left( \frac{C}{D} \right) \\
&\leq \cdots \leq \|f_0^{\mathrm{sur}}(\mathbf{x}) - f_0^{\mathrm{lin}}(\mathbf{x})\|_2 \cdot \exp\left( \frac{Cd}{D} \right) + \sum_{d'=1}^{d} \left( \frac{C}{D} \cdot \sup_{v_1, v_2 \in [0,T],\ |v_1 - v_2| \leq \frac{t}{D}} \|\boldsymbol{\eta}(v_1) - \boldsymbol{\eta}(v_2)\|_2 \right) \cdot \exp\left( \frac{Cd'}{D} \right) \\
&\leq 0 \cdot \exp(C) + \frac{Cd}{D} \cdot \sup_{v_1, v_2 \in [0,T],\ |v_1 - v_2| \leq \frac{t}{D}} \|\boldsymbol{\eta}(v_1) - \boldsymbol{\eta}(v_2)\|_2 \cdot \exp(C) \\
&\leq C \cdot \sup_{v_1, v_2 \in [0,T],\ |v_1 - v_2| \leq \frac{t}{D}} \|\boldsymbol{\eta}(v_1) - \boldsymbol{\eta}(v_2)\|_2, \quad\quad\quad\quad\quad\quad\quad\quad\quad\quad\quad\quad \text{(D.3)}
\end{aligned}
$$

where $C$ denotes any non-negative constant.

Combining Eqs. (D.1), (D.2), (D.3) leads to

$$\|f_v^{\mathrm{sur}}(\mathbf{x}_{v,s}^{\mathrm{sur}}) - f_v^{\mathrm{lin}}(\mathbf{x}_{v,s}^{\mathrm{lin}})\|_2$$

$$\le C \cdot \left( \exp\left( \frac{C}{D} \right) \cdot \left( C + \frac{C}{D} \right) + 1 \right) \cdot \sup_{v_1,v_2 \in [0,T],\ |v_1-v_2| \le \frac{t}{D}} \|\boldsymbol{\eta}(v_1) - \boldsymbol{\eta}(v_2)\|_2$$

$$\le C \cdot \sup_{v_1,v_2 \in [0,T],\ |v_1-v_2| \le \frac{t}{D}} \|\boldsymbol{\eta}(v_1) - \boldsymbol{\eta}(v_2)\|_2,$$

where $C$ denotes any non-negative constant.

Finally, according to Assumption 3, we know that $\boldsymbol{\eta}(t)$ is continuous on the closed interval $[0,T]$, which indicates it is also uniformly continuous on $[0,T]$. Therefore, as $D \to \infty$, we will have $\frac{t}{D} \to \infty$, which indicates for any $v \in [0,t]$ and $s \in [0,S]$ uniformly,

$$\sup_{v_1,v_2 \in [0,T],\ |v_1-v_2| \le \frac{t}{D}} \|\boldsymbol{\eta}(v_1) - \boldsymbol{\eta}(v_2)\|_2 \to 0.$$

Combining the above results and we finally get

$$\|f_v^{\mathrm{sur}}(\mathbf{x}_{v,s}^{\mathrm{sur}}) - f_v^{\mathrm{lin}}(\mathbf{x}_{v,s}^{\mathrm{lin}})\|_2 \to 0$$

uniformly for any $v \in [0,t]$ and $s \in [0,S]$ as $D \to \infty$.

The proof is completed. $\qquad\square$

**Lemma D.2.** *Suppose Assumptions 3 and 4 hold. For any $t \in [0,T]$, we construct $f_v^{\mathrm{sur}}$ on $[0,t]$ as that introduced in Appendix D. Then, for any $x \in [0,x]$, as $D \to \infty$,*

$$\|f_t^{\mathrm{lin}}(x) - f_t^{\mathrm{sur}}(x)\|_2 \to 0.$$

*Proof.* The difference between $f_t^{\mathrm{sur}}(x)$ and $f_t^{\mathrm{lin}}(x)$ can be bounded as follows,

$$\|f_t^{\mathrm{sur}}(x) - f_t^{\mathrm{lin}}(x)\|_2$$

$$\le \|f_0^{\mathrm{sur}}(x) - f_0^{\mathrm{lin}}(x)\|_2 + \int_0^t \underbrace{\|\hat{\Theta}_{\theta,0}(x,\mathbf{x})\|_2}_{\text{Constant}} \cdot \|\partial_{f(\mathbf{x})} \mathcal{L}(f_v^{\mathrm{sur}}(\mathbf{x}_{v,S}), \mathbf{y}) - \partial_{f(\mathbf{x})} \mathcal{L}(f_v^{\mathrm{lin}}(\mathbf{x})_{v,S}, \mathbf{y})\|_2 \mathrm{d}v$$

$$\le 0 + C \cdot \int_0^t \underbrace{K \cdot \|f_v^{\mathrm{sur}}(\mathbf{x}) - \partial_{f(\mathbf{x})} f_v^{\mathrm{lin}}(\mathbf{x})\|_2}_{\text{Assumption 4}} \mathrm{d}v$$

$$\le C \cdot T \cdot \sup_{v \in [0,t]} \|f_v^{\mathrm{sur}}(\mathbf{x}) - \partial_{f(\mathbf{x})} f_v^{\mathrm{lin}}(\mathbf{x})\|_2 = C \cdot \sup_{v \in [0,t]} \|f_v^{\mathrm{sur}}(\mathbf{x}) - \partial_{f(\mathbf{x})} f_v^{\mathrm{lin}}(\mathbf{x})\|_2,$$

where $C$ denotes any non-negative constant. Then, by applying Lemma D.1, we have that

$$\sup_{v \in [0,t]} \|f_v^{\mathrm{sur}}(\mathbf{x}) - \partial_{f(\mathbf{x})} f_v^{\mathrm{lin}}(\mathbf{x})\|_2 \to 0$$

as $D \to \infty$, which thus leads to

$$\|f_t^{\mathrm{sur}}(x) - f_t^{\mathrm{lin}}(x)\|_2 \to 0.$$

The proof is completed. $\qquad\square$

## D.3 CLOSED-FORM SOLUTIONS OF LINEARIZED AND PIECEWISE LINEAR DNNs

**Lemma D.3.** *Suppose Assumption 3 holds and the loss function used in AT is squared loss $\mathcal{L}(f(x),y) := \frac{1}{2}\|f(x) - y\|_2^2$. For any $t \in [0,T]$, we construct $f_v^{\mathrm{sur}}$ on $[0,t]$ as that introduced in Appendix D. Then, for any $x \in [0,x]$, we have*

$$\lim_{D \to \infty} f_t^{\mathrm{sur}}(x) = f_0(x) - \hat{\Theta}_{\theta,0}(x,\mathbf{x}) \cdot \hat{\Theta}_{\theta,0}^{-1}(\mathbf{x},\mathbf{x}) \cdot \left( I - e^{-\hat{\Theta}_{\theta,0}(\mathbf{x},\mathbf{x}) \cdot \hat{\Xi}(t)} \right) \cdot (f_0(\mathbf{x}) - \mathbf{y}),$$

*where $\hat{\Xi}(t) := \int_0^t \exp\left( \hat{\Theta}_{x,0}(\mathbf{x},\mathbf{x}) \cdot \boldsymbol{\eta}(v) \cdot S \right) \mathrm{d}v.$*

*Proof.* We first calculate the closed-form solution of $f_v^{\text{sur}}$ on each sub-interval. Specifically, for the $d$-th sub-interval $[a_{d-1}, a_d]$ and any $v \in [a_{d-1}, a_d]$, the dynamics of searching adversarial examples can be formalized following Eqs. (16) and (17) in Section 4.2 as below,

$$\partial_s f_v^{\text{sur}}(\mathbf{x}_{v,s}^{\text{sur}}) = \hat{\Theta}_{x,0}(\mathbf{x}, \mathbf{x}) \cdot \boldsymbol{\eta}^{\text{sur}}(v) \cdot (f_v^{\text{sur}}(\mathbf{x}_{v,s}^{\text{sur}}) - \mathbf{y})$$
$$= \hat{\Theta}_{x,0}(\mathbf{x}, \mathbf{x}) \cdot \boldsymbol{\eta}(a_{d-1}) \cdot (f_v^{\text{sur}}(\mathbf{x}_{v,s}^{\text{sur}}) - \mathbf{y}),$$

where

$$f_v^{\text{sur}}(\mathbf{x}_{v,0}^{\text{sur}}) = f_v^{\text{sur}}(\mathbf{x}).$$

Solving the above ordinary equation and we have

$$(f_v^{\text{sur}}(\mathbf{x}_{v,s}^{\text{sur}}) - \mathbf{y}) = \exp\left(\hat{\Theta}_{x,0}(\mathbf{x}, \mathbf{x}) \cdot \boldsymbol{\eta}(a_{d-1}) \cdot s\right) \cdot (f_v^{\text{sur}}(\mathbf{x}) - \mathbf{y}). \tag{D.4}$$

Then, for the AT dynamics of $f_v^{\text{sur}}$ on $[a_{d-1}, a_d]$, it can be formalized following Eqs. (14) and (15) in Section 4.2 as below,

$$\partial_v \theta_v^{\text{sur}} = -\partial_\theta^T f_0(\mathbf{x}) \cdot (f_v^{\text{sur}}(\mathbf{x}_{v,S}^{\text{sur}}) - \mathbf{y}),$$
$$\partial_v f_v^{\text{sur}}(\mathbf{x}) = -\hat{\Theta}_{\theta,0}(\mathbf{x}, \mathbf{x}) \cdot (f_v^{\text{sur}}(\mathbf{x}_{v,S}^{\text{sur}}) - \mathbf{y}).$$

Inserting Eq. (D.4) into the above ordinary equations, we further have

$$\partial_v \theta_v^{\text{sur}} = -\partial_\theta^T f_0(\mathbf{x}) \cdot \exp\left(\hat{\Theta}_{x,0}(\mathbf{x}, \mathbf{x}) \cdot \boldsymbol{\eta}(a_{d-1}) \cdot S\right) \cdot (f_v^{\text{sur}}(\mathbf{x}) - \mathbf{y}), \tag{D.5}$$

$$\partial_v f_v^{\text{sur}}(\mathbf{x}) = -\hat{\Theta}_{\theta,0}(\mathbf{x}, \mathbf{x}) \cdot \exp\left(\hat{\Theta}_{x,0}(\mathbf{x}, \mathbf{x}) \cdot \boldsymbol{\eta}(a_{d-1}) \cdot S\right) \cdot (f_v^{\text{sur}}(\mathbf{x}) - \mathbf{y}). \tag{D.6}$$

Solving Eq. (D.6) and we have for any $v \in [a_{d-1}, a_d]$,

$$(f_v^{\text{sur}}(\mathbf{x}) - \mathbf{y}) = \exp\left(-\hat{\Theta}_{\theta,0}(\mathbf{x}, \mathbf{x}) \cdot e^{\hat{\Theta}_{x,0}(\mathbf{x},\mathbf{x}) \cdot \boldsymbol{\eta}(a_{d-1}) \cdot S} \cdot (v - a_{d-1})\right) \cdot (f_{a_{d-1}}^{\text{sur}}(\mathbf{x}) - \mathbf{y}), \tag{D.7}$$

which indicates

$$(f_t^{\text{sur}}(\mathbf{x}) - \mathbf{y}) = (f_{a_D}^{\text{sur}}(\mathbf{x}) - \mathbf{y})$$
$$= \left(\prod_{d=D}^{1} \exp\left(-\hat{\Theta}_{\theta,0}(\mathbf{x}, \mathbf{x}) \cdot e^{\hat{\Theta}_{x,0}(\mathbf{x},\mathbf{x}) \cdot \boldsymbol{\eta}(a_{d-1}) \cdot S} \cdot (a_d - a_{d-1})\right)\right) \cdot (f_{a_0}^{\text{sur}}(\mathbf{x}) - \mathbf{y})$$
$$= \exp\left(-\hat{\Theta}_{\theta,0}(\mathbf{x}, \mathbf{x}) \cdot \frac{t}{D} \sum_{d=1}^{D} e^{\hat{\Theta}_{x,0}(\mathbf{x},\mathbf{x}) \cdot \boldsymbol{\eta}(a_{d-1}) \cdot S}\right) \cdot (f_0(\mathbf{x}) - \mathbf{y}). \tag{D.8}$$

Besides, by combining Eqs. (D.5) and (D.7), the closed-form solution of the difference between $\theta_{a_d}^{\text{sur}}$ and $\theta_{a_{d-1}}^{\text{sur}}$ is calculated as follows,

$$\theta_{a_d}^{\text{sur}} - \theta_{a_{d-1}}^{\text{sur}}$$
$$= \int_{a_{d-1}}^{a_d} \underbrace{-\partial_\theta^T f_0(\mathbf{x}) \cdot \exp\left(\hat{\Theta}_{x,0}(\mathbf{x}, \mathbf{x}) \cdot \boldsymbol{\eta}(a_{d-1}) \cdot S\right) \cdot (f_v^{\text{sur}}(\mathbf{x}) - \mathbf{y})}_{\text{Eq. (D.5)}} \, dv$$
$$= \int_{a_{d-1}}^{a_d} -\partial_\theta^T f_0(\mathbf{x}) \cdot e^{\hat{\Theta}_{x,0}(\mathbf{x},\mathbf{x}) \cdot \boldsymbol{\eta}(a_{d-1}) \cdot S} \cdot \underbrace{\exp\left(-\hat{\Theta}_{\theta,0}(\mathbf{x}, \mathbf{x}) \cdot e^{\hat{\Theta}_{x,0}(\mathbf{x},\mathbf{x}) \cdot \boldsymbol{\eta}(a_{d-1}) \cdot S} \cdot (v - a_{d-1})\right) \cdot (f_{a_{d-1}}^{\text{sur}}(\mathbf{x}) - \mathbf{y})}_{\text{Eq. (D.7)}} \, dv$$
$$= \left[\partial_\theta^T f_0(\mathbf{x}) \cdot \hat{\Theta}_{\theta,0}^{-1}(\mathbf{x}, \mathbf{x}) \cdot \exp\left(-\hat{\Theta}_{\theta,0}(\mathbf{x}, \mathbf{x}) \cdot e^{\hat{\Theta}_{x,0}(\mathbf{x},\mathbf{x}) \cdot \boldsymbol{\eta}(a_{d-1}) \cdot S} \cdot (v - a_{d-1})\right) \cdot (f_{a_{d-1}}^{\text{sur}}(\mathbf{x}) - \mathbf{y})\right]_{a_{d-1}}^{a_d}$$
$$= \partial_\theta^T f_0(\mathbf{x}) \cdot \hat{\Theta}_{\theta,0}^{-1}(\mathbf{x}, \mathbf{x}) \cdot \left(\exp\left(-\hat{\Theta}_{\theta,0}(\mathbf{x}, \mathbf{x}) \cdot e^{\hat{\Theta}_{x,0}(\mathbf{x},\mathbf{x}) \cdot \boldsymbol{\eta}(a_{d-1}) \cdot S} \cdot \frac{t}{D}\right) - I\right) \cdot (f_{a_{d-1}}^{\text{sur}}(\mathbf{x}) - \mathbf{y})$$
$$= \partial_\theta^T f_0(\mathbf{x}) \cdot \hat{\Theta}_{\theta,0}^{-1}(\mathbf{x}, \mathbf{x}) \cdot \underbrace{(f_{a_d}^{\text{sur}}(\mathbf{x}) - f_{a_{d-1}}^{\text{sur}}(\mathbf{x}))}_{\text{Eq. (D.7)}}.$$

The above equation illustrates that the model parameter $\theta_t^{\mathrm{sur}}$ of $f_v^{\mathrm{sur}}$ at the eventual training time $t$ can be calculated as below,

$$\theta_t^{\mathrm{sur}} - \theta_0 = \theta_{a_D}^{\mathrm{sur}} - \theta_{a_0}^{\mathrm{sur}} = \sum_{d=1}^D (\theta_{a_d}^{\mathrm{sur}} - \theta_{a_{d-1}}^{\mathrm{sur}})$$

$$= \sum_{d=1}^D \partial_\theta^T f_0(\mathbf{x}) \cdot \hat{\Theta}_{\theta,0}^{-1}(\mathbf{x}, \mathbf{x}) \cdot (f_{a_d}^{\mathrm{sur}}(\mathbf{x}) - f_{a_{d-1}}^{\mathrm{sur}}(\mathbf{x}))$$

$$= \partial_\theta^T f_0(\mathbf{x}) \cdot \hat{\Theta}_{\theta,0}^{-1}(\mathbf{x}, \mathbf{x}) \cdot (f_{a_D}^{\mathrm{sur}}(\mathbf{x}) - f_0(\mathbf{x}))$$

$$= \partial_\theta^T f_0(\mathbf{x}) \cdot \hat{\Theta}_{\theta,0}^{-1}(\mathbf{x}, \mathbf{x}) \cdot \left( \exp\left( -\hat{\Theta}_{\theta,0}(\mathbf{x}, \mathbf{x}) \cdot \frac{t}{D} \sum_{d=1}^D e^{\hat{\Theta}_{x,0}(\mathbf{x},\mathbf{x}) \cdot \boldsymbol{\eta}(a_{d-1}) \cdot S} \right) - I \right) \cdot (f_0(\mathbf{x}) - \mathbf{y}).$$

As a result, for any $x \in \mathcal{X}$,

$$\lim_{D\to\infty} f_t^{\mathrm{sur}}(x) = \lim_{D\to\infty} (f_0(x) + \partial_\theta f_0(x) \cdot (\theta_t^{\mathrm{sur}} - \theta_0))$$

$$= f_0(x) + \lim_{D\to\infty} \partial_\theta f_0(x) \cdot \partial_\theta^T f_0(\mathbf{x}) \cdot \hat{\Theta}_{\theta,0}^{-1}(\mathbf{x}, \mathbf{x}) \cdot \left( \exp\left( -\hat{\Theta}_{\theta,0}(\mathbf{x}, \mathbf{x}) \cdot \frac{t}{D} \sum_{d=1}^D e^{\hat{\Theta}_{x,0}(\mathbf{x},\mathbf{x}) \cdot \boldsymbol{\eta}(a_{d-1}) \cdot S} \right) - I \right) \cdot (f_0(\mathbf{x}) - \mathbf{y})$$

$$= f_0(x) + \hat{\Theta}_{\theta,0}(x, \mathbf{x}) \cdot \hat{\Theta}_{\theta,0}^{-1}(\mathbf{x}, \mathbf{x}) \cdot \left( \exp\left( -\hat{\Theta}_{\theta,0}(\mathbf{x}, \mathbf{x}) \cdot \lim_{D\to\infty} \left\{ \frac{t}{D} \sum_{d=1}^D e^{\hat{\Theta}_{x,0}(\mathbf{x},\mathbf{x}) \cdot \boldsymbol{\eta}(a_{d-1}) \cdot S} \right\} \right) - I \right) \cdot (f_0(\mathbf{x}) - \mathbf{y})$$

By the Darboux integral, when $D \to \infty$, we have

$$\lim_{D\to\infty} \left\{ \frac{t}{D} \sum_{d=1}^D e^{\hat{\Theta}_{x,0}(\mathbf{x},\mathbf{x}) \cdot \boldsymbol{\eta}(a_{d-1}) \cdot S} \right\} = \int_0^t \exp\left( \hat{\Theta}_{x,0}(\mathbf{x}, \mathbf{x}) \cdot \boldsymbol{\eta}(v) \cdot S \right) \mathrm{d}v,$$

which means

$$\lim_{D\to\infty} f_t^{\mathrm{sur}}(x) = f_0(x) + \hat{\Theta}_{\theta,0}(x, \mathbf{x}) \cdot \hat{\Theta}_{\theta,0}^{-1}(\mathbf{x}, \mathbf{x}) \cdot \left( \exp\left( -\hat{\Theta}_{\theta,0}(\mathbf{x}, \mathbf{x}) \cdot \hat{\Xi}(t) \right) - I \right) \cdot (f_0(\mathbf{x}) - \mathbf{y})$$

$$= f_0(x) - \hat{\Theta}_{\theta,0}(x, \mathbf{x}) \cdot \hat{\Theta}_{\theta,0}^{-1}(\mathbf{x}, \mathbf{x}) \cdot \left( I - e^{-\hat{\Theta}_{\theta,0}(\mathbf{x},\mathbf{x}) \cdot \hat{\Xi}(t)} \right) \cdot (f_0(\mathbf{x}) - \mathbf{y}),$$

where $\hat{\Xi}(t) := \int_0^t \exp\left( \hat{\Theta}_{x,0}(\mathbf{x}, \mathbf{x}) \cdot \boldsymbol{\eta}(v) \cdot S \right) \mathrm{d}v$.

The proof is completed. $\qquad\square$

**Proof of Theorem 3.** For any $t \in [0, T]$, suppose $f_v^{\mathrm{sur}}$ is constructed on the interval $[0, t]$ following that introduced in Appendix D and the loss function is squared loss $\mathcal{L}(f(x), y) := \frac{1}{2}\|f(x) - y\|_2^2$.

Notice that the squared loss is 1-smooth and thus satisfies Assumption 4, one can apply Lemma D.2 and have that

$$f_t^{\mathrm{lin}}(x) = \lim_{D\to\infty} f_t^{\mathrm{sur}}(x).$$

Then, by further applying Lemma D.3, we obtain the closed-form solution of $f_t^{\mathrm{lin}}$ for any $x \in \mathcal{X}$ as follows,

$$f_t^{\mathrm{lin}}(x) = \lim_{D\to\infty} f_t^{\mathrm{sur}}(x)$$

$$= f_0(x) - \hat{\Theta}_{\theta,0}(x, \mathbf{x}) \cdot \hat{\Theta}_{\theta,0}^{-1}(\mathbf{x}, \mathbf{x}) \cdot \left( I - e^{-\hat{\Theta}_{\theta,0}(\mathbf{x},\mathbf{x}) \cdot \hat{\Xi}(t)} \right) \cdot (f_0(\mathbf{x}) - \mathbf{y}),$$

where $\hat{\Xi}(t) := \int_0^t \exp\left( \hat{\Theta}_{x,0}(\mathbf{x}, \mathbf{x}) \cdot \boldsymbol{\eta}(\tau) \cdot S \right) \mathrm{d}\tau$.

The proof is completed. $\qquad\square$

# E ADDITIONAL DETAILS IN SECTION 5.1

## E.1 MISSING CALCULATIONS

This section presented calculation details omitted in Section 5.1.

**Calculation of** $e^{-\hat{\Theta}_{\theta,0}(\mathbf{x},\mathbf{x})\cdot\hat{\Xi}(t)}$**.**

By the decomposition $\hat{\Xi}(t) = QA(t)Q^T \cdot a(t)$, we have

$$e^{-\hat{\Theta}_{\theta,0}(\mathbf{x},\mathbf{x})\cdot\hat{\Xi}(t)} = \sum_{i=0}^{\infty} \frac{(-1)^i}{i!} \cdot \left(\hat{\Theta}_{\theta,0}(\mathbf{x},\mathbf{x})\hat{\Xi}(t)\right)^i = \sum_{i=0}^{\infty} \frac{(-a(t))^i}{i!} \cdot \left(\hat{\Theta}_{\theta,0}(\mathbf{x},\mathbf{x})QA(t)Q^T\right)^i.$$

For $\left(\hat{\Theta}_{\theta,0}(\mathbf{x},\mathbf{x})QA(t)Q^T\right)^i$, it can be rewritten as

$$\left(\hat{\Theta}_{\theta,0}(\mathbf{x},\mathbf{x})QA(t)Q^T\right)^i$$
$$= \underbrace{\hat{\Theta}_{\theta,0}(\mathbf{x},\mathbf{x})QA(t)Q^T \cdots \hat{\Theta}_{\theta,0}(\mathbf{x},\mathbf{x})QA(t)Q^T}_{i \text{ number of } \hat{\Theta}_{\theta,0}(\mathbf{x},\mathbf{x})QA(t)Q^T}$$
$$= \hat{\Theta}_{\theta,0}(\mathbf{x},\mathbf{x})QA(t)^{\frac{1}{2}} \cdot \underbrace{A(t)^{\frac{1}{2}}Q^T\hat{\Theta}_{\theta,0}(\mathbf{x},\mathbf{x})QA(t)^{\frac{1}{2}} \cdots A(t)^{\frac{1}{2}}Q^T\hat{\Theta}_{\theta,0}(\mathbf{x},\mathbf{x})QA(t)^{\frac{1}{2}}}_{(i-1) \text{ number of } A(t)^{\frac{1}{2}}Q^T\hat{\Theta}_{\theta,0}(\mathbf{x},\mathbf{x})QA(t)^{\frac{1}{2}}} \cdot A(t)^{\frac{1}{2}}Q^T$$
$$= QA(t)^{-\frac{1}{2}} \cdot A(t)^{\frac{1}{2}}Q^T\hat{\Theta}_{\theta,0}(\mathbf{x},\mathbf{x})QA(t)^{\frac{1}{2}} \cdot \left(A(t)^{\frac{1}{2}}Q^T\hat{\Theta}_{\theta,0}(\mathbf{x},\mathbf{x})QA(t)^{\frac{1}{2}}\right)^{i-1} \cdot A(t)^{\frac{1}{2}}Q^T$$
$$= QA(t)^{-\frac{1}{2}} \cdot \left(A(t)^{\frac{1}{2}}Q^T\hat{\Theta}_{\theta,0}(\mathbf{x},\mathbf{x})QA(t)^{\frac{1}{2}}\right)^i \cdot A(t)^{\frac{1}{2}}Q^T.$$

Combining the above results, we thus have

$$e^{-\hat{\Theta}_{\theta,0}(\mathbf{x},\mathbf{x})\cdot\hat{\Xi}(t)}$$
$$= \sum_{i=0}^{\infty} \frac{(-a(t))^i}{i!} QA(t)^{-\frac{1}{2}} \cdot \left(A(t)^{\frac{1}{2}}Q^T\hat{\Theta}_{\theta,0}(\mathbf{x},\mathbf{x})QA(t)^{\frac{1}{2}}\right)^i \cdot A(t)^{\frac{1}{2}}Q^T$$
$$= QA(t)^{-\frac{1}{2}} \cdot \left(\sum_{i=0}^{\infty} \frac{(-a(t))^i}{i!} \left(A(t)^{\frac{1}{2}}Q^T\hat{\Theta}_{\theta,0}(\mathbf{x},\mathbf{x})QA(t)^{\frac{1}{2}}\right)^i\right) \cdot A(t)^{\frac{1}{2}}Q^T$$
$$= QA(t)^{-\frac{1}{2}} \cdot \exp\left(-A(t)^{\frac{1}{2}}Q^T\hat{\Theta}_{\theta,0}(\mathbf{x},\mathbf{x})QA(t)^{\frac{1}{2}} \cdot a(t)\right) \cdot A(t)^{\frac{1}{2}}Q^T.$$

**Calculation of** $\exp\left(-A(\infty)^{\frac{1}{2}}Q^T \cdot \hat{\Theta}_{\theta,0}(\mathbf{x},\mathbf{x}) \cdot QA(\infty)^{\frac{1}{2}} \cdot a(\infty)\right)$**.**

By the decomposition $A(\infty)^{\frac{1}{2}}Q^T\hat{\Theta}_{\theta,0}(\mathbf{x},\mathbf{x})QA(\infty)^{\frac{1}{2}} = Q'D'Q'^T$, we have

$$\exp\left(-A(\infty)^{\frac{1}{2}}Q^T \cdot \hat{\Theta}_{\theta,0}(\mathbf{x},\mathbf{x}) \cdot QA(\infty)^{\frac{1}{2}} \cdot a(\infty)\right)$$
$$= \exp(-Q'D'Q'^T \cdot a(\infty))$$
$$= \sum_{i=0}^{\infty} \frac{(-a(\infty))^i}{i!}(Q'D'Q'^T)^i$$
$$= \sum_{i=0}^{\infty} \frac{(-a(\infty))^i}{i!} \underbrace{Q' \cdot D'^i \cdot Q'^T}_{\text{By } Q'Q'^T = I}$$
$$= Q' \cdot e^{-D'\cdot a(\infty)} \cdot Q'^T$$
$$\overset{(*)}{=} Q' \cdot \mathrm{Diag}(-\infty) \cdot Q'^T = 0,$$

where $(*)$ is by: (1) $a(\infty) = \infty$, and (2) every diagonal entry of $D'$ is positive.

### E.2 DNNs Behavior Under Large Adversarial Perturbation

In Section 5.1, we have assumed that when the adversarial perturbation scale is small enough, the symmetric matrix

$$H := A(\infty)^{\frac{1}{2}} Q^T \hat{\Theta}_{\theta,0}(\mathbf{x}, \mathbf{x}) Q A(\infty)^{\frac{1}{2}}$$

stays positive definite, which combines with the assumption $\lim_{t\to\infty} a(t) = \infty$ leads to the AT degeneration phenomenon. However, theoretically we can only prove that $H$ is positive semi-definite based on facts that (1) $A(\infty)$ is a diagonal matrix, and (2) $\hat{\Theta}_{\theta,0}(\mathbf{x}, \mathbf{x})$ is positive definite.

Further, when the adversarial perturbation scale is large where $\eta_1(t)S, \cdots, \eta_M(t)S$ are large, the symmetric matrix is likely not be positive definite. This is because in this case the elements in the matrix $A(\infty) := \frac{1}{a(\infty)} \int_0^\infty \exp(D\boldsymbol{\eta}(t)S) \mathrm{d}t$ may vary greatly, which thus erode the positive definiteness of $Q^T \hat{\Theta}_{\theta,0}(\mathbf{x}, \mathbf{x}) Q$ and results in $H$ not being a positive definite matrix. Therefore, we now analyze the remaining cases where $\lambda_{\min}(H) = 0$.

When $\lambda_{\min}(H) = 0$, following similar derivations as that in Section 5.1 and Appendix E.1, we will have that the exponential term in the AT dynamics in Eq. (18) not converging to zero matrix, *i.e.*,

$$\lim_{t\to\infty} e^{-\hat{\Theta}_{\theta,0}(\mathbf{x},\mathbf{x})\cdot\hat{\Xi}(t)} = QA(\infty)^{-\frac{1}{2}} \cdot e^{H\cdot a(\infty)} \cdot A(\infty)^{\frac{1}{2}} Q^T \neq 0, \quad \text{where} \quad \lambda_{\min}(H) = 0.$$

In this case, AT degeneration would not occur in long-term AT, and the adversarially trained wide DNN will eventually converge to a model that is different from that obtained in standard training. However, we also notice that a large adversarial perturbation could add strong noise that can destroy meaningful features within training data, which barriers DNN to effectively learning knowledge. Therefore, it would be interesting to study how to construct NTK models with large adversarial perturbations to achieve strong robustness in practice.

## F EXPERIMENT DETAILS

This section collects experiment details that are omitted from Section 6.

### F.1 PROJECTED GRADIENT DESCENT

We leverage projected gradient descent (PGD) (Madry et al., 2018) to find adversarial examples within constraint spaces in our experiments.

Formally, given a machine learning model $f : \mathcal{X} \to \mathcal{Y}$, a loss function $\mathcal{L} : \mathcal{Y} \times \mathcal{Y} \to \mathbb{R}^+$, and a perturbation radius $\rho > 0$, PGD aims to find the adversarial example $x^{\text{adv}}$ for a given input data point $(x, y)$ via solving the following maximization problem,

$$x^{\text{adv}} = \arg\max_{\|x'-x\|\leq\rho} \mathcal{L}(f(x'), y).$$

PGD will perform $K$ iterations of projection update to find optimal adversarial examples. In the $k$-th iteration, the update is as follows,

$$x^{(k)} = \prod_{\|x'-x\|\leq\rho} \left[ x^{(k-1)} + \alpha \cdot \text{Sign}\left( \partial_{x^{(k-1)}} \mathcal{L}(f(x^{(k-1)}), y) \right) \right],$$

where $x^{(k)}$ is the intermediate adversarial example found in the $k$-th iteration, $\alpha > 0$ is the step size, and $\prod_{\|x'-x\|\leq\rho}$ means the projection is calculated in a ball sphere centered at $x$, *i.e.*, $\{x' : \|x' - x\| \leq \rho\}$. The eventual adversarial example is $x^{\text{adv}} := x^{(K)}$.

### F.2 NETWORK ARCHITECTURES

Our experiments adopt two types of multi-layer DNNs, "MLP-x" and "CNN-x". The detailed architectures of MLP-x and CNN-x are presented in Table 3, where "Conv-$x(c)$" denotes a convolutional layer with kernel shape $x \times x$ and $c$ output channels, and "Dense$(c)$" denotes a fully-connected layer

Table 3: The detailed architectures of MLP-x and CNN-x, where $w$ is the network width.

| MLP-x | CNN-x |
|:---:|:---:|
| $\begin{bmatrix} \text{Dense}^1(w) \\ \text{ReLU} \end{bmatrix}$ | $\begin{bmatrix} \text{Conv-3}^1(w) \\ \text{ReLU} \end{bmatrix}$ |
| $\vdots$ | $\vdots$ |
| $\begin{bmatrix} \text{Dense}^{x-1}(w) \\ \text{ReLU} \end{bmatrix}$ | $\begin{bmatrix} \text{Conv-3}^x(w) \\ \text{ReLU} \end{bmatrix}$ |
| $\text{Dense}(10)$ | $\text{Flatten}$ |
| | $\text{Dense}(10)$ |

with $c$ output channels. Also note that there is a network width hyperparameter $w$ for the architectures in Table 3. For vanilla neural networks, $w$ is a finite value, while for NTK-based models, $w$ is infinite.

We then introduce the detailed model implementations of **Adv-NTK** and baselines **NTK** and **AT**.

**Adv-NTK & NTK.** We use the `Neural-Tangents` Python Library (Novak et al., 2020) which is developed based on the JAX autograd system (Bradbury et al., 2018) to implement NTK-based models in our experiments. The Adv-NTK model is constructed following Eq. (24), while the NTK model is constructed following Eq. (4). For every Adv-NTK and NTK models, the netowrk width is set as $w = \infty$, and the weight and bias standard deviations are set as 1.76 and 0.18, respectively.

**AT.** We use PyTorch (Paszke et al., 2019) to implement the finite-width neural networks in vanilla AT. The network width for the MLP-x architecture is set as $w = 512$, while that for the CNN-x architecture is set as $w = 256$. Other model hyperparameters follow the default settings of PyTorch.

### F.3 TRAINING AND EVALUATION

**General settings.** The loss function used in all experiments is the squared loss $\mathcal{L}(f(x), y) := \frac{1}{2}\|f(x) - y\|_2^2$. Whenever performing PGD to find adversarial examples, we always use $l_\infty$-perturbation. For a given perturbation radius $\rho$, the iteration number and the step size are always set as $K = 10$ and $\alpha = 2\rho/K$, respectively. For both CIFAR-10 and SVHN experiments, we randomly draw $12,000$ samples from the original trainset to train/construct models, and use the overall testset to assess model performances. No data augmentation is used. Every experiment is repeated 3 times.

**Adv-NTK.** We follow Algorithm 1 to train Adv-NTK models. $2,000$ training samples are used as the validation data, while the remaining $10,000$ ones are used to construct Adv-NTK models. In experiments on SVHN, each model is trained for 50 iterations, in which the batch size is set as 128 and learning rate is fixed to 1. In experiments on CIFAR-10, each model is trained for 50 iterations, in which the batch size is set as 128 and learning rate is fixed to 0.1. All other settings follow **general settings**.

**NTK.** Every NTK model is constructed following Eq. (4) with the overall $12,000$ training data. There is no need to train NTK models. All other settings follow **general settings**.

**AT.** We use SGD to train neural networks in AT following Eq.(1) via SGD for $20,000$, where the momentum factor is set as 0.9, the batch size is set as 128, and the weight decay factor is set as 0.0005. In experiments on SVHN, the learning rate is initialized as 0.01 and decay by a factor 0.1 every $8,000$ iterations. In experiments on CIFAR-10, the learning rate is initialized as 0.1 and decay by a factor 0.1 every $8,000$ iterations. All other settings follow **general settings**.

### F.4 EXPERIMENT RESULTS ON SVHN

This section presents the additional experiment results on the SVHN dataset.

From Table 4, we have similar observations as that from the results of CIFAR-10, which show that Adv-NTK can improve the robustness of infinite-width DNNs and sometimes achieve comparable

Table 4: Robust test accuracy (%) of models trained with different methods on SVHN. Every experiment is repeated 3 times. A high robust test accuracy suggests a strong robust generalization ability.

| | Depth | Adv. Acc. ($\ell_\infty$; $\rho = 4/255$) (%) | | | Adv. Acc. ($\ell_\infty$; $\rho = 8/255$) (%) | | |
|---|---|---|---|---|---|---|---|
| | | AT | NTK | Adv-NTK (Ours) | AT | NTK | Adv-NTK (Ours) |
| MLP-x + SVHN (Subset 12K) | 3 | 19.59±0.00 | 18.43±0.53 | **24.64±0.59** | 19.59±0.00 | 9.32±0.15 | **21.68±1.91** |
| | 4 | 19.59±0.00 | 21.47±0.26 | **34.93±1.17** | 19.59±0.00 | 9.00±0.17 | **29.49±0.43** |
| | 5 | 19.59±0.00 | 24.65±0.87 | **36.38±0.45** | 19.59±0.00 | 9.30±0.28 | **25.38±1.42** |
| | 8 | 19.59±0.00 | 28.63±0.18 | **32.35±0.66** | **19.59±0.00** | 10.74±0.37 | 16.22±0.35 |
| | 10 | 19.59±0.00 | **30.48±0.25** | 30.15±0.25 | **19.59±0.00** | 11.62±0.52 | 13.10±0.30 |
| CNN-x + SVHN (Subset 12K) | 3 | **57.07±0.41** | 7.17±0.52 | 22.83±0.76 | **34.92±0.88** | 2.87±0.22 | 20.20±1.05 |
| | 4 | **58.50±0.09** | 9.74±0.48 | 31.74±1.44 | 21.26±2.80 | 3.74±0.36 | **25.14±0.86** |
| | 5 | **54.48±0.97** | 11.60±0.27 | 32.68±0.86 | 19.59±0.00 | 3.85±0.23 | **20.75±2.38** |
| | 8 | 19.59±0.00 | 17.56±0.11 | **23.96±0.96** | **19.59±0.00** | 5.15±0.12 | 9.37±0.16 |
| | 10 | 19.59±0.00 | 21.61±0.51 | **21.98±0.49** | **19.59±0.00** | 5.94±0.37 | 7.08±0.18 |

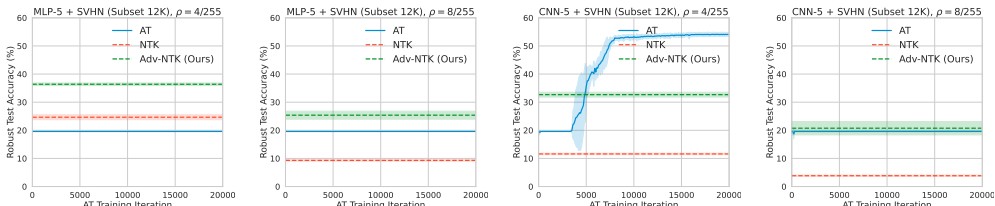

Figure 2: The robust test accuracy curves of finite-width MLP-5/CNN-5 along AT on SVHN. The robust test accuracy of infinite width DNNs learned by NTK and Adv-NTK are also plotted.

robust test accuracy as that of AT. However, we also notice that in the case "CNN-x + SVHN", AT is significantly better than Adv-NTK. We believe it is because the non-linearity of finite-width DNNs in AT can capture additional robustness, which will be left for future studies.

Besides, from Fig. 2, we find that in most of the cases, the finite-width DNN is not trainable in AT. However, in the case of CNN-5 with radius $\rho = 4/255$, the finite-width DNN does not suffer from robust overfitting and its robust test accuracy continuously increases along AT. We deduce that this is because the number of training iterations (which is $20,000$) is too short for this experiment that it almost acts like an early-stop regularization, which is shown to be effective in mitigating robust overfitting (Rice et al., 2020). Furthermore, it is worth noting that the robust overfitting phenomenon on the SVHN dataset has already been observed by Rice et al. (2020) (see Figure 9 in their paper).

# G  APPLYING ADV-NTK TO LARGE DNNS

This section discusses practical challenges and our preliminary results in applying Adv-NTK to common large DNNs.

## G.1  PRACTICAL CHALLENGES TOWARD LARGE DNNS

Our experiments in Section 6 only adopted MLPs and CNNs but not common larger DNNs like VGGs (Simonyan & Zisserman, 2014) and ResNets (He et al., 2016). The reason is that large DNNs usually adopt global average pooling (GAP) layers in their architectures. Although it is confirmed that GAP layers can effectively improve the generalization ability of NTK models, accurately calculating NTK matrices that involve GAP layers would consume an extremely large amount of GPU memory and also lead to an unbearable time usage (Arora et al., 2019; Han et al., 2022).

To improve the calculation efficiency of GAP layers, existing NTK literature has adopted Monte Carlo-based techniques to estimate the output of GAP layers (Novak et al., 2019) and made it affordable for vanilla NTK experiments. However, adapting these Monte Carlo-based techniques into our AT setting is not trivial since our setting needs to further analyze the process of adversarial perturbation, and a series of mathematical and engineering challenges would arise during this adaptation. Therefore, we will leave the practicality problem of Adv-NTK in future research.

Table 5: Robust test accuracy (%) of different ResNet models. Every experiment is repeated 3 times. A high robust test accuracy suggests a strong robust generalizability.

| Dataset | Architecture | Adv. Acc. ($\ell_\infty$; $\rho = 4/255$) (%) | | | Adv. Acc. ($\ell_\infty$; $\rho = 8/255$) (%) | | |
|---|---|---|---|---|---|---|---|
| | | AT | NTK | Adv-NTK (Ours) | AT | NTK | Adv-NTK (Ours) |
| CIFAR-10 | ResNet-18 | 28.01±0.79 | 17.09±0.25 | **28.43±0.27** | 15.11±0.65 | 4.07±0.13 | **21.46±0.47** |
| (Subset 12K) | ResNet-34 | 26.61±1.08 | 25.42±0.31 | **29.19±0.64** | 16.11±0.55 | 9.64±0.12 | **21.00±0.82** |
| SVHN | ResNet-18 | 48.10±1.08 | 24.18±0.34 | 32.70±0.26 | **30.36±1.92** | 10.05±0.30 | 13.96±0.16 |
| (Subset 12K) | ResNet-34 | 48.04±0.99 | 32.69±0.43 | 34.08±0.29 | **26.94±2.33** | 13.01±0.19 | 15.19±0.26 |

(a) Results on CIFAR-10.

(b) Results on SVHN.

Figure 3: The robust test accuracy curves of ResNet models along AT. The robust test accuracy of infinite width DNNs learned by NTK and Adv-NTK are also plotted.

### G.2 EXPERIMENTS ON RESNETS

Although we still could not address the practical challenges induced by GAP layers, in this section we present the experiments for ResNets that are **not using GAP layers**.

**Models.** We adopt two types of ResNet (He et al., 2016), ResNet-18 and ResNet-34, in our experiments. For finite-width models, GAP layers are adopted as that in standard ResNet architectures. For NTK models, we replace all GAP layers with flattening layers to reduce GPU memory usage.

**Experiment setup.** (1) For **Adv-NTK**, the learning rate is set as: $5 \times 10^{-5}$ for ResNet-18 on SVHN, $10^{-6}$ for ResNet-18 on CIFAR-10, $10^{-8}$ for ResNet-34 on SVHN, and $10^{-10}$ for ResNet-34 on CIFAR-10. Other settings follow that in Appendix F.3. (2) For **NTK**, all settings follow Appendix F.3. (3) For **AT**, the learning rate for both ResNet-18 and ResNet-34 on CIFAR-10 is set as 0.01. Other settings follow that in Appendix F.3.

**Results.** The robust test accuracies of ResNet models trained with different methods are reported in Table 5. The curves of robust test accuracy of finite-width ResNet models along AT are plotted in Fig. 3. From the results, one can find that Adv-NTK can effectively improve the robustness of infinite-width ResNet models. Meanwhile, on the SVHN dataset, finite-width ResNet models achieved stronger robustness than infinite-width ones. We think this is due to the use of GAP layers in finite-width ResNet models, and will leave the application of GAP layers in large-scale Adv-NTK for future studies.

