# OpenReview forum: "Theoretical Analysis of Robust Overfitting for Wide DNNs: An NTK Approach"
_ICLR.cc/2024/Conference — ICLR 2024 poster_

### Official Review · Reviewer_edb8 · 2023-10-28

**Soundness:** 3 good
**Presentation:** 3 good
**Contribution:** 2 fair
**Rating:** 6
**Confidence:** 3

**Summary:**

This paper applied neural tagent kernel techniques into adversarial training setting and proves that adversarial trained DNN can be approximated by a linearized DNN. For square loss it reveals a AT degreneration phenomena so that explains robust overfitting. The paper designed an algorithm AdvNTK based on the theoretical results.

**Strengths:**

I like the idea of analyzing robust overfitting from the time-dependent regularizer matrix that derived from adversarial NTK.

**Weaknesses:**

Adversarial trained DNN can be approximated by linearized DNN only when the perturbation size considered is small, however, the paper seems not give a quantitive result regarding how small it should be. I understand that the author using attack learning rate for a total of time S. Yet it’s unclear how large the attack rate is, as if the attack learning rate is infinitesimal, then the perturbation size is so small, and it’s unclear for me whether such adversarial training algorithm has any generalization guarantee in terms of robustness.

Moreover, I’m not sure why it’s interesting to study adversarial training within the NTK regimes, as in [1] show when the network is close to initialization, there’s no robust network at all.

In the proof of theorem 2 the dependency of polyt seems odd to me, as polyt depends on the norm of W_t, which can go to infinity. I understand that for neural tagent kernel the network width does not move far from initialization so you can argue norm of W_t is bounded,  as theorem Lemma C.5, yet this theorem’s proof also depends on polyt. Therefore the current presentation of omiting dependency on the norm of W_t, nor of x_t seems to hide something in the proof.

What’s the relationship between the experiment vs. theorems? How does the experiment validate theorems? The robust accuracy of both SVHN and CIFAR10 seems extremely lower than normal adversarial training. Is it because of the network architecture? Why cannot use standard architecture such as ResNet that people commonly use in practice? Can the author provide any simple experiment to confirm the AT degeneration phenomenon? In fact, it’s unclear to me the difference between this wording vs robust overfitting, and if they are the same thing then there’s no need to use a different paraphrase.

In terms of the writing, I think the author should put more discussion on the theorems, explain, and provide intuition.

[1] Wang, Y., Ullah, E., Mianjy, P., & Arora, R. (2022). Adversarial robustness is at odds with lazy training. Advances in Neural Information Processing Systems, 35, 6505-6516.

**Questions:**

This paper focuses on regression setting with squared loss. I’m wondering if the idea can be generalized to classification setting.

---

> ### Author Response · Authors · 2023-11-17
> **To Reviewer edb8 (1/3)**
>
> Thank you for your thorough review and constructive comments. All your concerns have been carefully responded below. The manuscript is carefully revised accordingly. We sincerely hope our responses fully address your questions.
>
> **Q1.1:** *Adversarial trained DNN can be approximated by linearized DNN only when the perturbation size considered is small, however, the paper does not give a quantitative result regarding how small it should be.*
>
> **A1.1:** We respectfully argue that the approximation based on linearized DNNs in AT does not require the perturbation size to be small. Instead, such an approximation mainly depends on if the network widths are large enough. The only requirement for the perturbation size is that the learning rate should be continuously differentiable (see Assumption 3 on Page 6). We have added Remark 2 in Section 4.3 on Page 6 in the revised version to clarify this result, please kindly refer for more details.
>
> **Q1.2:** *I understand that the author using attack learning rate for a total of time S. Yet it's unclear how large the attack rate is, as if the attack learning rate is infinitesimal, then the perturbation size is so small, and it's unclear for me whether such adversarial training algorithm has any generalization guarantee in terms of robustness.*
>
> **A1.2:** We currently have no generalization guarantee in terms of robustness as well as the scale of the learning rate matrix $\boldsymbol{\upeta}(t)$. Nevertheless, we respectfully note that all the theoretical results in Section 4 do not depend on the scale of the learning rate matrix $\boldsymbol{\upeta}(t)$. The only requirement for the learning rate matrix $\boldsymbol{\upeta}(t)$ is to satisfy Assumption 3 (in Section 4.3 on Page 6). Therefore, one can choose any large enough $\boldsymbol{\upeta}(t)$ to perform an effective AT as long as Assumption 3 is fulfilled.
>
> **Q2:** *Moreover, I'm not sure why it's interesting to study adversarial training within the NTK regimes, as in [r1] show when the network is close to initialization, there’s no robust network at all.*
>
> **A2:** Thanks. This is a good question. In short, the setting we studied is different from that in [r1] and thus it is of merit to study DNN in AT with NTK. Detailed reasons will then be explained. We have also added a discussion about [r1] in the revised version (see Section 2 on Page 2).
>
> Firstly, we would like to clarify that the setting we studied in our paper is more challenging and realistic than that in [r1]. Specifically, the two settings mainly differ in the following two aspects:
> - The results in [r1] only work for a two-layer neural network where the second-layer parameter is frozen, while our results work for neural networks with any (finite) number of layers and fully trainable parameters.
> - The results in [r1] only work for input data $x$ that is from the unit sphere $\mathbb{S}^{d-1}$, while our results work for any $x \in \mathbb{R}^d$.
>
> As a result, although it has been well studied that in the setting of [r1], there is no robust network in the NTK regime, it remains unknown what is the behavior of an adversarially trained deep neural network in the NTK regime. Therefore, studying DNN in AT with NTK theory is still of great importance.
>
> **Q3:** *In the proof of theorem 2 the dependency of $poly_t$ seems odd to me, as $poly_t$ depends on the norm of $W_t$, which can go to infinity. I understand that for a neural tangent kernel the network width does not move far from initialization so you can argue the norm of $W_t$ is bounded, as in Lemma C.5, yet this Lemma’s proof also depends on $poly_t$. Therefore the current presentation of omiting dependency on the norm of $W_t$, nor of $x_t$ seems to hide something in the proof.*
>
> **A3**: Firstly, we respectfully note that $\mathrm{Poly}\_t$ depends on $\frac{1}{\sqrt{n\_{l-1}}} || W^{(l)}\_t ||\_2$, rather than only $||W^{(l)}\_t||\_2$.
>
> Besides we did not argue that $|| W^{(l)}\_t ||\_2$ is bounded. Instead, we prove that as the network widths $n\_0,\cdots,n\_L$ go to infinity, $\frac{1}{\sqrt{n\_{l-1}}}||W^{(l)}\_t||\_2$ will be bounded while $||W^{(l)}\_t||\_2$ can go to infinity.
>
> Moreover, our proofs in Appendix C did not omit the dependency of $\mathrm{Poly}\_t$ on $\frac{1}{\sqrt{n\_{l-1}}}||W^{(l)}\_t||\_2$, $\frac{1}{\sqrt{n\_l}} ||x^{(l)}\_t(\boldsymbol{\mathrm{x}}\_{,t,S})||\_2$, and $\frac{1}{\sqrt{n\_l}} \|h^{(l)}\_t(\boldsymbol{\mathrm{x}}\_{,t,S})\|_2$. $\mathrm{Poly}\_t$ is the abbreviation of any deterministic polynomial with finite degree and finite positive constant coefficients, considering the aforementioned l2-norm terms. We use this abbreviation only for the sake of simplifying writing.
>
> Furthermore, the proof of Lemma C.5 relies on the dependency of $\mathrm{Poly}_t$ on these l2-norm terms. Specifically, the construction of the new polynomial $P(A_t)$ relies on the aforementioned dependency. Please kindly refer to the bottom of Page 24 in the revised manuscript for more details.

---

> > ### Author Response · Authors · 2023-11-17
> > **To Reviewer edb8 (2/3)**
> >
> > **Q4.1:** *What’s the relationship between the experiment vs. theorems? How does the experiment validate theorems?*
> >
> > **A4.1:** The main conclusion of our theorems is that the adversarial robustness brought by AT is captured by a regularization matrix $\Xi(t)$ (see Corollary 1 on Page 6), and our experiments aim to empirically verify if $\Xi(t)$ can indeed capture the robustness by directly optimizing the elements of $\Xi(t)$. The experiment results show that our designed strategy (i.e., optimizing $Xi(t)$) can improve the adversarial robustness of DNNs (see Table 1 on Page 9 and Table 4 on Page 44), which therefore validates our theorems’ finding.
> >
> > **Q4.2:** *The robust accuracy of both SVHN and CIFAR10 seems extremely lower than normal adversarial training. Is it because of the network architecture?*
> >
> > **A4.2:** Thanks for your question.
> >
> > **For vanilla AT**, we believe the low robust accuracy is because we only use a small subset of training data (12K data points) to train the finite-width DNNs. Current NTK experiments are usually conducted on small subsets for computational efficiency [r2, r3]. For example, in [r3] the experiments are mainly conducted on subsets of CIFAR-10 of size 5K and 10K. Our experiments follow similar settings, so datasets used in vanilla AT are also shrunk accordingly for fair comparisons.
> >
> > **For NTK/Adv-NTK**, the low accuracy of MLP is normal, and we believe the low accuracy of CNN is mainly because we did not use the global average pooling (GAP) layer in our CNN-based NTK models. Existing NTK literature has shown that GAP layers can significantly improve the performance of NTK models in standard training [r3, r4]. Therefore, it is reasonable to deduce that adopting the GAP layer in AT would also effectively improve the robust accuracy of CNN-based NTK models.
> >
> > From the above discussion, one can find that there are two promising solutions to improve the robust accuracy results in our paper:
> > - Increase the (used) training dataset.
> > - Use the GAP layer.
> >
> > We did not leverage these solutions because they would consume a large amount of computation resources if we want to accurately calculate them. Existing NTK literature has adopted Monte Carlo-based techniques to estimate the NTK kernel of large datasets and the output of GAP layers [r5], which makes the aforementioned two solutions realistic for vanilla NTK experiment. However, adapting these Monte Carlo-based techniques into our AT setting is not trivial since our setting needs to further analyze the process of adversarial perturbation, and a series of mathematical and engineering challenges would arise during this adaptation. Therefore, we will leave the practicality problem of Adv-NTK in future research but not in this paper.
> >
> > Finally, we would like to highlight that the main contribution of this paper is to provide a new theoretical explaination for the robust overfitting phenomenon when adversarially training DNNs, but not designing a practical AT algorithm. The role of our empirical evidence in Section 6 is to verify our theoretical result that the regularization matrix $\Xi(t)$ indeed captures the robustness brought by AT. The results in Table 1 on Page 9 and Table 4 on Page 44 show that the robust accuracy of Adv-NTK is significantly higher than that of NTK, which already verified our main theoretical contribution.

---

> ### Author Response · Authors · 2023-11-17
> **To Reviewer edb8 (3/3)**
>
> **Q4.3:** *Why cannot use standard architecture such as ResNet that people commonly use in practice?*
>
> **A4.3:** Thanks for your question. The reason that we did not use ResNet and other standard architectures in our experiments is that most of these architectures leverage global average pooling (GAP). However, as explained in A4.2, we currently cannot efficiently calculate NTK models that adopt GAP layers under AT. We originally intended to leave these architectures that adopted GAP for future research.
>
> Nevertheless, we have conducted additional experiments with ResNet-18 by replacing the GAP layer with the Flatten layer. The experiment results are shown as the following Table r1.
>
>
> | Dataset               | Radius | AT             | NTK            | Adv-NTK (Ours) |
> | --------------------- | ------ | -------------- | -------------- | -------------- |
> | CIFAR-10 (Subset 12K) | 4/255  | 28.01$\pm$0.79 | 17.09$\pm$0.25 | 28.43$\pm$0.27 |
> | CIFAR-10 (Subset 12K) | 8/255  | 15.11$\pm$0.65 | 4.07$\pm$0.13  | 21.46$\pm$0.47 |
> | SVHN (Subset 12K)     | 4/255  | 48.10$\pm$1.08 | 24.18$\pm$0.34 | 32.70$\pm$0.26 |
> | SVHN (Subset 12K)     | 8/255  | 30.36$\pm$1.92 | 10.05$\pm$0.30 | 13.96$\pm$0.16 |
>
> *Table r1: Robust test accuracy (%) of ResNet-18 (without GAP layer) trained with different methods.*
>
> Comparing Table r1 with our original experiment results (see Table 1 on Page 9 and Table 4 on Page 44), we can find that the robustness performance of the ResNet-18 trained with Adv-NTK is even worse than that of MLP-x/CNN-x. We deduce that this is mainly due to the fact that current ResNet-18 NTK models did not adopt the GAP layer. We will add a more comprehensive empirical analysis of ResNet and a discussion on its practicality in the next version of our manuscript.
>
> **Q4.4:** *Can the author provide any simple experiment to confirm the AT degeneration phenomenon?*
>
> **A4.4:** Thanks for your suggestion. We have added additional experiments and plotted the results in Fig.1 (on Page 9) and Fig.2 (on Page 44). Specifically, we plotted the robust test accuracy of finite-width MLP-5/CNN-5 along vanilla AT. Please kindly refer to Fig.1 and Fig.2 and corresponding discussions along with them.
>
> Besides, we would also like to highlight that [r6] has already empirically confirmed the AT degeneration/robust overfitting (note that "empirical" AT degeneration can be seen as robust overfitting, see **A4.5**) in 2020. Please kindly refer to [r6] for a more comprehensive background knowledge of robust overfitting.
>
> **Q4.5:** *In fact, it’s unclear to me the difference between this wording (AT degeneration) vs robust overfitting, and if they are the same thing then there’s no need to use a different paraphrase.*
>
> **A4.5:** Please note that the two wordings are different. Specifically: (1) "robust overfitting" refers to an empirical phenomenon that currently lacks explanations. (2) "AT degeneration" refers to a theoretical phenomenon that characterizes the cause of "robust overfitting", i.e., "a DNN in long-term AT will graduate degenerate to that obtained without AT".
>
> **Q5:** *In terms of the writing, I think the author should put more discussion on the theorems, explain, and provide intuition.*
>
> **A5:** Thanks for your suggestion. We have revised the manuscript to improve its readability as follows:
> - In Section 1 (on Page 1), we explained our motivation for using NTK theory to study robust overfitting.
> - In Section 1 (on Page 2), we added a new paragraph to summarize the main contributions of this paper.
> - In Section 4.3 (on Page 6), we added Remark 2 to highlight that Theorem 2 depends mainly on network widths but not adversarial perturbation scales.
>
> The manuscript will be continuously revised accordingly in the next version due to time limitation.
>
> **Q6:** *This paper focuses on regression setting with squared loss. I’m wondering if the idea can be generalized to classification settings.*
>
> **A6:** Thanks for your question. The regression setting with squared loss can be generalized to the classification by simply replacing the categorical label-encoding with the one-hot label-encoding during model training. Moreover, we respectfully note that our empirical analysis in Section 6 was conducted on the image classification task.
>
> **References:**
>
> [r1] Wang et al. “Adversarial robustness is at odds with lazy training.” NeurIPS 2022.
>
> [r2] Arora et al. "Harnessing the power of infinitely wide deep nets on small-data tasks." ICLR 2020.
>
> [r3] Han et al. "Fast neural kernel embeddings for general activations." NeurIPS 2022.
>
> [r4] Arora et al. "On exact computation with an infinitely wide neural net." NeurIPS 2019.
>
> [r5] Novak et al. "Fast finite width neural tangent kernel." ICML 2022.
>
> [r6] Rice et al. "Overfitting in adversarially robust deep learning." ICML 2020.

---

> ### Author Response · Authors · 2023-11-19
> **Follow-up on rebuttal**
>
> Dear Reviewer edb8,
>
> Thank you again for your valuable comments! We have carefully considered your comments and have provided our responses.
>
> Please let us know if our replies have satisfactorily addressed your concerns. Please do not hesitate to let us know if you have any further questions or if you require any additional clarification.
>
> Thank you very much!

---

> > ### Author Response · Authors · 2023-11-21
> >
> > Dear Reviewer edb8,
> >
> > Thanks again for your time and valuable comments.
> >
> > Since the discussion stage is about to end, we are writing to kindly ask if our replies have addressed your concerns. Please kindly let us know if you have any additional concerns and we are happy to discuss more.
> >
> > Thank you very much!

---

> > > ### Comment · Reviewer_edb8 · 2023-11-21
> > >
> > > Dear Author,
> > >
> > > Thank you for your response which addresses most of my questions. I'll increase the score accordingly. Please add the discussion as well as the new results to the final manuscript.

---

> > > > ### Author Response · Authors · 2023-11-22
> > > > **Thanks for your support and suggestion!**
> > > >
> > > > Thank you very much for recognizing our contributions and kind support!
> > > >
> > > > We will add the following in the final version of our manuscript accordingly:
> > > > - A new empirical analysis on ResNet architectures (without GAP layer).
> > > > - A discussion on how to improve the practicality of NTK models in AT when using ResNet or other common model architectures.
> > > >
> > > > Thanks for your suggestion!

---

### Official Review · Reviewer_4ePw · 2023-10-30

**Soundness:** 4 excellent
**Presentation:** 3 good
**Contribution:** 3 good
**Rating:** 8
**Confidence:** 3

**Summary:**

•	This paper theoretically explores the robust overfitting of Adversarial Training (AT). Specifically, it demonstrates that Deep Neural Networks (DNNs) trained with AT can be represented from the perspective of Neural Tangent Kernel (NTK). The paper further formulates the dynamics of Adversarial Training. Based on this formulation, it explains the reasons behind the occurrence of robust overfitting and proposes an algorithm, termed ADV-NTK, that can prevent robust overfitting in a manner akin to early stopping.

**Strengths:**

•	The paper tackles a critical and pressing issue: Overfitting in Adversarial Training.

•	The contribution of paper is based on a theoretically solid proof. They also offer a detailed and meticulous explanation enhancing the understandings.

**Weaknesses:**

•	The paper is written based on the logical flow of formulation and comparably not focusing on the motivation of the paper. For better readability, it seems to be a need for more appeal on how important task the paper tries to solve and what is the contributions of the paper.

•	In context of DNN-NTK, they replace the constrained-spaces condition with an additional learning rate term to control the strength of adversarial examples. (in Equation 7) However, unlike many attack mechanisms, it doesn't efficiently find a significant direction of attack. Despite this, does believe that the derived formula's proof is still not too loose?

•	The additional experiments are needed. In specific, it would be better to empirically verify 1) whether theoretically proven properties actually happen similarly in real-world dataset and 2) how much the robust overfitting problem has been addressed with the proposed Adv-NTK. For instance, tracking the performance (or loss) trend across iterations between vanilla AT and Adv-NTK.

**Questions:**

•	In the above part.

[Overall]
•	If the authors provide further experiments on Adv-NTK and supplementary explanations for the justification for learning rates, I agree that this paper is accepted.

---

> ### Author Response · Authors · 2023-11-17
> **To Reviewer 4ePw**
>
> Thank you for your constructive comments and kind support! All your concerns have been carefully responded below. The manuscript is carefully revised accordingly. We sincerely hope our responses fully address your questions.
>
> **Q1:** *The paper is written based on the logical flow of formulation and comparably not focusing on the motivation of the paper. For better readability, there seems to be a need for more appeal on how important the task the paper tries to solve and what are the contributions of the paper.*
>
> **A1:** Thanks for your suggestion. We have revised Section 1 according to your suggestion as follows:
> - In the first paragraph, we further explained why studying robust overfitting is important.
> - In the third paragraph, we explained the motivation for using NTK theory to study robust overfitting.
> - We added a new paragraph that summarizes the main contribution of this paper.
>
> Please also kindly refer to Section 1 for more details.
>
> **Q2:** *In the context of DNN-NTK, they replace the constrained-spaces condition with an additional learning rate term to control the strength of adversarial examples (in Eq.7). However, unlike many attack mechanisms, it doesn't efficiently find a significant direction of attack. Despite this, do you believe that the derived formula's proof is still not too loose?*
>
> **A2:** We respectfully argue that our gradient flow-based attack mechanism in Eq.(7) indeed finds significant direction to perform adversarial attacks, and thus our theoretical results are not loose compared with real-world settings.
>
> Specifically, the mechanism in Eq.(7) is very similar to projected gradient descent (PGD), a very common adversarial attack method in the real-world setting (see Appendix F.1). The only difference is that our mechanism based on Eq.(7) leverages a learning rate $\eta_i(t)$ and a total search time $S$ to control the adversarial strength (i.e., the strength of the ability to make models misbehave), while PGD uses a perturbation radius $\rho$ to control the adversarial strength. As long as $\eta_i(t) S$ is set to be large enough, our mechanism can find adversarial examples to effectively attack targeted DNNs, just like the PGD attack. Please also kindly refer to the first two paragraphs in Section 4.1 on Page 4 for a detailed discussion.
>
> **Q3:** *Additional experiments are needed. In specific, it would be better to empirically verify: (1) Whether theoretically proven properties actually happen similarly in real-world dataset. (2) How much the robust overfitting problem has been addressed with the proposed Adv-NTK. For instance, tracking the performance (or loss) trend across iterations between vanilla AT and Adv-NTK.*
>
> **A3:** Thanks for your suggestion. We have added additional experiments and plotted the results in Fig.1 (on Page 9) and Fig.2 (on Page 44). Specifically, we plotted the robust test accuracy of finite-width MLP-5/CNN-5 along vanilla AT. We also plotted the accuracy of infinite-width DNNs learned by NTK and Adv-NTK. The general finding is that: although Adv-NTK can achieve comparable or higher performance than the final model obtained by AT in some cases, it could not beat the best model along vanilla AT. This suggests there are still theoretical gaps between finite-width and infinite-width DNNs that need to be fulfilled.
>
> Besides, we would also like to highlight that our original experiment results in Table 1 (on Page 9) and Table 4 (on Page 44) have already empirically verified the raised questions. Specifically, on one hand, the NTK method in our experiments will learn infinite-width DNNs defined as Eq.(4) (in Section 3 on Page 3). On the other hand, as discussed in Section 5.1 (on Page 6), a long-term AT will result in an infinite-width DNN degenerate to the limit of Eq.(3) (in Section on Page 3), which however is exactly Eq.(4). Therefore, if Adv-NTK can learn infinite-width DNNs that are of better performance than DNNs learned by NTK (i.e., constructed based on Eq.(4)), then it is enough to illustrate that Adv-NTK can mitigate robust overfitting. And our results in Tables 1 and 4 indeed show that Adv-NTK performs better than NTK.

---

> ### Comment · Reviewer_4ePw · 2023-11-21
>
> Dear Authors,
>
> Thank you for your kind response.
> Most of my concerns are addressed by your response. I will raise my score to Accept.
>
> Sincerely,
>
> Reviewer 4ePw

---

> > ### Author Response · Authors · 2023-11-21
> > **Thanks!**
> >
> > Thank you very much for recognizing our contributions and kind support!

---

### Official Review · Reviewer_s7R7 · 2023-10-31

**Soundness:** 3 good
**Presentation:** 3 good
**Contribution:** 2 fair
**Rating:** 6
**Confidence:** 4

**Summary:**

This paper studies the problem of robust overfitting in adversarial training using the Neural Tangent Kernel (NTK). First, the authors extend the theoretical framework of NTK to adversarial training, introducing an adversarial regularization kernel and demonstrating that adversarially trained DNNs can be well approximated by their linearized DNNs. They then derive the closed-form dynamics of adversarial training for the linearized DNN and uncover a phenomenon called adversarial training degeneration: prolonged adversarial training leads to the degradation of the wide DNN to a state similar to that of a DNN with normal training, which results in robust overfitting. Based on these theoretical findings, the authors propose an adversarial training algorithm called Adv-NTK for infinite-width DNNs, and experimental results show that it can enhance the robustness of infinite-width DNNs to a level comparable to that of finite-width DNNs.

**Strengths:**

This paper provides a linear model-level explanation of the adversarial training degeneration phenomenon. The designed Adv-NTK algorithm enables infinite-width NTK models to achieve robustness similar to MLPs through adversarial training.

**Weaknesses:**

1: This paper focuses on linearized models but lacks a detailed explanation of why linearization approximation is applicable. Theorem 1 demonstrates the convergence of two kernels for initialization. However, the properties that remain constant over time appear to be submerged in Appendix C.2. It is recommended that the authors include an informal presentation of the results from Appendix C.2 between Theorems 1 and 2 and provide corresponding discussions.

2: Section 5.1 lacks a discussion regarding the impact of $\eta S$ on the adversarial training degeneration phenomenon.

**Questions:**

Are the results in Section 4 of this paper valid for adversarial training of any intensity? Intuitively, if the intensity of adversarial training is too high, some of the results in Chapter 4 may not be valid.

---

> ### Author Response · Authors · 2023-11-13
> **Request for clarifying "intensity of adversarial training"**
>
> Dear Reviewer s7R7,
>
> Thanks for your detailed review. We are currently working on the rebuttal to carefully answer your questions.
>
> To fully address all your concerns, we would like to ask what do you mean by
> **"intensity of adversarial training"**? Does it mean **the scale of the learning rate matrix $\boldsymbol{\upeta}(t)$**, or **the total perturbation time $S$**, or **both of them**?
>
> A clarification of the term "intensity of adversarial training" would be very helpful for us to prepare our answers!
>
> Thanks & sincerely,
>
> The Authors

---

> > ### Comment · Reviewer_s7R7 · 2023-11-13
> >
> > Dear Authors,
> >
> > Thanks for your response.
> >
> > I am concerned about whether the results still hold when the product $\eta(t) S$ (or the integral $\int\eta(t)dt$) is large. I don’t seem to see any limitations or assumptions in Theorem 2 about this (if I overlooked something, please tell me).
> >
> > Thanks,
> >
> > Reviewer s7R7

---

> ### Author Response · Authors · 2023-11-17
> **To Reviewer s7R7**
>
> Thank you for your thorough review and constructive comments. All your concerns have been carefully responded below. The manuscript is carefully revised accordingly. We sincerely hope our responses fully address your questions.
>
> **Q1.1:** *This paper focuses on linearized models but lacks a detailed explanation of why linearization approximation is applicable.*
>
> **A1.1:** Firstly, the idea that "large network widths lead to model linearization" comes from (vanilla) NTK theory and has already been theoretically proved in standard training (for example, see [r1, r2, r3, r4]). We respectively note that we have explained this linearization idea in both Section 2 (on Page 2) and Section 3 (on Page 3).
>
> Besides, in this paper, we generalized the linearization idea in vanilla NTK theory to the setting of AT and justified why "wide DNNs can still be linearized in AT" with rigorous proofs. We have added Remark 2 in Section 4.3 on Page 6 in the revised version to clarify this contribution, please also kindly refer for more details.
>
> **Q1.2:** *Theorem 1 demonstrates the convergence of two kernels for initialization. However, the properties that remain constant over time appear to be submerged in Appendix C.2. It is recommended that the authors include an informal presentation of the results from Appendix C.2 between Theorems 1 and 2 and provide corresponding discussions.*
>
> **A1.2:** Thanks for your suggestion. We have carefully revised our manuscript accordingly. Specifically, we made the following revisions:
>
> - We have added a new Appendix C.1 to explain the proof skeleton of Theorem 2.
> - We have informally presented the main results of Appendix C.3 (i.e., the old Appendix C.2) in Appendix C.1. They are Theorems C.1 and C.2, which illustrate that the two kernels remain constant during AT.
>
> Please also kindly refer to Appendix C.1 on Page 17 for more details.
>
> **Q2:** *Section 5.1 lacks a discussion regarding the impact of $\eta S$ on the adversarial training degeneration phenomenon.*
>
> **A2:** Thanks for your suggestion. We have added an additional discussion on the behavior of DNNs in long-term AT when large adversarial perturbations are present. Please kindly refer to Appendix E.2 on Page 42 for details.
>
> **Q3:** *Are the results in Section 4 of this paper valid for adversarial training of any intensity? Intuitively, if the intensity of adversarial training is too high, some of the results in Section 4 may not be valid.*
>
> **Clarification:** *I am concerned about whether the results still hold when the product $\boldsymbol{\upeta}(t)S$ (or the integral $\int \boldsymbol{\upeta}(t) \mathrm{d}t$) is large. I don’t seem to see any limitations or assumptions in Theorem 2 about this (if I overlooked something, please tell me).*
>
> **A3:** Thanks for your question and clarification. Yes, the results in Section 4 are valid for adversarial training of any intensity. Specifically, our theoretical result that "a wide DNN under AT behaves like a linearized DNN" does not depend on the scale of $\boldsymbol{\upeta}(t)S$. Instead, it mainly depends on if the network widths are large enough **(This is one of the main reasons why our theoretical result is exciting)**. The only requirement for $\boldsymbol{\upeta}(t)S$ is that the learning rate matrix $\boldsymbol{\upeta}(t)$ should be continuously differentiable (see Assumption 3 on Page 6).
>
> We have added Remark 2 in Section 4.3 on Page 6 in the revised version to highlight this contribution, please kindly refer for more details.
>
> **References:**
>
> [r1] Jacot et al. "Neural tangent kernel: Convergence and generalization in neural networks." NeurIPS 2018.
>
> [r2] Lee et al. "Wide neural networks of any depth evolve as linear models under gradient descent". NeurIPS 2019.
>
> [r3] Arora et al. "On exact computation with an infinitely wide neural net." NeurIPS 2019.
>
> [r4] Hron et al. "Infinite attention: NNGP and NTK for deep attention networks." ICML 2020.

---

> ### Author Response · Authors · 2023-11-19
> **Follow-up on rebuttal**
>
> Dear Reviewer s7R7,
>
> Thank you again for your valuable comments! We have carefully considered your comments and have provided our responses.
>
> Please let us know if our replies have satisfactorily addressed your concerns. Please do not hesitate to let us know if you have any further questions or if you require any additional clarification.
>
> Thank you very much!

---

> > ### Comment · Reviewer_s7R7 · 2023-11-20
> >
> > Dear Authors,
> >
> > Thank you for your response, which addressed most of my concerns. I will raise my score accordingly. It would be better if you could refer to the additional discussions provided in the appendix within the relevant sections of the main text.
> >
> > Thank you,
> >
> > Reviewer s7R7

---

> > > ### Author Response · Authors · 2023-11-20
> > > **Thanks for your support and suggestion!**
> > >
> > > Thank you very much for recognizing our contributions and kind support!
> > >
> > > We will add reference to Appendix E.2 (i.e., the additional discussions) in Section 5 in the next version of our manuscript.
> > > Thanks for your suggestion!

---

### Official Review · Reviewer_gmGK · 2023-11-10

**Soundness:** 3 good
**Presentation:** 2 fair
**Contribution:** 3 good
**Rating:** 6
**Confidence:** 3

**Summary:**

The paper explores the issue of robust overfitting in deep neural networks (DNNs) during adversarial training (AT). It extends neural tangent kernel (NTK) theory to explain this phenomenon for infinite width deep networks, showing that a wide DNN under AT can behave like a linearized DNN, leading to AT degeneration over time (assuming squared loss). To address this, the paper introduces Adv-NTK, an novel AT algorithm for infinite-width DNNs, designed to enhance network robustness. The effectiveness of Adv-NTK is demonstrated through experiments on real-world datasets, establishing its real potential in improving DNN robustness against adversarial attacks.

**Strengths:**

The paper attempts to tackle an important problem of overfitting in adversarial training from a theoretical perspective, which should be relevant to the broader community.

The empirical validation of the Adv-NTK algorithm using real-world datasets like SVHN and CIFAR-10 enhances the quality of the paper. This empirical approach ensures that the theoretical findings are not only sound in theory but also applicable and effective in real-world scenarios.

**Weaknesses:**

Its not clearly under what conditions the small step size can lead to linearization of the adversarial training of DDN.

In the proof, it does not try to find efficient direction, does that make any difference on the proof ?

**Questions:**

It would improve the paper if bound on the learning rate can be provided with respect to its training (or maybe some empirical results towards that).

---

> ### Author Response · Authors · 2023-11-13
> **Request for clarifying "efficient direction"**
>
> Dear Reviewer gmGK,
>
> Thanks for your detailed review and kind support. We are currently working on the rebuttal to carefully answer your questions.
>
> To fully address all your concerns, we would like to ask that in your question 2, what does **"efficient direction"** mean? Does it mean **the direction for searching adversarial examples**?
>
> If not, a clarification of the term "efficient direction" would be very helpful for us to prepare our answers!
>
> Thanks & sincerely,
>
> The Authors

---

> ### Author Response · Authors · 2023-11-17
> **To Reviewer gmGK**
>
> Thank you for your constructive comments and kind support! All your concerns have been carefully responded below. The manuscript is carefully revised accordingly. We sincerely hope our responses fully address your questions.
>
> **Q1:** *It is not clear under what conditions the small step size can lead to linearization of the adversarial training of DDN.*
>
> **A1:** We respectfully note that the linearization of DNNs in AT does not require the step size in adversarial perturbation to be small. Instead, the linearization of DNNs mainly depends on if the network widths are large enough.
>
> The idea that "large network widths lead to model linearization" has already been proved in vanilla NTK theory for standard training (for example, see [r1, r2, r3]). We further proved this idea in the setting of AT, and this is one of the most exciting contributions of our paper. We have added Remark 2 in Section 4.3 on Page 6 in the revised version to clarify this contribution, please kindly refer for more details.
>
> **Q2:** *In the proof, it does not try to find an efficient direction, does that make any difference on the proof?*
>
> **A2:** Firstly, we respectfully argue that the gradient flow-based attack mechanism (based on Eq.(7)) leveraged in our proof indeed finds efficient directions to perform adversarial attacks. Specifically, compared with the common projected gradient descent (PGD) attack (see Appendix F.1 on Page 42), the difference of our mechanism is that it leverages a learning rate $\eta_i(t)$ and a total search time $S$ to control the adversarial strength (i.e., the strength of the ability to make models misbehave), while PGD uses a perturbation radius $\rho$ to control the perturbation scale. As long as $\eta_i(t) S$ is set to be large enough, our mechanism can find adversarial examples to effectively attack targeted DNNs, just like the PGD attack.
>
> Besides, from the perspective of the proof, the main difference in using the gradient flow-based attack instead of the PGD attack is that the gradient flow-based attack enables our proof to analyze AT dynamics without explicitly modeling the boundaries of adversarial perturbation spaces. Please also kindly refer to the first two paragraphs in Section 4.1 on Page 4 for a detailed discussion.
>
> **Q3:** *It would improve the paper if bound on the learning rate can be provided with respect to its training (or maybe some empirical results towards that).*
>
> **A3:** Thanks for your suggestion. However, we could not agree that providing bounds on the learning rate is a good idea, since as explained in **A1**, our theoretical result that "a wide DNN under AT behaves like a linearized DNN" does not depend on the scale of the learning rate. Instead, it mainly depends on if the network widths are large enough. The only requirement for the learning rate is that it should be continuously differentiable (see Assumption 3 on Page 6).
>
> Nevertheless, the only thing the scale of learning rate could affect is the converging behavior of DNNs during long-term AT. We have added an intuitive discussion about this impact in the revised version, please kindly refer to Appendix E.2 on Page 42 for details.
>
> **References:**
>
> [r1] Jacot et al. "Neural tangent kernel: Convergence and generalization in neural networks." NeurIPS 2018.
>
> [r2] Lee et al. "Wide neural networks of any depth evolve as linear models under gradient descent". NeurIPS 2019.
>
> [r3] Arora et al. "On exact computation with an infinitely wide neural net." NeurIPS 2019.

---

> ### Author Response · Authors · 2023-11-21
> **Follow-up on rebuttal**
>
> Dear Reviewer gmGK,
>
> Thanks for your time in reviewing our paper and valuable comments!
>
> Since the discussion stage is about to end, we are writing to kindly ask if our replies have satisfactorily addressed your concerns. Please kindly let us know if you have any additional concerns and we are happy to discuss more.
>
> Thank you very much!

---

### Official Review · Reviewer_S6eo · 2023-11-10

**Soundness:** 3 good
**Presentation:** 3 good
**Contribution:** 2 fair
**Rating:** 6
**Confidence:** 2

**Summary:**

This paper studied the robust overfitting issue in adversarial training. Specifically, the author studied the training dynamics of adversarial training under the NTK regime. Theoretical results show that after long training the term that captures the robustness will fade away and the trained network will degenerate to the network with standard training. The author further proposed an algorithm for adversarial training under the NTK regime and conducted experiment. Results show that the proposed algorithm outperforms standard adversarial training and vanilla NTK.

**Strengths:**

1. The paper provides a novel theoretical view in robust overfitting in adversarial training, and the theoretical analysis gives an intuition on why adversarial training fails on a simplified regime (NTK).
2. The paper further proposed an algorithm applicable to NTK models. Experiment results aligns with the theoretical conclusion in the paper.

**Weaknesses:**

My concerns are listed as the follows:

1. The proposed algorithm can only work under NTK regime. Though the corresponding results can serve as an empirical evidence of the theoretical conclusion, it seems that the proposed algorithm has limited practical application.
2. The assumption in the paper seems too strong comparing with practical setting. The author are encouraged to consider more practical setting like GD and cross entropy loss.
3. In the studied setting, the scale of pertubation is also related to the norm of $\partial_x \mathcal{L}$, rather than depending on $S$, which introduces a discrepancy between the setting studied and real-world setting.
4. Some missing references in convergence of DNN: [1], [2], [3]

[1]Li, Yuanzhi, and Yingyu Liang. "Learning overparameterized neural networks via stochastic gradient descent on structured data." Advances in neural information processing systems 31 (2018).

[2]Zou, Difan, et al. "Gradient descent optimizes over-parameterized deep ReLU networks." Machine learning 109 (2020): 467-492.

[3]Allen-Zhu, Zeyuan, Yuanzhi Li, and Zhao Song. "A convergence theory for deep learning via over-parameterization." International conference on machine learning. PMLR, 2019.

**Questions:**

Please refer to the "weakness" section

---

> ### Author Response · Authors · 2023-11-13
> **Request for clarifying the term "perturbation"**
>
> Dear Reviewer S6eo,
>
> Thanks for your detailed review and kind support. We are currently working on the rebuttal to carefully answer your questions.
>
> To fully address all your concerns, we would like to make sure that in your question 3, does the term **"perturbation"** refer to the **"adversarial perturbation occurring in the search for adversarial examples"**?
>
> We are a bit confused by this question 3 because you also mentioned that **the dependence between "the perturbation" and "the norm of $\partial_x \mathcal L$" in our setting** introduces **a discrepancy between our setting and the real-world setting**. But in the real-world setting, projected gradient descent (PGD) is usually adopted to search adversarial examples, **which also requires calculating gradients based on $\partial_x \mathcal L$** (see Appendix F.1 in page 41).
>
> A clarification would be very helpful for us to prepare our answers!
>
> Thanks & sincerely,
>
> The Author

---

> > ### Comment · Reviewer_S6eo · 2023-11-13
> >
> > Dear Authors,
> >
> > Sorry for the confusion. Yes, my question is that in equation (1), the constraint of the adversarial example is $||x'-x|| \leq \rho$. Then how is $||x_{i,t,s}-x_i|| \leq \rho$ be guaranteed?
> >
> > Thanks

---

> ### Author Response · Authors · 2023-11-17
> **To Reviewer S6eo (1/2)**
>
> Thank you for your thorough review and constructive comments. All your concerns have been carefully responded below. The manuscript is carefully revised accordingly. We sincerely hope our responses fully address your questions.
>
> **Q1:** *The proposed algorithm can only work under the NTK regime. Though the corresponding results can serve as empirical evidence of the theoretical conclusion, it seems that the proposed algorithm has limited practical application.*
>
> **A1:** First of all, we would like to highlight that the ultimate goal of this paper is to give a theoretical explanation for the robust overfitting phenomenon when adversarially training DNNs, but not designing a practical AT algorithm. The role of our empirical evidence in Section 6 is to verify our theoretical finding that the regularization matrix $\Xi(t)$ indeed captures the robustness brought by AT. It is also unexpected for us that the proposed Adv-NTK can achieve comparable robustness with vanilla AT. However, making Adv-NTK further practical could be a bit out of the scope of the research goal of this paper. Therefore, we will leave this interesting research direction in future studies.
>
> Besides, we respectfully note that improving the practicality of algorithms that work under the NTK regime is a very active research direction and a variety of works such as [r1, r2, r3, r4, r5] are trying to improve the performance and efficiency of NTK models in real-world applications. For example, [r1] have made a five-layer NTK model (with convolutional layer and global average pooling layer (GAP; [r6])) under standard training to achieve almost 80% test accuracy on CIFAR-10 dataset with only 10,000 training data. To make our Adv-NTK as practical as that in [r1], we need to adapt a variety of calculation techniques to our AT setting, which includes GAP layers [r6], Monte Carlo-based NTK/gradient estimation [r2], single-precision calculation, and kernel compression [r7]. However, such adaptations would raise a series of mathematical and engineering challenges, and this is also one of the reasons that we will leave the practicality problem of Adv-NTK in future research but not in this paper.
>
> **Q2:** *The assumption in the paper seems too strong compared with the practical setting. The authors are encouraged to consider more practical settings like GD and cross entropy loss.*
>
> **A2:** Thanks for your suggestions.
>
> **About gradient descent (GD).** The NTK theory with GD for standard training has been well studied in [r8, r9]. We believe that by generalizing the results in [r8, r9] to our AT setting, we can prove results such as ADK/NTK kernel convergence, prediction convergence, parameter convergence, and DNN linearization when considering AT with GD. The main technical challenge would arise from analyzing our proposed adversarial example search strategy under GD.
>
> Nevertheless, we would also like to again highlight that the main goal and contribution of this paper is to propose the first theoretical explanation for robust overfitting in DNNs. The generalization of our results to the GD setting will be left in future research.
>
> **About cross-entropy loss.** We respectfully note that existing NTK literature has not yet proved a closed-form training dynamics solution for standard DNN training under cross-entropy loss. Since our paper is built upon previous NTK research, for now, we are unable to generalize our close-form AT dynamics for squared loss (i.e., Theorem 3 and Corollary 1 in Section 4.3 on Page 6) to that for cross-entropy loss.
>
> However, please note that the squared loss is also a suitable replacement for cross-entropy loss in many practical applications. The reason is in practice, cross-entropy loss is usually used to train classification models. Fortunately, by simply replacing the categorical label-encoding (in cross-entropy loss) with the one-hot label-encoding (in squared loss), classification models can also be trained with squared loss.

---

> > ### Author Response · Authors · 2023-11-17
> > **To Reviewer S6eo (2/2)**
> >
> > **Q3:** *In the studied setting, the scale of perturbation is also related to the norm of $\partial_x \mathcal L$, rather than depending on $S$, which introduces a discrepancy between the setting studied and real-world setting.*
> >
> > **Clarification:** *My question is that in equation (1), the constraint of the adversarial example is $|| x' - x || \leq \rho$. Then how is $|| x_{i,t,s} - x_i || \leq \rho$ be guaranteed?*
> >
> > A3: Thanks for your question and clarification.
> >
> > Firstly, we would like to clarify that we did not try to guarantee $\|x_{i,t,s} - x_i\| \leq \rho$ in our gradient flow-based adversarial example searching. Instead, we use the learning rate $\eta_i(t)$ and the total perturbation time $S$ to control the adversarial example strength (i.e., the strength of the ability to make models misbehave).
> >
> > Besides, we respectfully note that the discrepancy between our gradient flow-based method and that in the real-world setting, for example, projected gradient descent (PGD; see Appendix F.1 on Page 42), is very small. Specifically, both our method and PGD can find adversarial examples that can significantly increase model training loss. The only difference is that our method leverages a learning rate $\eta_i(t)$ and a total search time $S$ to control the strength of adversarial examples, while for PGD, a perturbation radius $\rho$ is used to control the strength of adversarial examples. From the perspective of searching adversarial examples during AT, such a difference is small. Please also kindly refer to the first two paragraphs in Section 4.1 on Page 4 for a detailed discussion.
> >
> > Finally, please note that both our gradient flow-based method and PGD need to (iteratively) calculate attack directions based on $\partial_x \mathcal L$. Therefore, perturbations generated by both the two methods are related to the norm of $\partial_x \mathcal L$.
> >
> > **Q4:** *Some missing references in convergence of DNN: [r10], [r11], [r12].*
> >
> > **A4:** Thanks. We have cited them in the revised version.
> >
> >
> > **References:**
> >
> > [r1] Han et al. "Fast neural kernel embeddings for general activations." NeurIPS 2022.
> >
> > [r2] Novak et al. "Fast finite width neural tangent kernel." ICML 2022.
> >
> > [r3] Zandieh et al. "Scaling neural tangent kernels via sketching and random features." NeurIPS 2021.
> >
> > [r4] Arora et al. "Harnessing the power of infinitely wide deep nets on small-data tasks." ICLR 2020.
> >
> > [r5] Novak et al. "Neural tangents: Fast and easy infinite neural networks in Python." ICLR 2020.
> >
> > [r6] Arora et al. "On exact computation with an infinitely wide neural net." NeurIPS 2019.
> >
> > [r7] Abedsoltan et al. "Toward large kernel models." ICML 2023.
> >
> > [r8] Du et al. "Gradient descent provably optimizes over-parameterized neural networks." ICLR 2019.
> >
> > [r9] Lee et al. "Wide neural networks of any depth evolve as linear models under gradient descent". NeurIPS 2019.
> >
> > [r10] Li et al. "Learning overparameterized neural networks via stochastic gradient descent on structured data." NeurIPS 2018.
> >
> > [r11] Zou et al. "Gradient descent optimizes over-parameterized deep ReLU networks." Machine learning (109) 2020.
> >
> > [r12] Allen-Zhu et al. "A convergence theory for deep learning via over-parameterization." ICML 2019.

---

> ### Author Response · Authors · 2023-11-19
> **Follow-up on rebuttal**
>
> Dear Reviewer S6eo,
>
> Thank you again for your valuable comments! We have carefully considered your comments and have provided our responses.
>
> Please let us know if our replies have satisfactorily addressed your concerns. Please do not hesitate to let us know if you have any further questions or if you require any additional clarification.
>
> Thank you very much!

---

> > ### Comment · Reviewer_S6eo · 2023-11-20
> >
> > Dear Authors:
> >
> > Thank you very much for your response.
> >
> > The authors have addressed most of my concerns and I will make necessary changes to my review and rating.
> >
> > Thanks,
> >
> > Reviewer S6eo

---

> > > ### Author Response · Authors · 2023-11-20
> > > **Thanks!**
> > >
> > > Thank you very much for recognizing our contributions and kind support!

---

### Meta-Review · Area_Chair_bSYQ · 2023-12-05

**Metareview:**

This paper studies adversarial training (AT) using the neural tangent kernel (NTK) theory and explains a phenomenon called robust overfitting. It then introduces an algorithm Adv-NTK for infinite-width DNNs to counter the robust overfitting, demonstrating comparable robustness to finite-width counterparts in experiments. With unanimous support from the reviewers, I recommend the acceptance of this paper.

**Justification For Why Not Higher Score:**

The impact of this work is hindered by the limitations of the neural tangent kernel (NTK) theory.

**Justification For Why Not Lower Score:**

The theoretical results based on NTK seem valuable, and the proposed algorithm is of potential practical interest.

---

### Decision · Program_Chairs · 2024-01-16

Accept (poster)